## Registered report

psychology

motivation, solitude, self-isolation, COVID-19, loneliness

**Author for correspondence:**
Netta Weinstein
e-mail: n.weinstein@reading.ac.uk

†Joint first authors.

# Motivation and preference in isolation: a test of their different influences on responses to self-isolation during the COVID-19 outbreak

## Netta Weinstein[1,2,†] and Thuy-Vy Nguyen[3,†]

[1]School of Psychology and Clinical Language Sciences, University of Reading, Reading, UK
[2]School of Psychology, Cardiff University, Cardiff, UK
[3]Department of Psychology, Durham University, Durham, UK

 NW, 0000-0003-2200-6617; T-VN, 0000-0003-0777-4204

This multi-wave study examined the extent that both preference and motivation for time alone shapes ill-being during self-isolation. Individuals in the USA and the UK are self-isolating in response to the COVID-19 outbreak. Different motivations may drive their self-isolation: some might see value in it (understood as the identified form of autonomous motivation), while others might feel forced into it by authorities or close others (family, friends, neighbourhoods, doctors; the external form of controlled motivation). People who typically prefer company will find themselves spending more time alone, and may experience ill-being uniformly, or as a function of their identified or external motivations for self-isolation. Self-isolation, therefore, offers a unique opportunity to distinguish two constructs coming from disparate literatures. This project examined preference and motivation (identified and external) for solitude, and tested their independent and interacting contributions to ill-being (loneliness, depression and anxiety during the time spent alone) across two weeks. Confirmatory hypotheses regarding preference and motivation were not supported by the data. A statistically significant effect of controlled motivation on change in ill-being was observed one week later, and preference predicted ill-being across two weeks. However, effect sizes for both were below our minimum threshold of interest.

# 1. Introduction

The purpose of this study was to understand how motivation and preference for solitude—two conceptually distinct constructs from disparate literatures that might have unique or interacting effects on individuals' responses to being in solitude—influence ill-being during self-isolation. To achieve this purpose, we focused on solitude during self-isolation operationalized in terms of both physical and social isolation, arguably its purest form [1].

When self-isolating, individuals reduce the number of others with whom they have contact by staying at home, but their physical isolation (being physically separated from others) and social isolation (being physically separate from, and also not interacting with, others) can vary. Many, including those living with others as well as those living alone, are likely to resort to virtual interactions, which replace in-person interactions as the main source of communication [2]. For those who live with others, time without virtual interactions can still be rich with in-person interactions. For those living alone, time without virtual interactions means truly being in solitude. Because our interest is in the influence of both preference and motivation for time alone on the psychological consequences for adults, we focus on a context when psychological isolation becomes the most significant issue: when living-alone adults do not interact with anyone in-person and virtually.

While self-isolating is an effective strategy to 'flatten the curve' of coronavirus infections by preventing the risks of contracting the virus, it may have psychological consequences studied in past research. To the extent that self-isolation results in loneliness, it may increase cardiovascular activation and cortisol levels, and reduce sleep, interfering with health and recovery [3]. Isolation is furthermore a risk factor for depression and anxiety among other indicators of ill-being [4,5]. Dispositional preference to be alone, and motivation for self-isolation, may both mitigate the potential psychological costs [1], though little is understood about their independent and interdependent contributions. This study will build our understanding of the roles these constructs play using transparent and robust research methods.

## 1.1. Preference and motivation for solitude

Few conclusions can be drawn from the existing literature about how preference for solitude predicts ill-being associated with time spent in isolation. Research shows that people who prefer to be alone [6] are broadly vulnerable and report ill-being, although there are mixed findings concerning their ill-being when they are in solitude, specifically [7]. Although no research of which we are aware has investigated this, it is also plausible that preference for solitude would lead individuals to tolerate it better and therefore report *less* ill-being when self-isolating.

A separate literature, self-determination theory (SDT), highlights the importance of motivation driving behaviour [8]. Motivations are 'hot', energizing reasons for behaviour in contrast to the 'cold' cognition characterizing preference; both meaningfully contribute to behaviour [9]. Based in SDT, less autonomous or self-driven motivation for solitude, and more controlled motivation reflecting pressure and choicelessness, relate to feeling lonely when alone (Nguyen *et al*. [1] Study 4). Further, endorsing autonomous reasons for spending time alone correlates negatively with ill-being outcomes like depression, loneliness and anxiety [10]. However, these studies have conflated enjoying being alone with finding value in the activity, because both are theorized to comprise autonomous motivation. One form of autonomous motivation is identified motivation, selecting to spend time alone because it is seen as being beneficial and valuable [1]. Identified motivation is particularly relevant in the case of self-isolation, since the reason for this behaviour is its importance to the health of the self-isolating individual and/or to society rather than because it is intrinsically enjoyable or rewarding—another aspect of autonomous motivation. Therefore, the extent to which this motivation is endorsed needs to be examined in its own right rather than as part of an autonomy composite. Further, one form of controlled motivation, external motivation or feeling pressured by others and choiceless [8], is relevant to instances of self-isolation which are felt to be driven by pressures from societal (government guidelines or mandates) or closer relationship (family, friends and doctors). Further, external motivation is important to understand the outcomes of solitude: forced solitude is thought to be the most detrimental form of solitude, and perhaps *only* when solitude is forced does it result in ill-being [11]. The current study will, therefore, examine the main effects of both identified (autonomous) and external (controlled) motivations—'hot' energizing reasons to self-isolate, and their moderating effects on the relationship between preference, a 'cold' reason, and ill-being.

## 1.2. Present research

Though preference and motivation for solitude show different patterns of influence on ill-being, those who prefer solitude often also report more autonomous motivation for it [1,10,12]. However, self-isolation is an occasion in which one's longstanding preference to be alone versus in the company of others, and one's current motivations to be alone, should be more strongly differentiated. Those who typically prefer social situations may now find additional value in time spent alone (e.g. for protecting their health), or alternatively they may feel pressured or choiceless around self-isolating. It may be either their preference or their motivation driving effects of ill-being during the time spent alone. Self-isolation is, therefore, ideal for gaining a deeper theoretical understanding of these two predictors of ill-being in solitude.

We pre-registered and collected Time 1 (14 March) of the research (OSF.IO/MN8KX) to minimize the amount of time respondents had spent in self-isolation before the start of the study. Following an initial review of the Registered Report, we collected Time 2 just over one week later (22 March) and Time 3 one week after Time 2 (29 March). The verbatim hypothesis, methodological and analytic sections of the Stage 1 manuscript were registered following Stage 1 in-principle acceptance and before Time 3 data collection at OSF (https://mfr.de-1.osf.io/render?url=https://osf.io/g9dxb/?direct%26mode=render%26action=download%26mode=render). See also the full Stage 1 accepted manuscript at OSF (https://mfr.de-1.osf.io/render?url=https://osf.io/y9xbt/?direct%26mode=render%26action=download%26mode=render), and the originally submitted research plan as an electronic supplementary material, table.

We had initially planned to collect two, not three time-points, and that those would take place two weeks apart. The move to weekly data collection and three time-points was made following reviewers' advice to recognize the quickly changing nature of communications related to COVID-19. For example, the UK Government announced school and bar/pub closures to take place by the end of 20 March 2020 (The Guardian, https://www.theguardian.com/world/2020/mar/20/london-pubs-cinemas-and-gyms-may-close-in-covid-19-clampdown). Furthermore, although the US Government had not yet made any decisions to take more stringent social distancing measures, we also expected a shift in motivation and perceived urgency of self-isolation on 23 March 2020, following an earlier recommendation by the US Centers for Disease Control and Prevention (CDC) to prevent gatherings of more than 50 people, and after state-wide school closures announced by more than 30 states on that date (CNN, https://edition.cnn.com/2020/03/15/health/us-coronavirus-sunday-updates/index.html). It seems, then, that weekly, not bi-weekly, measurements of ill-being were needed to understand ill-being before and after these significant changes to people's understanding and expectations for self-isolation.

We set out to test four sets of hypotheses guided by current theory and research on preference and motivation for solitude. Simply, these concerned the potential effects of preference on ill-being at Times 2 and 3 (Hypotheses 1:1, 1:2), of identified and external motivations on ill-being at Times 2 and 3, above and beyond preference (Hypotheses 2:1, 2:2), and the possible moderating effects of identified (Hypotheses 3:1, 3:2, 3:3) and external (Hypotheses 4:1, 4:2, 4:3) motivations on the main effect of preference and ill-being at Times 2 and 3.

### 1.2.1. Competing Hypothesis Set 1

For the link between preference for solitude and psychological consequences of isolation, we had competing hypotheses as follows:

*Hypothesis 1:1.* Preference for solitude would yield a negative correlation with ill-being one and two weeks later when participants report about time spent alone in the past week. This was based on the idea that for individuals with a greater preference for being alone, isolation would be a more pleasant experience.

*Hypothesis 1:2.* Alternatively, based on the previous positive correlation between preference for solitude and general loneliness [6], preference for solitude would be positively correlated with ill-being. This was based on the idea that preference for solitude is a symptom of psychological vulnerability (i.e. chronic loneliness) manifesting as negative emotions during the time in isolation.

### 1.2.2. Hypothesis Set 2

Research suggests that autonomous, and less controlled, motivations for solitude shape ill-being during solitude experiences (e.g. [10]). We focused on one form of autonomous motivation (identified motivation, or acting because of the value and importance of the activity) and one form of controlled

motivation (external motivation, or acting because one feels externally compelled and choiceless). We predicted that identified and external motivation for solitude would predict ill-being one and two weeks later. Importantly, we anticipated these relations would be in evidence above and beyond Time 1 preference for solitude, as well as Time 1 ill-being, and controlling for covariates listed above.

*Hypothesis 2:1.* Identified motivation for solitude would negatively correlate with residual change in ill-being one and two weeks later, above and beyond Time 1 preference for solitude.

*Hypothesis 2:2.* External motivation for solitude would positively correlate with residual change in ill-being one and two weeks later, above and beyond Time 1 preference for solitude.

## 1.2.3. Competing Hypothesis Set 3

It has been suggested, but not tested, that 'healthy' motivation for solitude (identified motivation and less external motivation) reflects adaptive self-regulatory capacity that might moderate the effects of preference for solitude [7]. In other words, healthy motivation for solitude might explain when the preference for solitude would not relate to ill-being. We tested three competing hypotheses involving moderation effects of preference × identified motivation.

*Hypothesis 3:1.* Following Hypothesis 1:1, if the preference for solitude yields a negative correlation with ill-being, this would only be the case when the identified motivation for solitude is high (estimated at 1 standard deviation (s.d.) above mean (*M*)) relative to when the identified motivation for solitude is low (−1 s.d.). In other words, we expected to see a *stronger* negative correlation between preference for solitude and residual change in ill-being from Times 1 to 2 and Times 1 to 3 at +1 s.d. level of identified motivation, relative to the negative correlation between preference for solitude and residual change in ill-being across one and two weeks at −1 s.d. level of identified motivation.

*Hypothesis 3:2.* Following Hypothesis 1:2, if the preference for solitude yields a positive correlation with ill-being, this would only be the case when the identified motivation for solitude is low (−1 s.d.) relative to when the identified motivation for solitude is high (+1 s.d.). In other words, we expected to see a *stronger* positive correlation between preference for solitude and residual change in ill-being from Times 1 to 2 and Times 1 to 3 at −1 s.d. level of identified motivation, relative to the positive correlation between preference for solitude and residual change in ill-being across one and two weeks at +1 s.d. level of identified motivation.

*Hypothesis 3:3.* It may also be the case that motivation distinguishes healthy versus unhealthy preference for solitude. Burger [6] suggested that some people prefer alone time because of shyness and anxiety, while others might prefer being alone because of their enjoyment of solitude. We therefore predicted that preference for solitude would yield a positive correlation with ill-being for those at −1 s.d. level of identified motivation for solitude. On the other hand, preference for solitude would yield negative correlation with ill-being for those at +1 s.d. level of identified motivation for solitude. For this hypothesis, we were less interested in the magnitude of difference between the two correlation coefficients, but expected to see *opposite* directions of relationship between preference for solitude with residual change in ill-being across one and two weeks at different levels (+1 s.d. versus −1 s.d.) of identified motivation.

## 1.2.4. Competing Hypothesis Set 4

Finally, Storr [11] proposed that only when solitude is forced (external motivation) does it become a detriment to psychological health. Therefore, it might be the unhealthy motivation for solitude that explains when the preference for solitude relates to ill-being. We outlined three competing hypotheses involving moderation effects of preference × external motivation.

*Hypothesis 4:1.* Following Hypothesis 1:1, if the preference for solitude yields a negative correlation with ill-being, this would only be the case when the external motivation for solitude is low (−1 s.d.) relative to when the external motivation for solitude is high (+1 s.d.). In other words, we expected to see a *stronger* negative correlation between preference for solitude and residual change in ill-being across one and two weeks at −1 s.d. level of external motivation, relative to the negative correlation between preference for solitude and residual change in ill-being across one and two weeks at +1 s.d. level of external motivation.

*Hypothesis 4:2.* Following Hypothesis 1:2, if the preference for solitude yields a positive correlation with ill-being, this would only be the case when the external motivation for solitude is high (+1 s.d.) relative to when the external motivation for solitude is low (−1 s.d.). In other words, we expected to see a *stronger* positive correlation between preference for solitude and residual change in ill-being across one and two weeks at +1 s.d. level of external motivation, relative to the positive correlation

between preference for solitude and residual change in ill-being across one and two weeks at −1 s.d. level of external motivation.

Hypothesis 4:3. Similar to Hypothesis 3:3, it may be the case that motivation distinguishes healthy versus unhealthy preference for solitude. We, therefore, predicted that preference for solitude would yield a positive correlation with ill-being for those at +1 s.d. level of external motivation for solitude. On the other hand, preference for solitude would yield negative correlation with ill-being for those at −1 s.d. level of external motivation for solitude. For this hypothesis, we were less interested in the magnitude of difference between the two correlation coefficients, but expected to see *opposite* directions of the relationship between preference for solitude with residual change in ill-being across one and two weeks at different levels (+1 s.d. versus −1 s.d.) of external motivation.

# 2. Material and methods

## 2.1. Participants and procedure

### 2.1.1. Recruiting strategy

Participants living in the UK or the USA aged 35+ years who had reported living alone were recruited via Prolific.co. To ensure a good spread across ages, 200 slots were specifically set aside for those aged 65 years or older. The geographical restriction was included so that the study captured a sample at the start of the national push for self-isolation (which was set by the UK Government 2 days later, and the USA days after that; this is as compared with countries such as China and Italy, which had already enforced strict restrictions on movement). We, furthermore, selected for individuals living alone so that their self-isolation would reflect time spent in solitude (physically alone and not interacting with others). We anticipated that those who were self-isolating *with others* might have less, if not similar, amounts of solitude as they typically would, since they were confined to a shared space together with their cohabitants. Finally, the inclusion criterion of 35+ years of age was set because this age represents a transition to adulthood, rather than young adulthood [13]. The focus on adults was important for two reasons: First, since COVID-19 is harmful to adults and older adults at higher rates, we anticipated that self-isolation is more relevant to, and more likely in, these individuals as compared with young or emerging adults, typically considered under 35 years [13]. Second, since risky behaviour is higher in adolescents to young adults under age 35 years [14–16], we sought a sample that would make more responsible and consistent health decisions to self-isolate regardless of their personal preferences.

### 2.1.2. Data termination rule

Our aim was to recruit 800 people in the initial, Time 1, sample recruited on Prolific. We aimed for power (95%, alpha level 0.05) to detect an effect size as small as $f^2 = 0.016$ for additional variance explained by the main effects of preference and motivation and interaction of these two variables, above and beyond covariates. Following Stage 1 review, covariates were: gender, age, follow-up in-person social contact, follow-up virtual contact (average of chat, text and phone frequency), subjective health at Time 1, virus diagnosis or suspicion at follow-up (Y/N), health anxiety at follow-up, and the stressor checklist at follow-up). In G*power, we performed sensitivity test to estimate the increase in variance explained by each of our predictors of interest (main effects of preference and identified or external motivation, and their interaction) out of a total of 11 variables entered into the regression model (two main effects, one interaction and eight covariates). This conservative effect size $f^2 = 0.016$ represents approximately 1.6% of variance explained by each of the main effects and interaction, after removing the variance explained by covariates. However, assuming up to 50% data loss due to attrition across the two time-points, with $n = 400$, we expected to have 95% power to detect an effect size as small as $f^2 = 0.033$.

### 2.1.3. Participants' characteristics and data collection

We received responses from 823 adults living in the UK and the USA through the online platform Prolific.co and these compose the final Time 1 sample.[1] Of these, 457 (55.46%) were women, 363

---

[1]We had two kinds of additional responses. First, we received duplicate and empty responses that our online software, Qualtrics, recorded but we excluded from all analyses. Second, we also shared the survey on social media platforms (i.e. Twitter, Facebook) with those who met one prescreen criterion: residing in a single person household (living alone). At the time of submission of this

(44.05%) were male and 2 (0.20%) reported another gender (1 did not respond). Their ages ranged from 24 to 87 years ($M = 52.93$, s.d. $= 12.13$).[2] Further examining this discrepancy with our recruitment selection criterion of being 35+ years of age, we saw that five participants reported an age below this (of which four were 32 years or older). Because we did not plan to exclude them at the data analysis step, they were retained. Although we selected only for individuals stating they are living alone, some participants reported marital status that implied they are living together: married $n = 29$ (3.5%) and living together as married $n = 23$ (2.8%), though the majority were single or never married 434 (52.67%), divorced 173 (21.00%), separated 28 (3.40%), or widowed 61 (7.40%). Although the living together as the married group reflected participants who should not have been eligible to take part in the study, they comprised a small part of the sample and we retained them because we had not set this as an exclusion criterion. Their employment status also varied, with 323 (39.20%) reporting full-time work; 99 (12.01%) part-time; 120 (14.56%) self-employment and 178 (21.60%) reporting they are retired. Finally, they varied by the location of their residence in a large city, 212 (25.73%), small city 257 (31.19%), rural area 105 (12.74%) and suburban area 249 (30.22%).

Guided by feedback received during Stage 1 review, the research team contacted participants for Time 2 of the project on 22 March 2020, just over one week after Time 1 survey was launched (14 March 2020), and for Time 3 on 29 March 2020, one week after Time 2. All participants who took part in Time 1 were contacted and given a chance to take part in Times 2 and 3.[3] We received ethical approval from Durham University's Ethics Committee (PSYCH-2020-03-11T23_41_08). Participants provided informed consent before taking part, received an in-part debrief following Wave 1 of the research, and were invited to skip items or withdraw from the study at any time.

## 2.2. Background measures

The full set of measures collected for all time-points are available at OSF.IO/MN8KX.

### 2.2.1. Subjective health (Time 1; covariate)

We assessed subjective health with one item from the World Value Survey (http://www.worldvaluessurvey.org/wvs.jsp). Specifically, participants were asked 'All in all, how would you describe your state of health these days? Would you say it is…', and received the options: *Very poor*; *Poor*; *Fair*; *Good*; *Very good*; *I'm not sure*.

### 2.2.2. Frequency of time spent alone (Times 1–3; quality check)

At Time 1, participants were asked: 'Considering the waking time you have, how much time *per day* do you typically spend *by yourself at home*?', and reported with options between 0 and 20+ waking hours. Additionally, participants were asked: 'How frequently in the past two weeks have you performed your daily activities at home by yourself?', using a scale with the anchors: *Most of the day*, *A few times a day*, *Once a day*, *Several times a week*, *Once a week* and *Almost never*. At Times 2, and 3, participants reported on these items, reflecting on the past week and on how much time they spent at home alone, '…not interacting with anyone in-person or virtually'. See Table 1 for means and correlations of these items and other measurements of the quantity of solitude.

### 2.2.3. Frequency of time spent with others (Times 1–3; quality check and covariate)

Participants also reported on the frequency of social interactions using instructions from Rook [17]. They were given the prompt: 'Think about the people with whom you socialize or enjoy conversations on a regular basis. How often do you interact with these people in general?', which was followed by four items specifying: In-person (face-to-face), Online (social media), Phone or Text (written messages). These items were paired with a Likert-type scale including (1 = *hourly or several times a day*, 2 = *once a day*, 3 = *every couple of days*, 4 = *once a week*, 5 = *less than once per week* and 6 = *not at all*). Responses to all

proposal, 18 people have filled out the survey. Because the number achieved through social media was too small for independent analyses and the criteria and recruiting method were different, we will exclude these participants from analyses.

[2]Of the sample, 32.4% were below 45 years of age, 23.7% were aged 45–54 years, 20.5% were aged 55–64 years, 20.3% were 65–74 years and 3.1% were 75 years or older.

[3]It should be noted that although the study was launched on 14 March, not all participants completed the Time 1 survey on the day, so some of them might not have the full one-week gap between Time 1 and surveys.

**Table 1.** Means (M), standard deviations (s.d.) and correlations (for variables measured at the same time-point) for continuous indicators of solitude, including continuous quality indicators, across three time-points.

| | T1 | T2 | T3 | same time-point correlations M T1–T3[b] | | | | | | | | | |
|---|---|---|---|---|---|---|---|---|---|---|---|---|---|
| | M (s.d.) | M (s.d.) | M (s.d.) | 1 | 2 | 3 | 4 | 5 | 6 | 1 | 2 | 3 | 4 |
| 1. No. of waking hours alone | 10.27 (5.34) | 11.52 (5.54) | 12.14 (5.25) | | 0.41** | −0.25** | −0.18** | 0.13** | 0.16** | | | | |
| 2. Frequency of alone activities | 5.36 (1.13) | 5.23 (1.29) | 5.28 (1.32) | 0.36** | | −0.14** | −0.09* | 0.11** | 0.06 | 0.29** | | | |
| 3. In-person interactions | 4.04 (1.43) | 2.78 (1.73) | 2.16 (1.60) | −0.44** | −0.19** | | 0.15** | −0.26** | −0.15** | −0.23** | −0.17** | | |
| 4. Virtual interactions | 4.03 (1.18) | 4.08 (1.21) | 4.00 (1.70) | −0.19** | −0.06 | 0.33** | | 0.09* | 0.13** | −0.16** | −0.05 | 0.12** | |
| 5. Endorsing isolating (no = 0; somewhat/yes = 1) | 0.48 (0.50) | 0.85 (0.36) | 0.88 (0.32) | 0.11** | 0.04 | −0.05 | 0.08 | | | 0.11** | 0.15** | −0.17** | 0.04 |
| 6. % of time spent alone as deliberate self-isolation[a] | 46.79% (30.95%) | 71.11% | 80.00% (0.27) | 0.18** | 0.02 | −0.02 | 0.05 | | | 0.08* | 0.04 | −0.18** | 0.13* |

[a]Data are only available for those who answer yes or somewhat to endorsing isolating.

[b]Correlations between variables at Time 1 are presented in triangle on the left, those at Time 2 are presented in the triangle in the middle, and those at Time 3 are presented in triangle on the right.

*p < 0.05; **p < 0.001; based on sample sizes > 614.

items were reverse-coded so that higher scores reflected higher levels of interactions. Then, items for online, phone and text messages were averaged to create a single 'virtual interactions' score ($\alpha_{\text{time 1}}$ = 0.56, $\alpha_{\text{time 2}}$ = 0.55 and $\alpha_{\text{time 3}}$ = 0.55).

### 2.2.4. Self-isolation (Times 1–3; quality check)

Participants responded to items: 'In the past week, did you self-isolate in response to the coronavirus outbreak (COVID-19)?' with options including *no*, *somewhat/in part* and *yes*. At Time 1, the percentage of participants who endorsed the 'somewhat/in part' or 'yes' responses was: 48%, at Time 2: 85% and at Time 3: 88%. If responding 'somewhat/in part' or 'yes', they were asked: 'In the past week, what percentage of your time spent at home alone was deliberate self-isolation (keeping away from other people and places) in response to the coronavirus outbreak (COVID-19)?'; see table 1 for means, standard deviations and correlations with other measurements of solitude.

## 2.3. Ill-being measures Time 1 (Times 1–3)

### 2.3.1. Ill-being composite

An ill-being composite was constructed by averaging the scales of ill-being described below (composite $\alpha_{\text{time 1}}$ = 0.94, $\alpha_{\text{time 2}}$ = 0.93 and $\alpha_{\text{time 3}}$ = 0.94). All measured at Time 1 were paired with the prompt: 'In the past 6 months, I have felt…', and at Times 2 and 3 with the prompt: 'In the past week, when I was home alone and not interacting with anyone in-person or virtually, I have felt…'.

### 2.3.2. Loneliness

Participants reported on the Depletion and Isolation subscales of the Loneliness Rating Scale [18], and specifically on its frequency, since the frequency of loneliness is more influential than its intensity [19]. They responded on a scale of 1 (*never*) to 7 (*always*). Depletion items included 'empty', 'secluded', 'alienated', withdrawn' and 'numb' ($\alpha_{\text{time 1}}$ = 0.94, $\alpha_{\text{time 2}}$ = 0.95 and $\alpha_{\text{time 3}}$ = 0.95). Isolation items included 'unloved', 'worthless', 'hopeless', 'abandoned' and 'deserted' ($\alpha_{\text{time 1}}$ = 0.96, $\alpha_{\text{time 2}}$ = 0.96 and $\alpha_{\text{time 3}}$ = 0.96).

### 2.3.3. Depression

The shortened 10-item version of the Center for Epidemiologic Studies Depression Scale (CES-D) [20] was used to measure depression. Participants reported how often: 'I felt depressed', 'I felt everything I did was an effort' and 'I felt fearful', with a scale ranging from 1 (*not at all*) to 6 (*most of the time*) ($\alpha_{\text{time1}}$ = 0.91, $\alpha_{\text{time2}}$ = 0.92 and $\alpha_{\text{time3}}$ = 0.92). This measure was transformed into a seven-point scale before it was combined with other measures to calculate the ill-being composite. To do this, we applied the linear stretch method [21] using the following formula: new score = $(7-1)\times$(old score $-1$)/ $(6-1)+1$.

### 2.3.4. Anxiety

Anxiety was measured with a validated six-item measure of the State-Trait Anxiety Inventory (STAI) as recommended by Tluczek *et al.* [22].[4] Items were paired with a scale ranging from 1 (*not at all*) to 7 (*very much so*) ($\alpha_{\text{time 1}}$ = 0.93, $\alpha_{\text{time 2}}$ = 0.92 and $\alpha_{\text{time 3}}$ = 0.93).

## 2.4. Predictor variables Time 1 (baseline)

### 2.4.1. Preference for solitude

We took six items from the 12-item measure Preference for Solitude scale [6]. We chose items that were generally worded so that they apply to most people whereas items that refer to specific circumstances were not included (e.g. 'One feature I look for in a job is the opportunity to spend time by myself').

[4]This is a partial set of the State-Trait Anxiety Inventory for Adults items used with the permission of the publisher. The instructions and response scales have been modified for the present study. We included the authorized sample items of the copyrighted version in the electronic supplementary materials, Appendix B. These sample items are not authorized for reuse or modification.

We drew two items from each of the three factors identified by Cramer and Lake [23] to cover all dimensions of this construct. Instructions were as follows: 'For each of the following pairs of statements, select the one that best describes you. In some cases, neither statements may describe you well or both may describe you somewhat. In those cases, please select the statement that best describes you or that describes you more often.' Respondents selected one of two options per item, e.g. 'I enjoy being around people' or 'I enjoy being by myself' ($\alpha = 0.72$).

### 2.4.2. Motivation for solitude

Identified (autonomous) and external (controlled) motivations for solitude were measured with a 10-item scale adapted from Nguyen *et al.* [1]. Participants were asked: 'Think of times when you will be by yourself, at home, in the next two weeks. Those are times when you do not interact with anyone in-person or virtually', followed by items measuring autonomous motivation through its form most applicable to the context of self-isolation, identified motivation (finding value and importance in the activity [8]). Five items for this subscale included 'Having time to myself is important and beneficial to me' (original item) and 'Spending time alone is important for protecting my health' (new item). Overall internal reliability was acceptable ($\alpha = 0.73$). Participants also responded to five items measuring controlled motivation through its form most applicable to the context of self-isolation, external motivation (feeling external pressures and choicelessness), with items such as 'I am forced into it' (original item), 'I would get in trouble with others if I didn't' (original item) and 'I feel I have no choice' (new item). Items were paired with a seven-point scale ranging from 1 (*this does not apply to me at all*) to 7 (*this applies to me very much*). Overall internal reliability was high ($\alpha = 0.84$).

## 2.5. Descriptive and covariate measures Times 2 and 3 following Stage 1 review

### 2.5.1. COVID-19 concerns

Two items assessed COVID-19 health concerns. First, participants responded to the items: 'I have been diagnosed with COVID-19' (Yes/No). 'I strongly suspect that I currently have a COVID-19 infection' (Yes/No). Participants who reported 'Yes' to either of these items were considered as having serious concerns about COVID-19.

### 2.5.2. Health anxiety

Typically a disposition-level measure, participants also responded to five items of the Health Anxiety Inventory [24] that measure health anxiety at a state level.[5] Participants selected from four options reflecting increasing concerns. For example, one item is: 'I have not worried about my health', 'I have occasionally worried about my health', 'I have spent much of my time worrying about my health' and 'I have spent most of my time worrying about my health' ($\alpha_{time\ 2} = 0.87$ and $\alpha_{time\ 3} = 0.87$).

### 2.5.3. Stressors checklist

Participants were asked: 'In the past week, did you experience any of the following? (tick all that apply)', and selected the options: 'Lost your job', 'Lost a substantial amount of earnings from your job', 'Were worried that COVID-19 would impact your job security', 'Did not have enough food', 'Could not access food', 'Could not access important medical supplies', 'Knew someone who was diagnosed with COVID-19' or 'Were diagnosed with any new health condition'. See table 2 for frequencies of each stressor experienced at Times 1 and 2.

### 2.5.4. Overall depression

We used the same measure described above for testing depression in solitude to also test depression overall, with the instructions to participants to reflect on how they felt generally (that is presumably, both within and outside of solitude) in the past week, at Times 2 ($\alpha = 0.91$) and 3 ($\alpha = 0.92$). Following

---

[5]Originally, we picked out six items from the Short Health Anxiety Inventory (SHAI). However, after we launched the Time 2 survey, we caught one erroneous item that said 'I noticed aches and pains less than/more than usual'. We excluded this item from the composite, so the final SHAI measure had five instead of six items.

**Table 2.** Percentages of people experiencing stressors in the past week at Times 2 and 3 (calculated out of total participants enrolled in the study). Items rephrased to be slightly shorter.

|  | T2% | T3% |
|---|---|---|
| 1. Lost your job | 4.06 | 3.85 |
| 2. Lost substantial earnings from job | 17.96 | 17.74 |
| 3. Worried that COVID-19 impact job security | 36.78 | 35.61 |
| 4. Did not have enough food | 12.42 | 9.80 |
| 5. Could not access food | 21.03 | 16.87 |
| 6. Could not access important medical supplies | 8.36 | 6.08 |
| 7. Knew someone diagnosed with COVID-19 | 8.24 | 14.27 |
| 8. Diagnosed with new health condition | 2.09 | 1.74 |

Stage 1 review, this measure was intended to correlate against our ill-being in solitude measures and provide an exploratory test of the correlations between ill-being in solitude, and ill-being in general.

# 3. Results

## 3.1. Effect size estimates and interpretation

Informed by related solitude work described below, we set criteria for smallest effect sizes of interest in our hypotheses to predetermine standards for what we will decide as meaningful when evaluating the results instead of relying on $p$-values. We used $-0.15$ *partial* $r$s as the smallest effect size of interest (equivalent to 0.023% of variance explained) to determine whether main effect hypotheses (H1:1, H1:2, H2:1 and H2:2) are supported. For moderation effects hypothesized in H3:1, H3:2, H4:1 and H4:2, when calculating the relative difference between two correlation coefficients, we used 0.10 difference between the two standardized coefficients (for example, between $-0.40$ and $-0.30$, or $-0.20$ and $-0.10$) as the smallest effect size of interest to determine whether this hypothesis is supported.

### 3.1.1. Rationale for choosing partial r around 0.15

A previous paper by Nguyen *et al.* [12] looked at the link between autonomous motivation (conceptually similar to high identified motivation and low external motivation) and well-being and ill-being variables. In Study 1, sampling first-year university students ($n = 147$), the authors measured motivation for spending time alone and loneliness experienced in the past two weeks (not specifically related to time spent alone) and assessed these variables three times. On average, the correlation between autonomous motivation and loneliness was approximately $r = -0.30$. Extraversion—a concept similar to preference for solitude—also correlated with loneliness at also around $r = -0.30$. However, because these correlations were obtained from a rather small sample, effect sizes were probably overestimated [25]. As such, we selected to place our benchmark at a partial $r$ half the size of those correlations obtained by past research.

### 3.1.2. Rationale for choosing 0.10 difference between regression coefficients

In Study 2 of Nguyen *et al.* [12], the researchers also examined the interaction between participants' perceived belonging with autonomous motivation predicting levels of loneliness experienced in the past one month; this moderation test is the closest available data to our hypothesized moderation effects (Hypothesis Sets 3 and 4) and yielded a coefficient of, $\beta = -0.32$, 95% CI [$-0.44, -0.20$], when predicting loneliness. At higher levels of autonomous motivation, perceived belonging related to lower loneliness, $\beta = -0.04$, 95% CI [$-0.21, 0.13$]. Again, findings relied on a sample of $n = 223$, so we expect the effect sizes were probably overestimated. Selecting regression coefficients half the size of those coefficients obtained by past research, we anticipated $r = -0.16$ at low levels of autonomous motivation, and $r = -0.02$ at high levels. The difference between these two coefficients was about $\Delta r = 0.14$, but a conservative expectation is $\Delta r = 0.10$.

## 3.2. Exclusion criteria

We set three criteria to guide which participants would be excluded from analyses: First, we tested for inattention with two items 'Choose "somewhat agree"' and 'Choose "Very true"' embedded in two different scales of Times 2 and 3. We excluded participants who either (i) failed to select the corresponding answers as instructed or (ii) failed at least one attention check item (if the participant misses the item, that is considered failing the attention check). Second, we anticipated that we would exclude participants who responded after we submitted the Stage 1 Registered Report, but when we downloaded the data, we saw those were duplicates or missing responses. The procedure for removing duplicates was included in our analysis script (OSF.IO/MN8KX). Finally, we excluded participants if we could not link their data across at least two time-points.

## 3.3. Confirmatory quality checks

This study does not involve an experimental manipulation, and for this reason, we did not include a manipulation check. However, the assumption underlying our hypotheses is that our sample, on the whole, will find themselves spending more time alone than usual as a function of the COVID-19 outbreak. As such, we conducted three sets of analyses (based on measures described as quality indicators in the materials description above) to test our belief that any changes in ill-being from Time 1 to Time 2 are due to the effects of social isolation.

### 3.3.1. Time spent alone

We expected significant (using the $p < 0.05$ cut-off) increases in frequency (i.e. 'How frequently have you performed your daily activities at home by yourself?') and length (i.e. 'Considering the waking time you have had in the past week, how much time per day did you typically spend at home alone?') of time spent alone from Time 1 to Time 2 and Time 1 to Time 3. Results of two paired samples $t$-tests showed that on average participants performed daily activities by themselves at home significantly less frequently at Time 2 ($M = 5.23$, s.d. $= 1.29$) as compared with Time 1 ($M = 5.35$, s.d. $= 1.13$; $t_{703} =$ 2.37, $p = 0.018$). The difference between Times 1 and 3 ($M = 5.28$, s.d. $= 1.32$) did not reach statistical significance ($t_{701} = 1.39$, $p = 0.164$). This pattern was not as we expected.

On the other hand, participants reported spending increased numbers of waking hours alone from Time 1 ($M = 10.26$, s.d. $= 5.34$) to Time 2 ($M = 11.52$, s.d. $= 5.54$; $t_{748} = -6.18$, $p < 0.001$), and from Time 1 to Time 3 ($M = 12.14$, s.d. $= 5.25$; $t_{751} = -8.71$, $p < 0.001$). As we expected, on average people reported spending more hours at home alone in the second and third assessment.

### 3.3.2. In-person interactions

We expected a significant (at $p < 0.05$) drop in *in-person* interactions from Time 1 to Time 2 and Time 1 to Time 3. Results showed that, when compared with Time 1 ($M = 4.04$, s.d. $= 1.43$), people reported having in-person interactions less frequently at Time 2 ($M = 2.78$, s.d. $= 1.73$; $t_{749} = 19.43$, $p < 0.001$) and Time 3 ($M = 2.16$, s.d. $= 1.60$; $t_{751} = 27.36$, $p < 0.001$).

### 3.3.3. Self-isolation questions

We expected a statistically significant (at $p < 0.05$) increase in the percentage of people endorsing 'yes' and 'in part' to isolating because of COVID-19, and an increase in the percentage of time spent alone from Time 1 to Time 2 and Time 1 to Time 3. Results showed significantly higher percentages of participants endorsing 'yes' and 'in part' to isolating because of COVID-19 at Time 2 (85%; $t_{735} = -19.52$, $p < 0.001$) and Time 3 (88%; $t_{735} = -21.54$, $p < 0.001$) as compared with Time 1 (48%). Further, among those who answer 'yes' and 'in part', these people also reported spending a larger proportion of time alone at home deliberately in response to COVID-19 at Time 2 (71%; $t_{333} = -15.37$, $p < 0.001$) and Time 3 (80%; $t_{328} = -17.20$, $p < 0.001$) as compared with Time 1.

Overall, out of five quality checks, four showed expected patterns. Participants indeed engaged in less in-person interactions and spent more waking hours alone between Time 1 and the two follow-up time-points. In addition, more participants reported self-isolating in response to COVID-19 at Times 2 and 3 than at Time 1, and those who self-isolated also spent higher percentages of time alone self-isolating in response to COVID-19.

**Table 3.** Means, standard deviations (s.d.) and correlations between the three main predictors (preference for solitude, identified and external motivation for solitude), and indicators of ill-being, at Time 1 (T1), Time 2 (T2) and Time 3 (T3).

| | M | s.d. | scale | preference | identified motivation | external motivation |
|---|---|---|---|---|---|---|
| 1. Preference—solitude | 0.83 | 0.24 | 0–1 | | | |
| 2. Identified motivation | 4.93 | 1.26 | 1–7 | 0.39** | | |
| 3. External motivation | 2.91 | 1.53 | 1–7 | −0.13** | 0.14** | |
| 4. T1 depletion | 2.53 | 1.29 | 1–7 | 0.04 | −0.09* | 0.21** |
| 5. T2 depletion | 2.48 | 1.37 | 1–7 | 0.00 | −0.05 | 0.27** |
| 6. T3 depletion | 2.48 | 1.41 | 1–7 | −0.03 | −0.06 | 0.25** |
| 7. T1 isolation | 2.16 | 1.24 | 1–7 | 0.00 | −0.13** | 0.24** |
| 8. T2 isolation | 1.99 | 1.27 | 1–7 | −0.02 | −0.10** | 0.26** |
| 9. T3 isolation | 1.98 | 1.31 | 1–7 | −0.03 | −0.11** | 0.24** |
| 10. T1 depression | 2.89 | 1.15 | 1–6 | 0.01 | −0.09* | 0.25** |
| 11. T2 depression | 2.99 | 1.27 | 1–6 | −0.01 | 0.00 | 0.30** |
| 12. T3 depression | 2.99 | 1.29 | 1–6 | −0.05 | −0.03 | 0.28** |
| 13. T1 anxiety | 3.42 | 1.47 | 1–7 | 0.01 | −0.09** | 0.21** |
| 14. T2 anxiety | 3.52 | 1.55 | 1–7 | −0.03 | −0.02 | 0.27** |
| 15. T3 anxiety | 3.55 | 1.56 | 1–7 | −0.07 | −0.02 | 0.29** |

*$p < 0.05$; **$p < 0.001$.

## 3.4. Confirmatory analyses

### 3.4.1. Correlations

Table 3 presents the means, standard deviations and planned Pearson correlations of our three main predictors and specific ill-being variables at all time-points. We tested unadjusted links between preference, identified motivation and external motivation to ill-being at Times 2 and 3. Those who preferred to be alone reported more identified motivation ($r = 0.39$), and less external motivation ($r = −0.13$) for solitude, suggesting that there is some overlapping between these variables but only to a small degree. With 15% shared variance between preference and identified motivation and less than 2% shared variance between preference and external motivation, preference and motivation were distinct reasons for seeking solitude. Further, no relations were evident of preference with ill-being constructs at any of the time-points, and weak and mixed relations were evident of identified motivation with ill-being at Times 1–3. The most consistent correlations were observed between external motivation and ill-being, which averaged $r = 0.26$ across ill-being indicators and time-points (table 3). For readers' interest, table 4 also presents correlations between our three main predictors and ill-being, our outcome of interest, along with study covariates.

### 3.4.2. Primary models

As planned, primary analyses regressed the composite of our outcomes (loneliness-depletion, loneliness-isolation, depression and anxiety) at Times 2 and 3, controlling for the Time 1 ill-being composite. Main models testing our hypotheses accounted for eight potentially confounding effects: gender, age, follow-up in-person social contact, follow-up virtual contact (average of chat, text and phone frequency), virus diagnosis or suspicion (Y/N), health anxiety, subjective health at Time 1 and the stressor checklist at follow-up.

For each model, we first plotted the distribution of the residual errors to check for normality. All plots indicated that residual errors were normally distributed, and therefore no transformation was needed. We tested our hypotheses using linear regression models as the assumption of normality was satisfied.

***Hypothesis 1:1, Hypothesis 1:2, Hypothesis 2:1 and Hypothesis 2:2 (concerning main effects of preference and motivations on ill-being).*** In two hierarchical linear regression analyses, we regressed Times 2 and 3 ill-being on covariates (listed above) at Step 1, Time 1 preference for solitude at Step 2 and Time 1 identified and external motivation at Step 3. Findings are summarized in table 5 and

**Table 4.** Means, standard deviations (s.d.) and correlations between the three main predictors (preference for solitude, identified and external motivation for solitude), study covariates, and ill-being at Time 1 (T1), Time 2 (T2) and Time 3 (T3). dx = diagnosis or self-diagnosis.

| | M | s.d. | preference for solitude | identified motivation | external motivation | Time 1 ill-being | Time 2 ill-being | Time 3 ill-being |
|---|---|---|---|---|---|---|---|---|
| 1. Gender | — | — | 0.05 | 0.12** | −0.01 | −0.01 | 0.03 | 0.03 |
| 2. Age | 53.05 | 12.00 | −0.10** | −0.05 | −0.08* | −0.21** | −0.18** | −0.15** |
| 3. T1 subjective health | 3.80 | 0.95 | −0.07* | −0.01 | −0.08* | −0.39** | −0.33** | −0.33** |
| 4. T1 in-person social | 4.06 | 1.43 | −0.16** | 0.02 | 0.10** | −0.18** | −0.12** | −0.13** |
| 5. T2 in-person social | 2.80 | 1.73 | −0.07 | −0.07 | −0.03 | −0.02 | −0.07 | −0.06 |
| 6. T3 in-person social | 2.16 | 1.60 | 0.03 | 0.00 | −0.06 | −0.04 | −0.09* | −0.08* |
| 7. T1 virtual social | 4.00 | 1.19 | −0.15** | 0.05 | 0.05 | −0.08* | 0.02 | 0.01 |
| 8. T2 virtual social | 4.07 | 1.22 | −0.13** | 0.07 | 0.04 | −0.10** | −0.01 | −0.04 |
| 9. T3 virtual social | 4.00 | 1.27 | −0.11** | 0.10** | 0.02 | −0.09* | −0.01 | −0.02 |
| 10. T2 stressor checklist | 1.18 | 1.07 | 0.00 | 0.00 | 0.05 | 0.25** | 0.30** | 0.29** |
| 11. T3 stressor checklist | 1.13 | 1.00 | 0.02 | 0.03 | 0.14** | 0.24** | 0.29** | 0.32** |
| 12. T2 health anxiety | 1.87 | 0.62 | 0.04 | 0.13** | 0.27** | 0.40** | 0.60** | 0.55** |
| 13. T3 health anxiety | 1.86 | 0.63 | −0.01 | 0.11** | 0.25** | 0.41** | 0.56** | 0.61** |
| 14. T2 COVID-19 dx | 0.02 | 0.16 | 0.05 | 0.05 | −0.02 | 0.10** | 0.11** | 0.10** |
| 15. T3 COVID-19 dx | 0.05 | 0.21 | 0.03 | 0.03 | 0.09* | 0.12** | 0.14** | 0.17** |
| 16. T1 ill-being | 2.84 | 1.24 | 0.03 | −0.12** | 0.26** | | 0.77** | 0.77** |
| 17. T2 ill-being | 2.84 | 1.30 | −0.01 | −0.04 | 0.31** | | | 0.89** |
| 18. T3 ill-being | 2.85 | 1.34 | −0.05 | −0.05 | 0.30** | | | |

*p < 0.05; **p < 0.001.

**Table 5.** Results of primary models testing Hypothesis Sets 1 and 2 and predicting Times 2 and 3 ill-being from covariates, preference for solitude and motivation for solitude. All results for specific predictors are taken from the final step of the analysis, including all predictors (control and independent variables).

| | Time 2 ill-being | | | | | | Time 3 ill-being | | | | | |
|---|---|---|---|---|---|---|---|---|---|---|---|---|
| | | 95% CI | | partial | | | | 95% CI | | partial | | |
| | $\beta$ | lower | upper | $r^2$ | $t$ | $p$ | $\beta$ | lower | upper | $r^2$ | $t$ | $p$ |
| controls | | | | | | | | | | | | |
| 1. Gender[a] | 0.03 | −0.01 | 0.07 | 0.00 | 1.33 | 0.184 | 0.02 | −0.02 | 0.06 | 0.00 | 1.08 | 0.281 |
| 2. Age[a] | −0.02 | −0.06 | 0.03 | 0.00 | −0.75 | 0.454 | −0.01 | −0.05 | 0.03 | 0.00 | −0.47 | 0.642 |
| 3. Ill-being[a] | 0.61 | 0.56 | 0.66 | 0.45 | 24.17 | 0.000 | 0.59 | 0.54 | 0.64 | 0.43 | 23.48 | 0.000 |
| 4. Subjective health[a] | 0.03 | −0.01 | 0.08 | 0.00 | 1.55 | 0.122 | 0.03 | −0.01 | 0.08 | 0.00 | 1.38 | 0.169 |
| 5. In-person interactions[b] | −0.03 | −0.07 | 0.01 | 0.00 | −1.61 | 0.108 | −0.03 | −0.07 | 0.01 | 0.00 | −1.34 | 0.181 |
| 6. Virtual interactions[b] | 0.01 | −0.03 | 0.05 | 0.00 | 0.53 | 0.599 | −0.02 | −0.07 | 0.02 | 0.00 | −1.13 | 0.257 |
| 7. Stressor checklist[b] | 0.08 | 0.03 | 0.12 | 0.02 | 3.59 | 0.000 | 0.09 | 0.04 | 0.13 | 0.02 | 3.95 | 0.000 |
| 8. Health anxiety[b] | 0.34 | 0.29 | 0.38 | 0.22 | 14.45 | 0.000 | 0.35 | 0.30 | 0.39 | 0.22 | 14.49 | 0.000 |
| 9. COVID-19 diagnosis[b] | 0.00 | −0.05 | 0.04 | 0.00 | −0.16 | 0.875 | 0.03 | −0.02 | 0.07 | 0.00 | 1.22 | 0.225 |
| preference | | | | | | | | | | | | |
| 10. Preference for solitude[a] | −0.03 | −0.07 | 0.02 | 0.00 | −1.24 | 0.214 | −0.06 | −0.10 | −0.01 | 0.01 | −2.48 | 0.013 |
| motivation | | | | | | | | | | | | |
| 11. Identified motivation[a] | −0.02 | −0.07 | 0.02 | 0.00 | −0.94 | 0.346 | −0.01 | −0.05 | 0.04 | 0.00 | −0.27 | 0.787 |
| 12. External motivation[a] | 0.06 | 0.01 | 0.10 | 0.01 | 2.61 | 0.009 | 0.04 | 0.00 | 0.08 | 0.00 | 1.77 | 0.077 |
| **Variance explained at each step** | | | | | | | | | | | | |
| Step 1 (#1–9 added) | $R^2 = 0.70$ | | | | | | $R^2 = 0.69$ | | | | | |
| Step 2 (#10 added) | $\Delta R^2 = 0.01$ | | | | | | $\Delta R^2 = 0.01$ | | | | | |
| Step 3 (#11, 12 added) | $\Delta R^2 = 0.01$ | | | | | | $\Delta R^2 = 0.00$ | | | | | |

[a]Measured at Time 1.

[b]Measured at Times 2 and 3, according to the measurement of the outcome variable.

showed that preference and both types of motivation for solitude explained little variance in residual change in ill-being from Time 1 to Time 2, as well as from Time 1 to Time 3. The external motivation for time alone predicted an increase in Time 2 ill-being ($\beta = 0.06$, 95% CI [0.01, 0.10], $p = 0.009$), yielding a (*partial r*) *pr* of 0.10, which is smaller than the effect size of interest we determined *a priori*. Likewise, preference for solitude predicted a decrease in Time 3 ill-being ($\beta = -0.06$, 95% CI [−0.10, −0.01], $p = 0.013$), yielding a *pr* of 0.10, which is also smaller than the effect size of interest.

*Hypothesis 3:1, Hypothesis 3:2 and Hypothesis 3:3 (concerning interacting effects with identified motivation).* In two hierarchical linear regression analyses, we regressed follow-up (Times 2 and 3) ill-being on covariates at Step 1, Time 1 preference for solitude and Time 1 identified motivation at Step 2, and the interaction term of preference for solitude and identified motivation for solitude at Step 3. Findings are summarized in table 6 and did not show evidence of interaction between preference for solitude and identified motivation for time alone in either Time 2 or Time 3 analysis.

*Hypothesis 4:1, Hypothesis 4:2 and Hypothesis 4:3 (concerning interacting effects with external motivation).* In two hierarchical linear regression analyses, we regressed follow-up (Times 2 and 3) ill-being on covariates at Step 1, Time 1 preference for solitude and Time 1 external motivation at Step 2, and the interaction term of preference for solitude and external motivation for solitude at Step 3. As summarized in table 7, we also did not find evidence supporting our hypothesis that there is an interaction between preference for solitude and identified motivation for time alone in either Time 2 or Time 3 analysis.

## 3.5. Summary of confirmatory analyses

Planned unadjusted correlations showed a consistent link between external motivation and higher ill-being as measured through loneliness-depletion, loneliness-isolation, depression and anxiety at all time-points. However, after accounting for controls including Time 1 ill-being, we found little evidence suggesting that initial preference for solitude and motivation behind spending time alone were predictive of change in ill-being across a period of two weeks. Instead, we found that health anxiety and life stressors, measured simultaneously with the outcome variables, linked strongly with increases in ill-being at Times 2 and 3. To the extent that individuals felt anxiety surrounding their health, they reported increased ill-being in their solitude reported at Times 2 and 3. Further, with more stressors experienced in the past week (of which job and food insecurity were most predominant, table 2), participants reported more ill-being during their solitude across two weeks. Only health anxiety, however, yielded *prs* > 0.45 at both Times 2 and 3, which is larger than our smallest effect size of interest, whereas life stressors yielded an independent effect size below our predetermined threshold at both Times 2 and 3. Other characteristics like age and gender, subjective health, frequency of in-person interactions or virtual interactions in the past week also did not independently predict Times 2 and 3 ill-being in our fully adjusted models.

# 4. Exploratory analyses

We conducted additional exploratory—unplanned—tests to better understand how change in ill-being occurred across the two weeks of the study and what this means for our three main predictors.

## 4.1. Change in ill-being across early weeks of self-isolation

First, we were interested in the extent participants experienced change in ill-being over the course of two weeks, over and above their pre-existing levels of ill-being. We sought to test, in part, public concerns that self-isolation in itself would yield loneliness and depression (e.g. CNBC.com, 2020; TheGuardian.com, 2020; Health.com, 2020; NewYorker.com, 2020). On one hand, our sample may more easily access online resources since they participated in an online study, but on the other hand, they represented older adults that have been the focus of primary concern (average age approximately $M = 52$ years), who are living alone—presumably both of these were risk factors for more ill-being.

Paired sample *t*-tests comparing Time 1 ill-being with Times 2 and 3 ill-being suggested that levels of ill-being over two weeks did not differ from participants' prior ill-being levels. At the beginning of the study, the average level of ill-being experienced in the past six months was $M = 2.84$, s.d. = 1.24. One week after, the average ill-being was no different ($M = 2.84$, s.d. = 1.30; $t_{749} = 0.17$, $p = 0.863$), and two weeks after, the average ill-being was no different ($M = 2.85$, s.d. = 1.34; $t_{751} = -0.49$, $p = 0.626$). When we

**Table 6.** Results of primary models testing Hypothesis Set 3 predicting Times 2 and 3 ill-being from covariates, preference and identified motivations for solitude, and preference × identified motivation. All results for specific predictors are taken from the final step of the analysis, including all predictors (control and independent variables).

| | Time 2 ill-being | | | | | | Time 3 ill-being | | | | | |
| | | 95% CI | | partial | | | | 95% CI | | partial | | |
| | $\beta$ | lower | upper | $r^2$ | $t$ | $p$ | $\beta$ | lower | upper | $r^2$ | $t$ | $p$ |
|---|---|---|---|---|---|---|---|---|---|---|---|---|
| **controls** | | | | | | | | | | | | |
| 1. Gender[a] | 0.03 | −0.01 | 0.07 | 0.00 | 1.26 | 0.209 | 0.02 | −0.02 | 0.06 | 0.00 | 1.06 | 0.287 |
| 2. Age[a] | −0.02 | −0.06 | 0.03 | 0.00 | −0.80 | 0.421 | −0.01 | −0.06 | 0.03 | 0.00 | −0.54 | 0.589 |
| 3. Ill-being[a] | 0.62 | 0.57 | 0.67 | 0.47 | 25.18 | 0.000 | 0.60 | 0.55 | 0.65 | 0.45 | 24.24 | 0.000 |
| 4. Subjective health[a] | 0.04 | −0.01 | 0.08 | 0.00 | 1.67 | 0.096 | 0.03 | −0.01 | 0.08 | 0.00 | 1.44 | 0.150 |
| 5. In-person interactions[b] | −0.03 | −0.07 | 0.01 | 0.00 | −1.63 | 0.104 | −0.03 | −0.07 | 0.01 | 0.00 | −1.40 | 0.162 |
| 6. Virtual interactions[b] | 0.01 | −0.03 | 0.05 | 0.00 | 0.57 | 0.570 | −0.03 | −0.07 | 0.02 | 0.00 | −1.19 | 0.236 |
| 7. Stressor checklist[b] | 0.07 | 0.03 | 0.12 | 0.02 | 3.45 | 0.001 | 0.09 | 0.04 | 0.13 | 0.02 | 4.02 | 0.000 |
| 8. Health anxiety[b] | 0.35 | 0.30 | 0.39 | 0.24 | 14.95 | 0.000 | 0.35 | 0.30 | 0.40 | 0.23 | 14.69 | 0.000 |
| 9. COVID-19 diagnosis[b] | −0.01 | −0.05 | 0.03 | 0.00 | −0.38 | 0.705 | 0.03 | −0.02 | 0.07 | 0.00 | 1.25 | 0.211 |
| **main effects** | | | | | | | | | | | | |
| 10. Preference for solitude[a] | −0.04 | −0.09 | 0.01 | 0.00 | −1.73 | 0.085 | −0.07 | −0.11 | −0.02 | 0.01 | −2.71 | 0.007 |
| 11. Identified motivation[a] | −0.01 | −0.05 | 0.03 | 0.00 | −0.41 | 0.678 | 0.00 | −0.04 | 0.05 | 0.00 | 0.14 | 0.889 |
| **interaction** | | | | | | | | | | | | |
| 12. Preference × identified[a] | 0.00 | −0.04 | 0.03 | 0.00 | −0.21 | 0.833 | 0.00 | −0.04 | 0.04 | 0.00 | −0.08 | 0.938 |
| **variance explained at each step** | | | | | | | | | | | | |
| Step 1 (#1–9 added) | $R^2 = 0.70$ | | | | | | $R^2 = 0.69$ | | | | | |
| Step 2 (#10, 11 added) | $\Delta R^2 = 0.01$ | | | | | | $\Delta R^2 = 0.01$ | | | | | |
| Step 3 (#12 added) | $\Delta R^2 = 0.00$ | | | | | | $\Delta R^2 = 0.00$ | | | | | |

[a]Measured at Time 1.
[b]Measured at Times 2 and 3, according to the measurement of the outcome variable.

**Table 7.** Results of primary models testing Hypothesis Set 4 predicting Times 2 and 3 ill-being from covariates, preference and external motivations for solitude and preference × external motivation. All results for specific predictors are taken from the final step of the analysis, including all predictors (control and independent variables).

| | Time 2 ill-being | | | | | | Time 3 ill-being | | | | | |
|---|---|---|---|---|---|---|---|---|---|---|---|---|
| | | 95% CI | | partial | | | | 95% CI | | partial | | |
| | β | lower | upper | r² | t | p | β | lower | upper | r² | t | p |
| controls | | | | | | | | | | | | |
| 1. Gender[a] | 0.02 | −0.02 | 0.07 | 0.00 | 1.21 | 0.228 | 0.02 | −0.02 | 0.06 | 0.00 | 1.04 | 0.299 |
| 2. Age[a] | −0.02 | −0.06 | 0.03 | 0.00 | −0.71 | 0.480 | −0.01 | −0.05 | 0.03 | 0.00 | −0.45 | 0.656 |
| 3. Ill-being[a] | 0.61 | 0.57 | 0.66 | 0.46 | 24.91 | 0.000 | 0.59 | 0.55 | 0.64 | 0.44 | 24.16 | 0.000 |
| 4. Subjective health[a] | 0.04 | −0.01 | 0.08 | 0.00 | 1.58 | 0.115 | 0.03 | −0.01 | 0.08 | 0.00 | 1.40 | 0.163 |
| 5. In-person interactions[b] | −0.03 | −0.07 | 0.01 | 0.00 | −1.47 | 0.142 | −0.03 | −0.07 | 0.01 | 0.00 | −1.30 | 0.193 |
| 6. Virtual interactions[b] | 0.01 | −0.03 | 0.05 | 0.00 | 0.44 | 0.660 | −0.03 | −0.07 | 0.02 | 0.00 | −1.17 | 0.240 |
| 7. Stressor checklist[b] | 0.08 | 0.03 | 0.12 | 0.02 | 3.59 | 0.000 | 0.08 | 0.04 | 0.13 | 0.02 | 3.96 | 0.000 |
| 8. Health anxiety[b] | 0.33 | 0.29 | 0.38 | 0.22 | 14.46 | 0.000 | 0.35 | 0.30 | 0.39 | 0.23 | 14.57 | 0.000 |
| 9. COVID-19 diagnosis[b] | 0.00 | −0.05 | 0.04 | 0.00 | −0.21 | 0.831 | 0.03 | −0.02 | 0.07 | 0.00 | 1.22 | 0.224 |
| main effects | | | | | | | | | | | | |
| 10. Preference for solitude[a] | −0.03 | −0.08 | 0.01 | 0.00 | −1.63 | 0.103 | −0.06 | −0.10 | −0.02 | 0.01 | −2.78 | 0.006 |
| 11. External motivation[a] | 0.05 | 0.01 | 0.09 | 0.01 | 2.42 | 0.016 | 0.04 | 0.00 | 0.08 | 0.00 | 1.74 | 0.082 |
| interaction | | | | | | | | | | | | |
| 12. Preference × external[a] | −0.02 | −0.06 | 0.02 | 0.00 | −0.83 | 0.409 | −0.01 | −0.05 | 0.04 | 0.00 | −0.29 | 0.769 |
| **variance explained at each step** | | | | | | | | | | | | |
| Step 1 (#1–9 added) | R² = 0.70 | | | | | | R² = 0.69 | | | | | |
| Step 2 (#10, 11 added) | ΔR² = 0.01 | | | | | | ΔR² = 0.01 | | | | | |
| Step 3 (#12 added) | ΔR² = 0.00 | | | | | | ΔR² = 0.00 | | | | | |

[a]Measured at Time 1.
[b]Measured at Times 2 and 3, according to the measurement of the outcome variable.

coded for participants who scored higher than mid-point (four on seven-point scale), we found that 20% fell under this category at all three time-points. In the discussion section, we elaborated on these percentages in relation to recent data reported by a review by Brooks *et al.* [26] on percentages of people experiencing psychological distress while in quarantine in previous outbreaks (i.e. H1N1, SARS, etc.). Convergent evidence suggested that we did not observe expected inclines in ill-being across these two weeks of self-isolation.

Note that in this sample, when we coded for those who reported spending 100% of their day self-isolating due to COVID-19, we found that 6% fell under this category at Time 1, 21% at Time 2 and 34% at Time 3. Further, when we compared those who reported spending 100% of their day at Times 2 and 3 self-isolating due to COVID-19, with others who spent a lower percentage of time, ill-being scores between those two groups did not meaningfully differ (see electronic supplementary material).

## 4.2. Relations of ill-being in solitude and global depression

Second, following advice in Stage 1 review, we were interested in whether measures of ill-being in solitude were distinguished from depressive symptoms, overall. That is, we wanted to know whether ill-being in solitude can be distinguished from global ill-being, occurring in the same timeframe. Ill-being at Time 2 correlated, $r = 0.84$, with overall depressive symptoms at Time 2, and ill-being at Time 3 correlates $r = 0.86$ with overall depressive symptoms at Time 3. Most notable for us, *depression* in solitude correlated, $r = 0.93$ and $r = 0.94$, with depression in general at Times 2 and 3, respectively. These high correlations meant that we do not have evidence that, during this time, reports of ill-being during solitude were different from overall ill-being; the two were, for all intents and purposes, synonymous.

## 4.3. Links of preference and motivation to ill-being

In Pearson correlation analyses between major study variables (reported in table 4), we observed strong relations between Times 1, 2 and 3 ill-being ranging from $r$s = 0.78–0.89, and in exploratory analyses, we found that not much change occurred in ill-being. We were, therefore, concerned that the vulnerability model that drove our views that preference and motivation would have predictive value for later ill-being, controlling for earlier ill-being, was unwarranted. In other words, the data did not seem to support a view that self-isolating living-alone adults would have the potential for increases in ill-being that could be buffered or exacerbated by our predictors.

We, therefore, conducted two linear regression models similar to the models used to test Hypotheses 1:1, 1:2, 2:1 and 2:2, except that this time we did not control for Time 1 ill-being. By doing this, we simply tested whether preference and motivation measured at Time 1 would be linked with later assessments of ill-being. This analysis should be interpreted with caution, because is reflects, essentially, a series of partial correlations. While one measure precedes the other, this should not be interpreted as a causal effect. For this reason, although vulnerability due to health concerns, demographics and actual quantity of solitude was accounted for, many other possible confounds may play a role in inflating these relations. Findings are summarized in the electronic supplementary material.

A number of meaningfully significant links were in evidence in these models. First, in this model, older age related to *lower* ill-being, $pr = -0.17$, and the relation of stressors slightly increased and met our planned effect size of interest, $pr = 0.20$ at both Times 2 and 3. As before, the relation of health anxiety with ill-being measured at the same time-point was by far the most robust one identified, $pr = 0.51$ at Time 2 and $pr = 0.52$ at Time 3. Accounting for covariates, external motivation linked to higher ill-being at Time 2, $pr = 0.22$, and at Time 3, $pr = 0.17$. In summary, these analyses show non-causal links over the short-term that may not extend across longer periods.

# 5. Discussion

Confirmatory results did not support our hypotheses regarding the role of preference and motivation for solitude on ill-being across time. The links between preference and motivation for solitude, and later ill-being at Times 2 and 3, which controlled for Time 1 ill-being along with a number of vulnerability factors, did not meet the threshold we had set for the minimum effect size of interest. Our minimum effect size of interest was set based on daily experiences of solitude in studies of undergraduate students [12]. Although we had assumed a correlation half the size of the one identified in this

literature ($r = 0.30$ [12]), the effect observed for external motivation was even lower, $pr = 0.09$ at Time 2 and $pr = 0.06$ at Time 3, than our minimum effect of interest, $pr = 0.15$ set at the outset.

Similarly, the effect of preference was $pr = 0.04$ at Time 2 and $pr = 0.09$ at Time 3. Thus, observed increases in ill-being for those who feel pressured to self-isolate, and those who prefer to spend time alone more than with others, are very small. Further, these two variables did not interact to predict change in ill-being over the course of two weeks. The conclusion of this finding is that, even when individuals might prefer time alone or endorsed the importance of self-isolating, for example, for their own or others' benefit, this study did not reveal evidence supporting that it ameliorates or exacerbates their loneliness, depression or anxiety during the time spent alone. Even with additional exploratory analyses to test relations of both preference and motivation to ill-being at each of the two follow-up time-points, we did not observe an effect above our minimum effect size of interest, except in the case of external motivation predicting higher ill-being one week later.

We see three possibilities for why we observed lower effect size than those previously observed, which merit further study. First, it could be argued that any effects of identified and external motivation for solitude do not apply to *extreme* solitude. However, even in studies of forced solitude in the extreme, such as solitary confinement in prisons where choicelessness and external motivation are extremely high, motivation is linked to negative mental health consequences for prisoners [27,28]. It is unlikely that our study captured the extreme cases of isolating individuals, and that for this reason preference and motivation played a less important role.

On a similar note, the sample obtained in this research was not comparable to other cases of isolation due to quarantine. Obligatory quarantine of those who have been exposed to a contagious illness causes great distress to quarantined individuals, but quarantined cases and those who are in voluntary self-isolation like the cases we are studying in our sample are in very different circumstances [26]. The levels of distress experienced among those in quarantine are likely to be higher than those in voluntary self-isolation. For example, in a study looking at experiences of horse owners during quarantine due to an equine influenza outbreak in Australia, 34% of those in quarantine reported distress; this is notably higher than the percentage of those experiencing distress in the Australian general population [29]. In another sample of parents impacted by H1N1 and SARS outbreaks, 28% of parents in quarantine reported trauma-related disorder, compared with 6% of parents not in quarantine [30]. In our sample, as reported in the exploratory tables, 13% of the participants scored higher than the mid-point on ill-being (higher than four on seven-point range) at Time 1, 12% at Time 2 and 14% at Time 3.

Further reflecting on the circumstances in which our participants found themselves, although our quality checks regarding the percentage of time spent alone and in self-isolation showed significant increases on these indicators, we also identified that the amount of time spent doing daily activities at home, alone, did not increase. While this finding was unexpected, it is likely this is due to the fact that our living-alone adults already undertook a large number of tasks on their own, so that their daily at-home activities did not change much during the COVID-19 crisis. Thus, perhaps despite more time spent alone and in self-isolation, our participants may have experienced a gentler transition into self-isolation wherein some daily habits were uninterrupted, and therefore costs to wellness were lower.

Second, it is possible that there was little change for preference and motivation for solitude to predict. While previous research suggested that prolonged self-isolation would yield psychological consequences; in the current study, we did not see evidence of change in ill-being over the course of two weeks of self-isolation. It is worth noting that the two-week span of the study represented a fairly short duration, and that ill-being changes may be observable as individuals spend additional weeks and months in self-isolation. Given this, it is worthwhile to engage ongoing research efforts tracking ill-being to identify a 'breaking point' in which increases in ill-being may be observed. Though we cannot speak to this possibility at present, it may be that nonlinear trends of increasing ill-being will be evidenced across longer spans of time.

In our sample, which primarily varied in ages from 35 to 87 years, participants across the adulthood spectrum who reported engaging in less in-person interaction and spending more waking hours alone felt similarly during this time of self-isolation. These findings speak to concerns regarding older adults during self-isolation. Certainly, older adults, who are more vulnerable to COVID-19 exposure, more highly restricted to the time at home, and generally less comfortable with meeting their own needs online, benefit from the help and support of the community. However, at least within our sample, we did not find evidence that current circumstances yield depression, loneliness or anxiety, despite media warnings about negative consequences of the rapid move to self-isolating during the period of the study (CNBC.com, 2020; TheGuardian.com, 2020; Health.com, 2020; NewYorker.com, 2020). Thus,

concerns that self-isolation is harmful to the mental health of adults were unsubstantiated in our findings. In fact, few of our predictors could affect change in ill-being at all, which remained low through the duration of the study.

Third, this study was conducted amidst precipitous changes in governmental regulations and guidelines, as well as individuals' understanding of and plans for self-isolation, and although preference is seen as a dispositional quality that should not much change [6], motivations may have shifted within the first week of the study. Since we did not measure motivation past Time 1, we cannot answer that question. However, we attempted to acknowledge the quickly changing landscape of COVID-19 by, part-way through the study, moving to weekly data collections following Stage 1 reviewer advice. Two days before Time 2 data collection, the UK shut down non-essential public places and residents were asked to stay at home nearly all the time (The Guardian, https://www.theguardian.com/world/2020/mar/20/london-pubs-cinemas-and-gyms-may-close-in-covid-19-clampdown); 1 day after the USA saw state-wide school closures by more than half of states (CNN, https://edition.cnn.com/2020/03/15/health/us-coronavirus-sunday-updates/index.html). Since Time 2 referred primarily to the week before these changes took place, it reflected initial motivations more closely. Time 3 had seen additional extended self-isolation and self-isolation driven by more pressures, such as highly limited access to groceries for those who were home (e.g. The Guardian, https://www.theguardian.com/commentisfree/2020/mar/27/the-guardian-view-on-empty-supermarket-shelves-panic-is-not-the-problem). Certainly, it is reasonable to assume that the sense of urgency and pressure around self-isolation changed following these decisions, and similarly that perceived importance may have changed. If it is the case, it speaks to the malleability of motivations as a function of communications and circumstances surrounding COVID-19. Thus, a fruitful topic of future research would be to experimentally and longitudinally test motivations to examine the effects of these communications on ill-being.

Interestingly, the effect size for identified motivation changing ill-being was close to zero. We did not find evidence that when people recognized the importance of self-isolating, this protected them from ill-being during the time spent alone. Future research may consider other outcomes of identified motivation, for example, on behavioural intention [31]. Providing people recognize the importance of self-isolation, they may be more likely to cooperate and engage in this behaviour [8]. Yet, our findings suggest that when they do so, they may experience the same mental health costs or benefits within solitude of this cooperation.

Importantly, even with the little increased ill-being, the one predictor that robustly predicted its change was a concern about health, which was measured simultaneously with ill-being-dependent variables. Higher health concerns measured at the same time as the dependent variable related to more loneliness, depression and anxiety. Thus, health anxiety was by far the most compelling vulnerability for individuals in self-isolation, a finding which suggests that practical and emotional support for health needs is particularly important during self-isolation.

## 6. Limitations

These findings should be viewed in light of two limitations of this study design. First, as described previously, the study took place during a time of fast-paced changes to circumstances, perceived health concerns and expectations for the duration of lifestyle restrictions, all of which changed between each time-point collected. On one hand, these changes contribute to the strength and richness of the study, since longitudinal analyses stretched the capacity of our predictors to test their boundary conditions in these unusual circumstances. Further, because our predictor measures were collected at the start of the study (Time 1) only, we cannot know how preference and motivation for solitude might have changed during this time.

A second major concern is that it is unclear whether we tested ill-being in solitude or ill-being in general. In fact, depression in solitude correlated ($r = 0.93$ (Time 2) and 0.95 (Time 3)) with a general measure of depression we had collected following Stage 1 reviewer recommendations to conduct this test. Understandably, when individuals spend a majority of their waking hours in solitude these two should be nearly identical. However, we cannot be certain whether solitude that defines daily experiences, as it does now when individuals are in self-isolation, behaves the same as solitude that is incidental in daily life under normal circumstances. In future research or in meta-analytic work later, it would be insightful to compare the two conditions to understand where they differ and where they share characteristics.

# 7. Conclusion

The experience of living-alone individuals self-isolating is not atypical. They represent anywhere from 23.9% to 50% of households in the UK and the USA [32,33], the two populations from which we sampled. In this study, we did not find evidence that the mental health of these individuals had been negatively affected by self-isolation in its early weeks, a finding that clearly differentiates this group from those who find themselves in forced quarantine [26]. Further, we did not find evidence in support of our planned hypotheses above the minimum effect size of interest. Other studies have similarly failed to evidence change in wellness as a function of motivation in solitude, though they evidenced direct links with concurrent wellness [12]. However, we had anticipated that protective or aggravating predictive factors would play a stronger role predicting change because of the unexpected and sudden increases in the time spent alone in response to the COVID-19 outbreak.

One might conclude that the absence of evidence for increased ill-being reflects resilience in the face of self-isolation. However, such an interpretation is unwarranted without additional data that speak directly to the mechanisms underlying resilience. Specific groups might react differently to self-isolation with different personal or social resources, but testing this requires direct comparisons with a lower resilience group who report increased ill-being under similar conditions. We would caution researchers to be sensitive to possible nuances in reactions to self-isolation, and to avoid making assumptions and *post hoc* interpretations based on studies of forced quarantine or loosely applied psychological models. Doing so may raise false hopes and misdirect limited resources.

Ethics. This research was approved by the Durham University's Ethics Committee (PSYCH-2020-03-11T23_41_08).

Data accessibility. Data are available on OSF (OSF.IO/MN8KX), where readers can find our pre-registrations, raw data and code. Analyses were primarily conducted in R (except for reliabilities), and we offer the R code, as well as SPSS files and syntax for composites and reliabilities.

Authors' contributions. N.W. and T.-V.N. worked in a fully collaborative fashion on this project. Together, they conceptualized the study, developed methods, ethical documentation, and analytic plan, prepared registrations and study materials, and wrote Stage 1 of the registered report. At Stage 2, they once again collaborated to analyse data and write the full report.

Competing interests. We declare we have no competing interests.

Funding. This work was supported in part by a grant from the European Research Council (ERC; SOAR; 851890).

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
