## [Reviewer comments · Royal Society Open Science]

Review History

RSOS-200458.R0 (Original submission)

Review form: Reviewer 1 (Catherine Hobbs)

Do you have any ethical concerns with this paper?

No

Recommendation?

Accept with minor revision

Comments to the Author(s)

This is an interesting piece of research making use of the current climate to evaluate an important and scientifically valid research question. The hypotheses were well considered and the methodology is plausible to allow these hypotheses to be answered. The quality checks were thought through in light of the naturalistic design. Overall, the methodology of the study was sufficiently detailed to allow further replication, and prevent undisclosed flexibility. Whilst the first phase of data has already been collected this seems justified given the required rapid commencement of the study. A few minor comments are detailed below:

- Some clarification as to why a 2 week follow-up period was selected may be helpful. Given that this is a quickly changing situation I wonder if motivations are likely to fluctuate throughout a 2-week period. The authors may consider a greater number of shorter-term follow-ups or expanding upon their justification for why 2 weeks was selected.
- The authors have chosen to limit their sample to individuals living alone. Whilst this limits the influence of interaction with close others on ill-being it does represent the more extreme cases in the community. It would be worth highlighting and justifying this.
- When measuring 'Motivation for solitude' the scale asks participants to think about times when they are not interacting with anyone in person or virtually. Given that virtual interaction does not have any health implications this would seem to be measuring general motivation for solitude, rather than the motivation in this particular health environment. It may be worth separating in person and virtual interactions in the 2nd phase of data collection to distinguish between these.
- Have any checks for data quality been included in the Qualtrics questionnaires? It may be worth considering adding attention check questions in the 2nd phase of the study to assess data quality.
- Further justification of what basis the smallest effects of interest were chosen on may be useful.
- The authors may consider incorporating a brief global measure of change in wellbeing in the 2nd follow-up to compare to their composite measures

Review form: Reviewer 2 (Paul Thompson)

Do you have any ethical concerns with this paper?

No

Recommendation?

Major revision

Comments to the Author(s)

I enjoyed reading the authors registered report. They have interesting and well posed research questions which would, with some further modifications, provide a useful addition to the literature. Under the current study setup, the statistical analyses are OK, but given my comments in my report may need some adjustment. The authors have provided sufficient detail in the report and open access materials to allow this study to be replicated in my opinion. The hypotheses are generally appropriate but given my comments in the report (Appendix A), might need minor adjustment.

Review form: Reviewer 3 (Jim Fryer)

Do you have any ethical concerns with this paper?

No

Recommendation?

Accept in principle

Comments to the Author(s)

Manuscript Number: RSOS-200458

Title: Motivation and preference in isolation: A test of their different influences on responses to self-isolation during the COVID-19 outbreak

The current proposal is intended to examine the effects of preference for isolation and quality of motivation for self-isolation. In general, higher levels of ill-being (e.g., depressive symptoms,

anxiety, physical health issues, etc.) are expected to be associated with lower preference for solitude, lower identified regulation, and higher external regulation (as well as the appropriate interactive effects).

This work has a strong theoretical basis, the research questions are thoroughly explained, and the analytic approach is clearly designed to properly test the hypotheses. This will be a meaningful and timely contribution to the literature, and I'm very eager to see this work carried through to completion.

My only (relatively minor) concern is that there may be confusion between physical isolation and social isolation, both as the participants respond to the items, and in the eventual interpretation of these findings. As the authors know (very well), according to Self-Determination Theory, a lack of satisfaction of basic psychological needs is associated with an increase in ill-being. These research questions focus on the impact of autonomy support for self-isolation, but the measurement may also inadvertently pick up lack of satisfaction for relatedness.

a. According to the questionnaire documents on the OSF, the frequency of social interaction questions are different across the two time points (not including the time frame):

T1: "How frequently, in general, do you perform your daily activities at home by yourself?"

T2: "How frequently, in the past 2 weeks, have you performed your daily activities while at home alone, not interacting with anyone in person or virtually?"

The T1 item focuses on physical isolation, while the T2 item also includes social isolation. This measure is only intended to "serve as a quality indicator testing whether participants were in solitude across two weeks" (p. 10). However, if the authors observe increases in ill-being, this may be a potential alternate explanation. It is certainly possible that a person (especially one high in preference for solitude) might be able to be physically alone, even in self-isolation, but still maintain connections to others.

b. In addition to frequency, participants will also respond to items asking the ways in which they interacted with others. In-person interactions are expected to decrease (which is a good check, as mentioned by the authors), and the "face-to-face item will be included as a covariate in all main models" (p. 10). I would suggest an additional set of analyses controlling for all modes of contact (face-to-face, social media, phone/text).

Decision letter (RSOS-200458.R0)

20-Mar-2020

Dear Dr Weinstein,

The Editors assigned to your Stage 1 Registered Report ("Motivation and Preference in Isolation: A test of their different influences on responses to self-isolation during the COVID-19 outbreak") have now received comments from four reviewers. We would like you to revise your paper in accordance with the referee and editors suggestions which can be found below (not including confidential reports to the Editor).

When submitting your revised manuscript, you must respond to the comments made by the referees and upload a file "Response to Referees" in "Section 2 - File Upload". Please use this to document how you have responded to the comments, and the adjustments you have made. In order to expedite the processing of the revised manuscript, please be as specific as possible in your response.

Kind regards,
 Professor Chris Chambers
 Royal Society Open Science
 openscience@royalsociety.org

Associate Editor Comments to Author (Professor Chris Chambers):

Associate Editor: 1

Comments to the Author:

Four expert reviewers have now assessed the Stage 1 manuscript include three field experts (Rev 1, 3, 4) and one statistical expert (Rev 2). Reviewer 4 is also an associate editor at the journal. All of the reviews are positive overall but indicate areas requiring revision prior to IPA. To briefly summarise: further methodological detail and justification of design decisions is required (such as inclusion criteria, the basis of the 2 week follow-up period, and the smallest effect size of interest; see Reviewer 1). Note that Reviewer 4 (point 5) offers a potential solution to this latter issue. Reviewer 2 notes several key areas in the analysis plan requiring attention, with particular concerns about confounding variables, and the most major concern being the ability of the design to single out the effects of self-isolation. Reviewer 3 raises a related concern about distinguishing the effects of physical isolation from the effects of social isolation.

All concerns appear to be readily addressable through a swift and thorough revision, therefore a Major Revision is invited. My sincere thanks to all four reviewers for providing such insightful and constructive reviews on such an extraordinary timescale.

Comments to Author:

Reviewer: 1

Comments to the Author(s)

This is an interesting piece of research making use of the current climate to evaluate an important and scientifically valid research question. The hypotheses were well considered and the methodology is plausible to allow these hypotheses to be answered. The quality checks were thought through in light of the naturalistic design. Overall, the methodology of the study was sufficiently detailed to allow further replication, and prevent undisclosed flexibility. Whilst the first phase of data has already been collected this seems justified given the required rapid commencement of the study. A few minor comments are detailed below:

- Some clarification as to why a 2 week follow-up period was selected may be helpful. Given that this is a quickly changing situation I wonder if motivations are likely to fluctuate throughout a 2-week period. The authors may consider a greater number of shorter-term follow-ups or expanding upon their justification for why 2 weeks was selected.

- The authors have chosen to limit their sample to individuals living alone. Whilst this limits the influence of interaction with close others on ill-being it does represent the more extreme cases in the community. It would be worth highlighting and justifying this.

- When measuring 'Motivation for solitude' the scale asks participants to think about times when they are not interacting with anyone in person or virtually. Given that virtual interaction does not have any health implications this would seem to be measuring general motivation for solitude,

rather than the motivation in this particular health environment. It may be worth separating in person and virtual interactions in the 2nd phase of data collection to distinguish between these.

- Have any checks for data quality been included in the Qualtrics questionnaires? It may be worth considering adding attention check questions in the 2nd phase of the study to assess data quality.
- Further justification of what basis the smallest effects of interest were chosen on may be useful.
- The authors may consider incorporating a brief global measure of change in wellbeing in the 2nd follow-up to compare to their composite measures

Reviewer: 2

Comments to the Author(s)

I enjoyed reading the authors registered report. They have interesting and well posed research questions which would, with some further modifications, provide a useful addition to the literature. Under the current study setup, the statistical analyses are OK, but given my comments in my report may need some adjustment. The authors have provided sufficient detail in the report and open access materials to allow this study to be replicated in my opinion. The hypotheses are generally appropriate but given my comments in the report, might need minor adjustment.

[See detailed comments in attached report]

Reviewer: 3

Comments to the Author(s)

Manuscript Number: RSOS-200458

Title: Motivation and preference in isolation: A test of their different influences on responses to self-isolation during the COVID-19 outbreak

The current proposal is intended to examine the effects of preference for isolation and quality of motivation for self-isolation. In general, higher levels of ill-being (e.g., depressive symptoms, anxiety, physical health issues, etc.) are expected to be associated with lower preference for solitude, lower identified regulation, and higher external regulation (as well as the appropriate interactive effects).

This work has a strong theoretical basis, the research questions are thoroughly explained, and the analytic approach is clearly designed to properly test the hypotheses. This will be a meaningful and timely contribution to the literature, and I'm very eager to see this work carried through to completion.

My only (relatively minor) concern is that there may be confusion between physical isolation and social isolation, both as the participants respond to the items, and in the eventual interpretation of these findings. As the authors know (very well), according to Self-Determination Theory, a lack of satisfaction of basic psychological needs is associated with an increase in ill-being. These research questions focus on the impact of autonomy support for self-isolation, but the measurement may also inadvertently pick up lack of satisfaction for relatedness.

a. According to the questionnaire documents on the OSF, the frequency of social interaction questions are different across the two time points (not including the time frame):

T1: "How frequently, in general, do you perform your daily activities at home by yourself?"

T2: "How frequently, in the past 2 weeks, have you performed your daily activities while at home alone, not interacting with anyone in person or virtually?"

The T1 item focuses on physical isolation, while the T2 item also includes social isolation. This measure is only intended to "serve as a quality indicator testing whether participants were in

solitude across two weeks” (p. 10). However, if the authors observe increases in ill-being, this may be a potential alternate explanation. It is certainly possible that a person (especially one high in preference for solitude) might be able to be physically alone, even in self-isolation, but still maintain connections to others.

b. In addition to frequency, participants will also respond to items asking the ways in which they interacted with others. In-person interactions are expected to decrease (which is a good check, as mentioned by the authors), and the “face-to-face item will be included as a covariate in all main models” (p. 10). I would suggest an additional set of analyses controlling for all modes of contact (face-to-face, social media, phone/text).

Reviewer: 4 (Associate Editor)

Comments to the Author(s)

The study will address an important question related to motivation of self-isolation and its relationship to ill-being. The registered report is well documented in its methods, hypotheses and statistical analyses. I have only few comments:

- 1) It would be better to explain why >35 years old has been set as inclusion criteria. Related to this, not sure it makes sense to pull together the data gathered via Social media, given they have different inclusion criteria.
- 2) “In the past 2 weeks, were you specifically following ____ recommendations to self-isolate (or to stay home and away from other people and places)” I would give an “other” option, that is open end
- 3) Control Measures Time 1 (baseline). For all these measures, it is not clear why you ask about the last 6 months. I would imagine you will have a recency effect (at least in some) given the outbreak of the Corona Virus. I would rather ask in the past weeks, and see how this change from T1 to T2
- 4) Typo on line 47. Should be march 14
- 5) check this preprint <https://psyarxiv.com/epcyb/> for possible sample size calculation

Author's Response to Decision Letter for (RSOS-200458.R0)

See Appendix B.

RSOS-200458.R1 (Revision)

Review form: Reviewer 1 (Catherine Hobbs)

Do you have any ethical concerns with this paper?

No

Recommendation?

Accept with minor revision

Comments to the Author(s)

Thank you to the authors for responding to my comments and for providing additional details regarding their rationale for certain decisions. My previous concerns have been taken into account and addressed well by the authors. Regarding the inclusion of attention checks, participants should be excluded who either fail to select the correct answer or miss an item. This

section currently reads as though the authors only intend to exclude those who meet both criteria.

Review form: Reviewer 2 (Paul Thompson)

Do you have any ethical concerns with this paper?

No

Recommendation?

Accept in principle

Comments to the Author(s)

I am happy with the response of the authors. The authors have addressed my major concerns with the addition of relevant covariates and a number of minor points. The time scale for the study has been justified and an additional time point adds value to the proposed analyses. The clarification and rationale of details in the paper requested by other reviewers has improved the paper greatly.

Some minor suggestions have been avoided by stating "outside the scope of the current study" which I am happy to let go as I don't have the field specific knowledge to argue its inclusion further and I suspect will have minimal impact.

I am happy to accept in principle as it stands provided that the other reviewers concur on the field specific elements.

Decision letter (RSOS-200458.R1)

25-Mar-2020

Dear Dr Weinstein

On behalf of the Editor, I am pleased to inform you that your Manuscript RSOS-200458.R1 entitled "Motivation and Preference in Isolation: A test of their different influences on responses to self-isolation during the COVID-19 outbreak" has been accepted in principle for publication in Royal Society Open Science. The reviewers' and editors' comments are included at the end of this email.

****PLEASE NOTE:** There is one final minor revision to make before registering the protocol. Please see the Associate Editor's comment below. ******

You may now progress to Stage 2 and complete the study as approved. Before commencing data collection we ask that you:

- 1) Update the journal office as to the anticipated completion date of your study.
- 2) Register your approved protocol on the Open Science Framework (<https://osf.io/>) or other recognised repository, either publicly or privately under embargo until submission of the Stage 2 manuscript. Please note that a time-stamped, independent registration of the protocol is mandatory under journal policy, and manuscripts that do not conform to this requirement cannot be considered at Stage 2. The protocol should be registered unchanged from its current approved state, with the time-stamp preceding implementation of the approved study design.

Following completion of your study, we invite you to resubmit your paper for peer review as a Stage 2 Registered Report. Please note that your manuscript can still be rejected for publication at Stage 2 if the Editors consider any of the following conditions to be met:

- The results were unable to test the authors' proposed hypotheses by failing to meet the approved outcome-neutral criteria.
- The authors altered the Introduction, rationale, or hypotheses, as approved in the Stage 1 submission.
- The authors failed to adhere closely to the registered experimental procedures. Please note that any deviations from the approved experimental procedures must be communicated to the editor immediately for approval, and prior to the completion of data collection. Failure to do so can result in revocation of in-principle acceptance and rejection at Stage 2 (see complete guidelines for further information).
- Any post-hoc (unregistered) analyses were either unjustified, insufficiently caveated, or overly dominant in shaping the authors' conclusions.
- The authors' conclusions were not justified given the data obtained.

We encourage you to read the complete guidelines for authors concerning Stage 2 submissions at <https://royalsocietypublishing.org/rsos/registered-reports#ReviewerGuideRegRep>. Please especially note the requirements for data sharing, reporting the URL of the independently registered protocol, and that withdrawing your manuscript will result in publication of a Withdrawn Registration.

Once again, thank you for submitting your manuscript to Royal Society Open Science and we look forward to receiving your Stage 2 submission. If you have any questions at all, please do not hesitate to get in touch. We look forward to hearing from you shortly with the anticipated submission date for your stage two manuscript.

on behalf of Professor Chris Chambers (Associate Editor) and Chris Chambers (Registered Reports Editor, Royal Society Open Science)
openscience@royalsociety.org

Associate Editor Comments to Author (Professor Chris Chambers):

Associate Editor: 1

Comments to the Author:

Three of the original reviewers assessed the revised manuscript. All are satisfied and recommend in-principle acceptance. Reviewer 1 makes the following recommendation: "Regarding the inclusion of attention checks, participants should be excluded who either fail to select the correct answer or miss an item. This section currently reads as though the authors only intend to exclude those who meet both criteria." This seems sensible, and if the authors agree, please make this minor change *before* formally registering the protocol as instructed (point 2 above). To minimise unnecessary bureaucracy you do not need to resubmit the Stage 1 manuscript with this revision, but please ensure that it is enacted in the registered protocol. When updating the journal office as to the completion date of the study (point 1 above), please attach to that email a copy of the registered protocol with this change made, or if the registered protocol is public then simply include a link to the protocol.

Reviewer: 1

Comments to the Author(s)

Thank you to the authors for responding to my comments and for providing additional details regarding their rationale for certain decisions. My previous concerns have been taken into account and addressed well by the authors. Regarding the inclusion of attention checks, participants should be excluded who either fail to select the correct answer or miss an item. This section currently reads as though the authors only intend to exclude those who meet both criteria.

Reviewers' comments to Author:

Reviewer: 2

Comments to the Author(s)

I am happy with the response of the authors. The authors have addressed my major concerns with the addition of relevant covariates and a number of minor points. The time scale for the study has been justified and an additional time point adds value to the proposed analyses. The clarification and rationale of details in the paper requested by other reviewers has improved the paper greatly.

Some minor suggestions have been avoided by stating "outside the scope of the current study" which I am happy to let go as I don't have the field specific knowledge to argue its inclusion further and I suspect will have minimal impact.

I am happy to accept in principle as it stands provided that the other reviewers concur on the field specific elements.

Reviewer: 4 (Associate Editor)

The authors addressed all my comments.

Author's Response to Decision Letter for (RSOS-200458.R1)

See Appendix C.

RSOS-200458.R2 (Revision)

Review form: Reviewer 1 (Catherine Hobbs)

Is the manuscript scientifically sound in its present form?

Yes

Are the interpretations and conclusions justified by the results?

Yes

Is the language acceptable?

Yes

Do you have any ethical concerns with this paper?

No

Have you any concerns about statistical analyses in this paper?

No

Recommendation?

Accept with minor revision

Comments to the Author(s)

The introduction, rationale and stated hypotheses are the same as the approved stage 1 submission, and the authors adhered to the registered experimental procedures. The unregistered exploratory statistical analyses seem justified. The authors' conclusions are well justified in light of their data. The data collected by the authors adequately allowed them to test their proposed hypotheses though confirmatory quality checks. However, my one concern is the failure of the quality check for daily activities performed at home which was in the opposite direction to what was expected. I feel that this warrants further commentary.

Review form: Reviewer 2 (Paul Thompson)

Is the manuscript scientifically sound in its present form?

Yes

Are the interpretations and conclusions justified by the results?

Yes

Is the language acceptable?

Yes

Do you have any ethical concerns with this paper?

No

Have you any concerns about statistical analyses in this paper?

No

Recommendation?

Accept as is

Comments to the Author(s)

I think the authors have done a good job in following their stage 1 plan (post-reviewer comments). The results have been reported accurately and I'm pleased to say that the authors have not tried to explain away null findings with an overly positive spin. I think the exploratory analyses are OK, and have equally been reported fairly. The discussion and conclusions seem justified and their explanations of pattern of results seems plausible but I would default to the other reviewers views here given that I can only really comment on the statistical elements.

Many thanks, Paul Thompson

Review form: Reviewer 3 (Jim Fryer)

Is the manuscript scientifically sound in its present form?

Yes

Are the interpretations and conclusions justified by the results?

Yes

Is the language acceptable?

Yes

Do you have any ethical concerns with this paper?

No

Have you any concerns about statistical analyses in this paper?

No

Recommendation?

Accept with minor revision

Comments to the Author(s)

Manuscript Number: RSOS-200458.R2

Title: Motivation and preference in isolation: A test of their different influences on responses to self-isolation during the COVID-19 outbreak

I reviewed the initial submission of this manuscript (as Reviewer 3), and I appreciate the willingness of the authors to consider our suggestions. The authors' central hypotheses were not confirmed; however, the rationale and hypotheses are identical to what was approved in the Stage 1 submission, the additional exploratory analyses are sound and informative, the authors adhered to their pre-registered procedures, and the conclusions are justified given the data. This work represents solid science, and I will be very happy to see it published.

Three minor comments:

1. It doesn't seem as though my previous comment about the measurement of social interaction was addressed. As mentioned previously, the questions about time spent alone asked at Times 2 and 3 add "not interacting with anyone in person or virtually." This change was not mentioned in the description of the Background Measures (in the section titled "Frequency of time spent alone (Times 1-3; quality check)"). Analyses with these items (from the section on "Confirmatory Quality Checks," in the "Time spent alone" subheading) found no significant differences in the amount of time spent performing daily activities alone between Time 1 and Time 2 or 3 - this pattern of results was not expected, and could be potentially due to this differences.
2. In Tables 3 and 4, it would be helpful to include the possible and observed ranges for the variables, especially since there are several different scales used. I'm sure the reader would appreciate the context for the means and standard deviations, without having to constantly refer back to the Methods section. Also, the numbering of the rows for Table 3 appears to start at 2.
3. One other potential explanation for the primary nonsignificant results is that a significant change in ill-being would be observed with a longer time frame. Perhaps two weeks just isn't long enough to see a notable increase in anxiety and depression, due to self-isolation. I would be very interested to see the results of another wave of data collection, several weeks after Time 3, if that was feasible for the authors. (To be clear, this is only intended to be another possible interpretation of the findings, not a request for yet another data collection prior to the publication of this manuscript.)

Decision letter (RSOS-200458.R2)

Dear Dr Weinstein:

On behalf of the Editor, I am pleased to inform you that your Stage 2 Registered Report RSOS-200458.R2 entitled "Motivation and Preference in Isolation: A test of their different influences on responses to self-isolation during the COVID-19 outbreak" has been deemed suitable for publication in Royal Society Open Science subject to minor revision in accordance with the referee suggestions. Please find the referees' comments at the end of this email.

The reviewers and Subject Editor have recommended publication, but also suggest some minor revisions to your manuscript. Therefore, I invite you to respond to the comments and revise your manuscript.

Please also ensure that all the below editorial sections are included where appropriate -- if any section is not applicable to your manuscript, please can we ask you to nevertheless include the heading, but explicitly state that the heading is inapplicable. An example of these sections is attached with this email.

- Ethics statement

- Data accessibility

[http://datadryad.org/submit?journalID=RSOS&manu=\(Document not available\)](http://datadryad.org/submit?journalID=RSOS&manu=(Document not available))

- Competing interests

- Authors' contributions

- Acknowledgements

- Funding statement

Because the schedule for publication is very tight, it is a condition of publication that you submit the revised version of your manuscript within 7 days (i.e. by the 28-Apr-2020). If you do not think you will be able to meet this date please let me know immediately.

Supplementary files will be published alongside the paper on the journal website and posted on the online figshare repository (<https://figshare.com>). The heading and legend provided for each supplementary file during the submission process will be used to create the figshare page, so please ensure these are accurate and informative so that your files can be found in searches. Files

on figshare will be made available approximately one week before the accompanying article so that the supplementary material can be attributed a unique DOI.

Please note that Royal Society Open Science will introduce article processing charges for all new submissions received from 1 January 2018. Registered Reports submitted and accepted after this date will ONLY be subject to a charge if they subsequently progress to and are accepted as Stage 2 Registered Reports. If your manuscript is submitted and accepted for publication after 1 January 2018 (i.e. as a full Stage 2 Registered Report), you will be asked to pay the article processing charge, unless you request a waiver and this is approved by Royal Society Publishing. You can find out more about the charges at <https://royalsocietypublishing.org/rsos/charges>. Should you have any queries, please contact openscience@royalsociety.org.

on behalf of Professor Chris Chambers
(Registered Reports Editor, Royal Society Open Science)
openscience@royalsociety.org

Associate Editor Comments to Author (Professor Chris Chambers):

Associate Editor: 1

Comments to the Author:

The Stage 2 manuscript was returned to all three Stage 1 reviewers plus the specialist editor (listed here as Reviewer 4). All assessments are positive but nevertheless recommend some minor revisions to clarify adherence to protocol, consider additional data presentations, and address various limitations or explanations of the results in the Discussion. Provided the authors respond thoroughly to each point raised, full acceptance is unlikely to require further in-depth review.

Reviewer comments to Author:

Reviewer: 1

Comments to the Author(s)

The introduction, rationale and stated hypotheses are the same as the approved stage 1 submission, and the authors adhered to the registered experimental procedures. The unregistered exploratory statistical analyses seem justified. The authors' conclusions are well justified in light of their data. The data collected by the authors adequately allowed them to test their proposed hypotheses though confirmatory quality checks. However, my one concern is the failure of the quality check for daily activities performed at home which was in the opposite direction to what was expected. I feel that this warrants further commentary.

Reviewer: 2 (statistical & design review)

Comments to the Author(s)

I think the authors have done a good job in following their stage 1 plan (post-reviewer comments). The results have been reported accurately and I'm pleased to say that the authors have not tried to explain away null findings with an overly positive spin. I think the exploratory analyses are OK, and have equally been reported fairly. The discussion and conclusions seem

justified and their explanations of pattern of results seems plausible but I would default to the other reviewers views here given that I can only really comment on the statistical elements.

Many thanks, Paul Thompson

Reviewer: 3

Comments to the Author(s)

Manuscript Number: RSOS-200458.R2

Title: Motivation and preference in isolation: A test of their different influences on responses to self-isolation during the COVID-19 outbreak

I reviewed the initial submission of this manuscript (as Reviewer 3), and I appreciate the willingness of the authors to consider our suggestions. The authors' central hypotheses were not confirmed; however, the rationale and hypotheses are identical to what was approved in the Stage 1 submission, the additional exploratory analyses are sound and informative, the authors adhered to their pre-registered procedures, and the conclusions are justified given the data. This work represents solid science, and I will be very happy to see it published.

Three minor comments:

1. It doesn't seem as though my previous comment about the measurement of social interaction was addressed. As mentioned previously, the questions about time spent alone asked at Times 2 and 3 add "not interacting with anyone in person or virtually." This change was not mentioned in the description of the Background Measures (in the section titled "Frequency of time spent alone (Times 1-3; quality check)"). Analyses with these items (from the section on "Confirmatory Quality Checks," in the "Time spent alone" subheading) found no significant differences in the amount of time spent performing daily activities alone between Time 1 and Time 2 or 3 - this pattern of results was not expected, and could be potentially due to this differences.
2. In Tables 3 and 4, it would be helpful to include the possible and observed ranges for the variables, especially since there are several different scales used. I'm sure the reader would appreciate the context for the means and standard deviations, without having to constantly refer back to the Methods section. Also, the numbering of the rows for Table 3 appears to start at 2.
3. One other potential explanation for the primary nonsignificant results is that a significant change in ill-being would be observed with a longer time frame. Perhaps two weeks just isn't long enough to see a notable increase in anxiety and depression, due to self-isolation. I would be very interested to see the results of another wave of data collection, several weeks after Time 3, if that was feasible for the authors. (To be clear, this is only intended to be another possible interpretation of the findings, not a request for yet another data collection prior to the publication of this manuscript.)

Reviewer: 4 (specialist editor)

The study address an important question related to motivation of self-isolation and its relationship to ill-being. The report is well documented in its methods, hypotheses and statistical analyses. I have few comments mainly related to the sample, and exclusion criteria:

- 1) In the RR it is indicated that only participants >35 years are included, but in the sample description it is stated that the age ranges from 19 to 87 years old. As the authors mention in a footnote, the younger sample is belonging to a second pool of the same survey taken at t1, which has been then removed from the final analyses. I would suggest to state more clearly the age range of the final sample analyzed, and the socio-demographical data (gender distribution, employment status, etc) related to this

- 2) I would also recommend to give indications about the final sample analyzed after outliers removal (how many were in total, plus socio-dem characteristics)
- 3) It would be interesting to know the distribution of ages across decades (how many 35-45, 46-55 etc)
- 4) why being widow is considered an exclusion criteria? I cannot find where it was stated
- 5) related to point 1-2, I am not completely sure the analyses have been conducted on the final, selected sample.

Author's Response to Decision Letter for (RSOS-200458.R2)

See Appendix D.

Decision letter (RSOS-200458.R3)

Dear Dr Weinstein:

It is a pleasure to accept your manuscript entitled "Motivation and Preference in Isolation: A test of their different influences on responses to self-isolation during the COVID-19 outbreak" in its current form for publication in Royal Society Open Science. Congratulations on being the first Stage 2 Registered Report accepted as part of the journal's COVID-19 fasttrack initiative.

on behalf of Professor Chris Chambers (Subject Editor)
openscience@royalsociety.org

Appendix A

Review: Motivation and Preference in Isolation: A test of their different influences on responses to self-isolation during the COVID-19 outbreak by Weinstein, N. and Nguyen, T.

Reviewed by: Paul Thompson, Department of Experimental Psychology, University of Oxford (statistical aspects only; I have no prior expert knowledge of ill-being or the effects of self-isolation)

Overview:

The registered report proposes a longitudinal observational study to examine whether preference and motivation for time alone effects individual's ill-being during a period of self-isolation. The evolving pandemic of COVID-19 provides a rare setting to test hypotheses relating to self-isolation with a large and representative population sample. The authors seek to identify whether there is potential for any negative psychological consequences to one of the current non-pharmaceutical interventions under consideration in the UK to reduce COVID-19 mortality and healthcare demand; hence this study appears both timely and relevant.

The authors provide a well written and concise summary of current and past research relating to this area of research (although I am not expert in this area, so default to the other reviewers and editor for their judgement on this aspect). I think that the study asks some interesting questions that are relevant and would be a useful addition to the literature. The authors provide good amount of detail and additional materials to adequately assess their plan.

The authors plan to test four hypothesis sets:

1. the potential effects of preference on ill-being at Time 2 (Hypothesis 1:1, 1:2),
2. the potential effects of identified and external motivations on ill-being at Time 2, above and beyond preference (Hypothesis 2:1, 2:2),
3. the possible moderating effects of identified motivations on the main effect of preference and ill-being at Time 2 (Hypothesis 3:1, 3:2; 3:3),
4. the possible moderating effects of external motivations on the main effect of preference and ill-being at Time 2 (Hypothesis 4:1, 4:2; 4:3).

I think that these hypotheses seem justified with theoretical motivations supported by previous work, both by the authors themselves and others. Many aspects of the study design have been well thought through with consideration of covariates and potential influential factors. There are, however, some details that need some attention and more explanation in my opinion.

I have some concerns that there are potential mediators, moderators and confounding variables that could affect how inferences relating to these hypotheses could be interpreted and findings generalised. I think the authors' proposed statistical analyses are by no means incorrect but perhaps do not capture the whole story. Hence, I would recommend major revisions in this instance (please note I did consider minor revisions but I think the point 1. - below - may substantially change the analysis plan).

Specific points that need to be addressed:

1. ***“To the extent that self-isolation results in loneliness, it may increase cardiovascular activation and cortisol levels, and reduce sleep, interfering with health and recovery (Cacioppo et al., 2002). Isolation is furthermore a risk factor for depression and anxiety among other indicators of ill-being (Cuenya, Fosachea, Mustaca, & Kamenetzky, 2012; Franck, Fosachea, Mustaca, & Kamenetzky, 2016).”***

My primary concern is how the authors will tease apart whether changes in the anxiety or other ill-being indicators in participants are resultant from self-isolation and not concerns over the virus and its impact on their health? Perhaps a second group could be studied that isolated in family units or couples or housemates/friends?

It would also be worth noting that referred self-isolation (usually high-risk groups) may have already greater ill-being. This can be controlled at baseline but assumes that changes in these groups are comparable to the lower risk groups which I think is unlikely. Higher risk groups are also far more likely to self-isolate for longer periods so changes may again follow a different pattern.

Is ‘isolation setting’ a factor? For example, if you self-isolate in a luxury one bed apartment with scenic surroundings and views vs a damp ground floor flat with poor facilities and minimal natural light. Perhaps this affects the relationship?

2. ***“Significance. While self-isolating is an effective strategy to “flatten the curve” of coronavirus infections by preventing the risks of contracting the virus, it may have psychological consequences studied in past research”***

My question relates to the time frame of the isolation period and duration of the study. From the above statement, do the authors refer to self-isolation in this study as a preventative step prior to contracting the virus rather than self-isolation due to observing symptoms?

If yes (preventative), then why was the justification for study follow-up defined at two weeks? Is this sufficient time to see potential changes in ill-being? Are all participants entering the study on their first day of isolation?

If no (symptoms), the study duration is presumably 14 days to match the proposed isolation period to avoid transmission in response to showing symptoms? If so, the more vulnerable groups are not considered in this study as they are likely to self-isolate for much longer periods prior to symptoms, so may behave differently than those in a predefined short period of self-isolation. It would be useful to include a reference to the 14 day isolation period to ensure that this is clear to future readers. Also, if the period of self-isolation refers to those with symptoms, then how do we differentiate that the individual’s ill-being is resultant on the psychological effect of isolation and not ‘feeling bad’ due to symptoms of COVID-19?

From a different perspective, I was unsure whether use of social media or viewing of news reports with a negative or worrying message during this period would also increase ill-being for certain individuals (moderating effect)?

Finally, do the authors record 'significant events' within the two week follow-up that might affect the ill-being indicators of the individual (other than isolation itself); for example, redundancy, loss of earnings, availability of food or medical supplies. This may cause a significant change ill-being response.

Minor comments:

1. Thank you to the authors for considering reliability of composite measures. I wholeheartedly agree with their approach and consideration of minimum $\alpha = .70$.
2. Smallest effect size of interest is a good approach in this circumstance as there is no precedent given that the authors are the first to test this hypothesis. I think that the proposed effect sizes of interest seem reasonable and avoiding the reliance on interpreting the p-values alone is good. I would ask the authors to also consider including confidence intervals or standard errors around the estimates to give the reader a sense of uncertainty around the estimates.
3. Both partial r and hierarchical regression are reasonable choices under the current study set up, but I think given my earlier comments, this may need some adjustment and potentially a different approach. It is difficult to recommend an alternate approach as it will depend on how the authors respond to my earlier comments. Path analysis or causal inference could be options?
4. Do the authors record whether participants have a pre-existing ill-being related condition such as depression? Should this be considered too?
5. It was great to see that the authors will be correctly evaluating their model fit by considering the distribution of the residuals (many authors skirt this issue and it can be detrimental to the findings). The authors are most likely well aware but by transforming the outcome variable, the estimates are also then interpreted on this scale. If the authors are wanting to avoid this complication, they can consider a generalized linear model instead (this is not a specific comment to be addressed as the authors suggestion is perfectly valid).

Appendix B

Dear Prof. Chambers,

We are happy to be submitting a majorly revised version of the Stage 1 manuscript, "Motivation and Preference in Isolation: A test of their different influences on responses to self-isolation during the COVID-19 outbreak" (RSOS-200458). We thank you and the reviewers for providing us an attentive review in a very short time. The recommendations have been extremely constructive.

As a result of the changes we have made, the revised Stage 1 manuscript offers more explicit rationales for methodological decisions, and we have included additional measures as controls, resulting in a more robust test of our hypothesized effects.

After some consideration, we have decided to conduct an additional assessment that took place yesterday (March 22). We saw Reviewer 1's point 1, that in this quickly changing landscape experiences of self-isolation may shift more frequently than they typically would. This second time-point is described in the new report and below. Simply, it is identical to the follow-up we had initially proposed, with the addition of a number of surveys included based on reviewer recommendations. Specifically, we measure ill-being (now across 1 week), our own control and descriptive variables described in the previous submission (e.g., in-person social contact; virtual social contact [now a covariate]), and new control variables suggested by reviewers (e.g., health anxiety; presence of stressors, COVID-19 diagnosis).

We should also note that since we now plan to statistically control for health concerns in a number of ways, we have removed *physical symptoms* from our ill-being composite. We recognised that although physical symptoms are often regarded as a form of ill-being, they were problematic in the context of COVID-19 since they could be either a moderator, control, or outcome variable.

Below we address each editor and reviewer comment in turn. To facilitate the review, where we felt specific manuscript content was particularly important to read we have copied it into the letter as green text. New or modified text within the manuscript is also signified in green.

Editor Comments

Point 1. *Further methodological detail and justification of design decisions is required (such as inclusion criteria, the basis of the 2 week follow-up period, and the smallest effect size of interest; see Reviewer 1). Note that Reviewer 4 (point 5) offers a potential solution to this latter issue.*

Response. We now more consistently justify methodological and analytic decisions we have made. On pp. 18-19 we have justified the effect size of interest, and we have also copied the new text into the letter under Reviewer 1, point 5.

Point 2. *Reviewer 2 notes several key areas in the analysis plan requiring attention, with particular concerns about confounding variables, and the most major concern being the ability of the design to single out the effects of self-isolation. Reviewer 3 raises a related concern about distinguishing the effects of physical isolation from the effects of social isolation.*

Response. Thank you for highlighting this point. In revising our manuscript, we have made our focus on solitude explicit in the Purpose section of our proposal (p. 3). Summarised here, we seek to understand the experience of pure solitude within the context of self-isolation. In other words, our focus was to understand what happens when people experience both the social and physical isolation aspect of self-isolation together. The combination is more challenging and also a purer form of solitude (Nguyen et al., 2018) and thus findings framed in this manner will speak not only to self-isolation, but to the literature surrounding solitude more broadly.

Reviewer 1

This is an interesting piece of research making use of the current climate to evaluate an important and scientifically valid research question. The hypotheses were well considered and the methodology is plausible to allow these hypotheses to be answered. The quality checks were thought through in light of the naturalistic design. Overall, the methodology of the study was sufficiently detailed to allow further replication, and prevent undisclosed flexibility. Whilst the first phase of data has already been collected this seems justified given the required rapid commencement of the study. A few minor comments are detailed below:

Response. Thank you!

Point 1. *Some clarification as to why a 2 week follow-up period was selected may be helpful. Given that this is a quickly changing situation I wonder if motivations are likely to fluctuate throughout a 2-week period. The authors may consider a greater number of shorter-term follow-ups or expanding upon their justification for why 2 weeks was selected.*

Response. The reviewer was totally correct. Given recent turns of events in the US and UK, we have added a one-week follow-up survey on March 22, but still plan to conduct a final survey on March 29. This means the two-part study is now a three-part study with an initial (1 week) and delayed (2 weeks) measurement of ill-being and control variables. We describe this decision on p. 6 of the manuscript, specifically, “The move towards weekly data collection (and three time points) was made following reviewer advice to recognise the quickly changing nature of communications related to COVID-19. For example, the UK government announced school and bar/pub closures to take place by the end of March 20th, 2020 (The Guardian), Furthermore, although the US government has not yet made any decisions to take more stringent social distancing measures, we can also expect a shift in motivation and perceived urgency of self-isolation on March 23th, 2020, following recent recommendation on March 15th, 2020, by CDC to prevent gatherings of more than 50 people, and by statewide school closures announced by more than 30 states (CNN). It seems, then, that weekly, not bi-weekly, measurements of ill-being are needed to understand ill-being before and after these significant changes to people’s understanding and expectations for self-isolation.”

Point 2. *The authors have chosen to limit their sample to individuals living alone. Whilst this limits the influence of interaction with close others on ill-being it does represent the more extreme cases in the community. It would be worth highlighting and justifying this.*

Response. We have added to the manuscript a justification for why we are studying this population of living-alone adults (p. 3, under Purpose). Specifically, we state: “When self-isolating, individuals reduce the number of others with whom they have contact by staying at home, but whether it involves physical isolation (being physically separated from others) and social isolation (being physically separate from, and also not interacting with, others) can vary. Many, including those living with others as well as those living alone, are likely to resort to virtual interactions, which replace in-person interactions as the main source of communication (Chambers, D. 2013). For those who live with others, time without virtual interactions can still be rich with in-person interactions. For those living alone, time without virtual interactions means truly being in solitude. Because our interest is in the influence of both preference and motivation for time alone on the psychological consequences of adults, we focus on a context when psychological isolation becomes the most significant issue: when living-alone adults do not interact with anyone in person and virtually.”

We also wanted to add here that there are notable percentages of adults in the United States and United Kingdom that live in single-adult households. According to Our World in Data, living alone adults in the US take up more than 50% of the population, driven mainly by older adults (2016 data: 39.98% age 89, 25.08% age 75, 17.89% age 60, 9.45% age 45, and less than 15% younger than 30). According to the data from the Office for National Statistics, in 2019, the

highest one-person household proportion is 35% of households in Scotland and the lowest is 23.9% in London. Therefore, we do not believe living-alone adults represent extreme cases, though certainly considering the context of self-isolation, this constitutes a particularly vulnerable group. We will explore this issue in the Discussion section of the Stage 2 report.

Point 3. *When measuring 'Motivation for solitude' the scale asks participants to think about times when they are not interacting with anyone in person or virtually. Given that virtual interaction does not have any health implications this would seem to be measuring general motivation for solitude, rather than the motivation in this particular health environment. It may be worth separating in person and virtual interactions in the 2nd phase of data collection to distinguish between these.*

Response. We see the reviewer's point, and believe that it can be addressed with additional clarity to the manuscript in two places. First, please see our response to Editor, Point 2 and the new paragraph in the Purpose section of the manuscript (p. 3), where we better explain why the focus of the study is in social AND physical isolation (that is, hypotheses were not relevant to virtual interactions). Simply, the combination of both represents the 'purest' form of solitude, and the most challenging (Nguyen et al., 2018).

The second important detail we have better communicated in the current draft of the manuscript (p. 13) is that all dependent variables refer to ill-being when at home, and not interacting with anyone in-person or virtually. Although our measures posted on OSF used this phrasing, we had neglected to include it in the description of the method (instead, we stated that measures would assess being "home and alone").

We could ask participants to report ill-being variables related to both time alone without virtual interactions and again those outcomes without in-person interactions, but this would make the survey much longer, and those two measures of ill-being are likely to be highly correlated. Therefore, when we made the decision to test the more extreme self-isolation situations to understand the psychological challenges of people when in full solitude during this outbreak, and because we felt it would be more informative to the broader solitude literature.

Nguyen, T. V. T., Ryan, R. M., & Deci, E. L. (2018). Solitude as an approach to affective self-regulation. *Personality and Social Psychology Bulletin*, 44(1), 92-106.

Point 4. *Have any checks for data quality been included in the Qualtrics questionnaires? It may be worth considering adding attention check questions in the 2nd phase of the study to assess data quality.*

Response. On the bottom of p. 21, we describe the new attention check questions included in Time 2: "We will test for inattention with two items "Choose "somewhat agree"" and "Choose "Very true"" embedded in two different scales. We will exclude participants who 1) fail to select the corresponding answers as instructed, and 2) fail at least one attention check item (if the participant misses the item, that is considered failing the attention check)."

Point 5. *Further justification of what basis the smallest effects of interest were chosen on may be useful.*

Response. We choose the smallest effects of interest as a conservative test of the effect we will decide to be meaningful. We added the following justifications to the manuscript (p. 18-19):

Rationale for choosing partial r around 0.15. Rationale for choosing partial r around 0.15.

A previous paper by Nguyen, Werner, and Soenens (2019), looked at the link between autonomous motivation (conceptually similar to high identified motivation and low external motivation) and well-being and ill-being variables. In Study 1, sampling first-year university students ($n = 147$), the authors measured motivation for spending time alone and loneliness experienced in the past 2 weeks (not specifically related to time spent alone) and assessed

these variables three times. On average, the correlation between autonomous motivation and loneliness was approximately $r = -.30$. Extraversion - a concept similar to preference for solitude - also correlated with loneliness at also around $r = -.30$. However, because these correlations were obtained from a rather small sample, effect sizes were likely overestimated (Yartoki, 2009). As such, we decide to go with the partial r half the size of those correlations obtained by past research.

Rationale for choosing .10 difference between regression coefficients. Rationale for choosing .10 difference between regression coefficients. In Study 2 of Nguyen et al. (2019), the researchers also examined the interaction between participants' perceived belonging with autonomous motivation predicting levels of loneliness experienced in the past one month; this moderation test is the closest available data to our hypothesized moderation effects (Hypothesis sets 3 and 4) and yielded a coefficient of, $\beta = -.32$, CI 95% = $[-0.44, -0.20]$, when predicting loneliness. A high levels of autonomous motivation, perceived belonging related to loneliness, $\beta = -0.04$, CI 95% = $[-0.21, 0.13]$. Again, findings relied on a sample of $n = 223$, we expect the effect sizes were likely overestimated. So if we go with regression coefficients half the size of those coefficients obtained by past research, we might anticipate, $r = -0.16$ at low levels of autonomous motivation, and, $r = -0.02$ at high levels. The difference between these two coefficients was about $\Delta r = .14$, but a conservative expectation is $\Delta r = .10$.

Yarkoni, T. (2009). Big correlations in little studies: Inflated fMRI correlations reflect low statistical power—Commentary on Vul et al. (2009). *Perspectives on Psychological Science*, 4(3), 294-298.

Point 6. *The authors may consider incorporating a brief global measure of change in wellbeing in the 2nd follow-up to compare to their composite measures*

Response. We agree it would be very interesting to include a global measure of well-being in this study. However, we are concerned that, were we to include it as another dependent variable, this doubles the number of analyses to be conducted. We were also concerned that measuring ill-being as we do in relation to solitude would make the survey too long. After some thought we have decided that a compromise between the ideal option (measuring global ill-being) and these practical considerations is the following:

We have included our measure of depression used as part of the ill-being composite, this time measuring depression globally. Global depression seemed the most appropriate outcome because all three of the other measures within solitude: loneliness, depression, and anxiety, should be strongly related to it (Erzen & Çikrikci, 2018; Pollack, 2005; Weeks et al., 1980).

At present we do not plan to analyse it within confirmatory analyses, but are open to reconsidering this view if it is felt to be important. We would instead like to test it in an exploratory analysis linked with the ill-being in solitude measures. This would allow us to explore what percentage of variance is shared between ill-being in time spent alone, and global ill-being. We see this as a way of evidencing the importance of ill-being during solitude.

Erzen, E., & Çikrikci, Ö. (2018). The effect of loneliness on depression: A meta-analysis. *International Journal of Social Psychiatry*, 64(5), 427-435.

Pollack, M. H. (2005). Comorbid anxiety and depression. *The Journal of clinical psychiatry*, 66, 22-29.

Weeks, D. G., Michela, J. L., Peplau, L. A., & Bragg, M. E. (1980). Relation between loneliness and depression: A structural equation analysis. *Journal of personality and social psychology*, 39(6), 1238.

Reviewer: 2

The authors plan to test four hypothesis sets:

1. the potential effects of preference on ill-being at Time 2 (Hypothesis 1:1, 1:2), 2. the potential effects of identified and external motivations on ill-being at Time 2, above and beyond preference (Hypothesis 2:1, 2:2), 3. the possible moderating effects of identified motivations on the main effect of preference and ill-being at Time 2 (Hypothesis 3:1, 3:2; 3:3), 4. the possible moderating effects of external motivations on the main effect of preference and ill-being at Time 2 (Hypothesis 4:1, 4:2; 4:3).

I think that these hypotheses seem justified with theoretical motivations supported by previous work, both by the authors themselves and others. Many aspects of the study design have been well thought through with consideration of covariates and potential influential factors. There are, however, some details that need some attention and more explanation in my opinion.

I have some concerns that there are potential mediators, moderators and confounding variables that could affect how inferences relating to these hypotheses could be interpreted and findings generalised. I think the authors' proposed statistical analyses are by no means incorrect but perhaps do not capture the whole story. Hence, I would recommend major revisions in this instance (please note I did consider minor revisions but I think the point 1. - below - may substantially change the analysis plan).

Response. Thank you very much for your attention to the manuscript!

Specific points that need to be addressed:

Point 1a. *“To the extent that self-isolation results in loneliness, it may increase cardiovascular activation and cortisol levels, and reduce sleep, interfering with health and recovery (Cacioppo et al., 2002). Isolation is furthermore a risk factor for depression and anxiety among other indicators of ill-being (Cuenya, Fosachecca, Mustaca, & Kamenetzky, 2012; Franck, Fosachecca, Mustaca, & Kamenetzky, 2016).”*

My primary concern is how the authors will tease apart whether changes in the anxiety or other ill-being indicators in participants are resultant from self-isolation and not concerns over the virus and its impact on their health? Perhaps a second group could be studied that isolated in family units or couples or housemates | friends?

Response. To account for concerns over symptoms or illness, we included two new assessments to Time 2 and 3 measurements (also described in p. 13 of the manuscript): 1) we have added two items asking whether participants have been diagnosed or seriously suspect having COVID-19, and 2) we have included five items¹ from the Short Health Anxiety Inventory (Salkovskis et al., 2002) which target health concerns directly. Although the full measure was developed as an individual difference, we closely adapted it as a measure of state health anxiety over a period of one week.

Salkovskis, P. M., Rimes, K. A., Warwick, H. M. C., & Clark, D. M. (2002). The Health Anxiety Inventory: development and validation of scales for the measurement of health anxiety and hypochondriasis. Psychological medicine, 32(5), 843-853.

Point 1b. *It would also be worth noting that preferred self-isolation (usually high-risk groups) may have already greater ill-being. This can be controlled at baseline but assumes that changes in these groups are comparable to the lower risk groups which I think is unlikely. Higher risk groups are also far more likely to self-isolate for longer periods so changes may again follow a different pattern.*

¹ Originally, we picked out six items from the Short Health Anxiety Inventory (SHAI). However, after we launched the Time 2 survey, we caught one erroneous item that said “I noticed aches and pains less than more than usual”. We will exclude this item from the composite. In the manuscript, we describe that the final SHAI measure as a 5-item scale.

Response. It is reasonable to anticipate that those who prefer spending time alone are also vulnerable to ill-being (and, in fact, one of our hypotheses!); however, it is worth noting that we measure general preference for solitude, not preference for self-isolation out of health concerns related to COVID-19. We can further account for vulnerability in two ways: 1) by controlling for the single item subjective health measure we had included at Time 1 and meant as a descriptive measure (see p. 7 for its description), and 2) because now, in response to point 1a, we measure and control for health concerns and suspicion of being infected with COVID-19 in the past two weeks (p. 12-13). In other words, setting aside psychological vulnerability operationalised as baseline ill-being, the new health anxiety measure should capture those who are more vulnerable in a way that specifically shapes the nature of their self-isolation. We will also make sure to discuss the issue of vulnerability in the Discussion section of the full manuscript.

Point 1c. *Is 'isolation setting' a factor? For example, if you self-isolate in a luxury one bed apartment with scenic surroundings and views vs a damp ground floor flat with poor facilities and minimal natural light. Perhaps this affects the relationship?*

Response. Thank you for this idea. We agree that the question of how physical space influences experiences with self-isolation is a very interesting one, but we feel it is outside the scope of the current study. In part, because we are testing three predictors and proposing 2-way interactions, we are concerned about the reliability of any findings that try to moderate those effects without ad-hoc justification.

Point 2. *"Significance. While self-isolating is an effective strategy to "flatten the curve" of coronavirus infections by preventing the risks of contracting the virus, it may have psychological consequences studied in past research"*

My question relates to the time frame of the isolation period and duration of the study. From the above statement, do the authors refer to self-isolation in this study as a preventative step prior to contracting the virus rather than self-isolation due to observing symptoms?

If yes (preventative), then why was the justification for study follow-up defined at two weeks? Is this sufficient time to see potential changes in ill-being? Are all participants entering the study on their first day of isolation?

If no (symptoms), the study duration is presumably 14 days to match the proposed isolation period to avoid transmission in response to showing symptoms? If so, the more vulnerable groups are not considered in this study as they are likely to self-isolate for much longer periods prior to symptoms, so may behave differently than those in a predefined short period of self-isolation. It would be useful to include a reference to the 14 day isolation period to ensure that this is clear to future readers. Also, if the period of self-isolation refers to those with symptoms, then how do we differentiate that the individual's ill-being is resultant on the psychological effect of isolation and not 'feeling bad' due to symptoms of COVID-19?

Response. Generally, we are interested in both those who are in self-isolation for prevention and symptoms. In other words, holding constant symptoms and concerns over health, we want to understand the experience of being alone, in self-isolation. Yet the reviewer brings up a great point, which is that physical health and a COVID-19 diagnosis very likely affects both our predictors and outcomes. To account for this we will measure in Time 2, and control for, diagnosis of or the suspicion that one has COVID-19 (described p. 12).

To clarify the period of time selected, the decision to conduct a 2-week follow-up was a practical one. Previous investigations around extreme isolation suggested that negative effects could emerge after a period as short as 48 hours (BBC). Because following up every 2 days was not practical, we relied on the information around how long a person should self-isolate in the case that they might have contracted COVID-19.

First, at the point of selecting the study, the global response was a two-week self-isolation period and we wanted to ensure that the push towards self-isolation would not end before collecting Time 2. We judged that two weeks would be sufficient time spent alone at home to see sufficient changes, and also not too long that the findings would inform long-term but not exceedingly unusual solitude experiences.

Second, the decision to follow up with ill-being outcomes is also informed by research that often measures well-being and ill-being outcomes during the past few weeks, ranging from 1 week to 3 weeks between the 1st and follow-up surveys (Baker, Tou, Bryan, & Knee, 2017; Franz et al., 2012).

Following Reviewer 1, Point 1, we recognise the concern that these two weeks have seen significant changes due to other factors unrelated to social isolation, so have selected to add one assessment in between the two previously proposed time points.

Point 3. *From a different perspective, I was unsure whether use of social media or viewing of news reports with a negative or worrying message during this period would also increase ill-being for certain individuals (moderating effect)?*

Response. Similar to our response to point 1c, we agree that exposure to social media and conflicting news outlets is a fascinating topic of study. However, we feel we must focus the current study on the topic of preference and motivation for spending time alone, which involves testing a number of novel and somewhat complex research questions.

Point 4. *Finally, do the authors record 'significant events' within the two week follow-up that might affect the ill-being indicators of the individual (other than isolation itself); for example, redundancy, loss of earnings, availability of food or medical supplies. This may cause a significant change ill-being response.*

Response. Thank you for this point, you are absolutely correct these events are important to account for. We now measure significant events with a 'stressor checklist' inspired by the list above and supplementing with several related themes. See the measure constructed on p. 13; we intend to include it as a covariate in all the main models.

Minor comments:

1. *Thank you to the authors for considering reliability of composite measures. I wholeheartedly agree with their approach and consideration of minimum $\alpha = .70$.*

Response. Thank you for this point.

2. *Smallest effect size of interest is a good approach in this circumstance as there is no precedent given that the authors are the first to test this hypothesis. I think that the proposed effect sizes of interest seem reasonable and avoiding the reliance on interpreting the p-values alone is good. I would ask the authors to also consider including confidence intervals or standard errors around the estimates to give the reader a sense of uncertainty around the estimates.*

Response. We agree. We will also report confidence intervals around estimates.

3. *Both partial r and hierarchical regression are reasonable choices under the current study set up, but I think given my earlier comments, this may need some adjustment and potentially a different approach. It is difficult to recommend an alternate approach as it will depend on how the authors respond to my earlier comments. Path analysis or causal inference could be options?*

Response. Following peer review feedback, we have decided to add a time point one week after Time 1 survey. This will allow us to catch any change within 1 week of social isolation, given how quickly events have escalated in the past week.

After very serious consideration, we would like to continue to use hierarchical regression to predict change in ill-being from Time 1 to Time 2, and from Time 2 to Time 3. With this approach, we will be able to predict short-term change one week after the initial assessment, and also change after isolation has been prolonged and further exacerbated by public debates two weeks after the initial assessment. In interpreting results at two weeks after baseline (Stage 2 submission), we will discuss Time 3 results in light of public debates and discussions happening at the time.

4. *Do the authors record whether participants have a pre-existing ill-being related condition such as depression? Should this be considered too?*

Response. We measure depression at baseline, and while we do not measure clinical levels of depression, and are not well-equipped to make clinical judgments, the CES-D measure reflecting symptom expression across six months is considered a valid reflection of depressive symptoms (e.g., Weissman et al., 1977). It will be controlled for as part of our ill-being indicator (p. 9).

Weissman, M. M., Sholomskas, D., Pottenger, M., Prusoff, B. A., & Locke, B. Z. (1977). Assessing depressive symptoms in five psychiatric populations: a validation study. *American journal of epidemiology*, 106(3), 203-214.

5. *It was great to see that the authors will be correctly evaluating their model fit by considering the distribution of the residuals (many authors skirt this issue and it can be detrimental to the findings). The authors are most likely well aware but by transforming the outcome variable, the estimates are also then interpreted on this scale. If the authors are wanting to avoid this complication, they can consider a generalized linear model instead (this is not a specific comment to be addressed as the authors suggestion is perfectly valid).*

Response. Thank you for this suggestion. Generalized linear models could address cases where assumption of normality of the residuals (outcome variables) is violated. We will use generalized linear models instead of simple linear models if errors are not normally distributed.

Reviewer: 3

The current proposal is intended to examine the effects of preference for isolation and quality of motivation for self-isolation. In general, higher levels of ill-being (e.g., depressive symptoms, anxiety, physical health issues, etc.) are expected to be associated with lower preference for solitude, lower identified regulation, and higher external regulation (as well as the appropriate interactive effects). This work has a strong theoretical basis, the research questions are thoroughly explained, and the analytic approach is clearly designed to properly test the hypotheses. This will be a meaningful and timely contribution to the literature, and I'm very eager to see this work carried through to completion.

Response. Thank you!

Point 1. *My only (relatively minor) concern is that there may be confusion between physical isolation and social isolation, both as the participants respond to the items, and in the eventual interpretation of these findings. As the authors know (very well), according to Self-Determination Theory, a lack of satisfaction of basic psychological needs is associated with an increase in ill-being. These research questions focus on the impact of autonomy support for self-isolation, but the measurement may also inadvertently pick up lack of satisfaction for relatedness.*

a. According to the questionnaire documents on the OSF, the frequency of social interaction questions are different across the two time points (not including the time frame):

T1: "How frequently, in general, do you perform your daily activities at home by yourself?"

T2: "How frequently, in the past 2 weeks, have you performed your daily activities while at home alone, not interacting with anyone in person or virtually?"

The T1 item focuses on physical isolation, while the T2 item also includes social isolation. This measure is only intended to "serve as a quality indicator testing whether participants were in solitude across two weeks" (p. 10). However, if the authors observe increases in ill-being, this may be a potential alternate explanation. It is certainly possible that a person (especially one high in preference for solitude) might be able to be physically alone, even in self-isolation, but still maintain connections to others.

Response. We see this being a very valid point. However, it might be more appropriate for a study focused on global wellness as a function of both solitude and social connection. In the focus of the current study, which we now make more explicitly in the manuscript (e.g., p. 3), T2 is a more specific indicator of the dynamics under study: namely, experiences while alone and not interacting with others virtually. On a somewhat different note but perhaps more important for addressing this issue, we now control for virtual social interactions as well as in-person ones following your suggestion "b" below. We feel this additional control variable also accounts for your well-placed concern, that being at home, alone, can still be a social experience for those individuals who prefer social situations.

b. In addition to frequency, participants will also respond to items asking the ways in which they interacted with others. In-person interactions are expected to decrease (which is a good check, as mentioned by the authors), and the "face-to-face item will be included as a covariate in all main models" (p. 10). I would suggest an additional set of analyses controlling for all modes of contact (face-to-face, social media, phone/text).

Response. This is a good point. We will include virtual interactions as an additional control. Please see p. 23 where we describe our new analytic strategy including which controls we will account for.

Reviewer: 4 (Associate Editor)

The study will address an important question related to motivation of self-isolation and its relationship to ill-being. The registered report is well documented in its methods, hypotheses and statistical analyses. I have only few comments:

Point 1a. *It would be better to explain why >35 years old has been set as inclusion criteria.*

Response. Our explanation in the revised manuscript (p. 21) was that: ...the inclusion criterion of 35+ years of age was set because this age represents a transition to adulthood, rather than young adulthood (Coleman, 1972). The focus on adults was important for two reasons: First, since COVID-19 targets adults and older adults at higher rates, we anticipated that self-isolation is more relevant to, and more likely in, these individuals as compared to young or emerging adults, typically considered under 35 years (Coleman, 1972). Second, since risky behavior is higher in adolescents to young adults under age 35 years (Bingham & Shope, 2004; Holliday, 1988; Mata et al., 2011), we sought a sample that would make more responsible and consistent health decisions to self-isolate regardless of their personal preferences. We apologise, we should have provided this explanation from the start, and we do so now in the Inclusion section (pp. 20 and 21).

Bingham, C. R., & Shope, J. T. (2004). Adolescent developmental antecedents of risky driving among young adults. *Journal of studies on alcohol*, 65(1), 84-94.

Holliday, S. G. (1988). Risky-choice behaviour: A life-span analysis. *The International Journal of Aging and Human Development*, 27(1), 25-33.

Mata, R., Josef, A. K., Samanez-Larkin, G. R., & Hertwig, R. (2011). Age differences in risky choice: a meta-analysis. *Annals of the New York Academy of Sciences*, 1235, 18-29.

Coleman, J. S. (1972). How do the young become adults?. *Review of Educational Research*, 42(4), 431-439.

Point 1b. *Related to this, not sure it makes sense to pull together the data gathered via Social media , given they have different inclusion criteria.*

Response. It is a good point. Since we were not able to recruit enough participants through social media to assess them separately, we will exclude them from analyses. We have noted they were collected but excluded now as Footnote 1 (p. 7).

Point 2. *“In the past 2 weeks, were you specifically following ____ recommendations to self-isolate (or to stay home and away from other people and places)” I would give an “other” option, that is open end*

Response. We have now done so (see new materials on OSF [https://osf.io/mn8kx/?view_only=bca3f190b56847d8994f2fe886dcf0ba] and described in the text of the Registered Report (p. 9).

Point 3. *Control Measures Time 1 (baseline). For all these measures, it is not clear why you ask about the last 6 months. I would imagine you will have a recency effect (at least in some) given the outbreak of the Corona Virus. I would rather ask in the past weeks, and see how this change from T1 to T2*

Response. We thought long and hard about this issue, and we think that is a good point. The reasoning behind measuring Time 1 ill-being in the past 6 months is that we did not want to be assessing a shock reaction to the COVID-19 outbreak as things had been rapidly changing in the weeks before the study. Since we could not be confident of what short-term ill-being reflected at that point, we felt it would be more beneficial to assess tendencies towards ill-being (vulnerability). By reminding the participants to think about time before the outbreak becomes salient, we hope to capture general susceptibility to ill-being (depression, anxiety, loneliness) as an individual difference covariate. Our reasoning, that we should account for vulnerability at Time 1, was similar to Reviewer 2, point 1a.

Point 4. *Typo on line 47. Should be march 14*

Response. Thank you for pointing this out, this has now been corrected and we have double-checked all dates, since there now seem to be many of them.

Point 5. Check this preprint <https://psyarxiv.com/epcyb/> for possible sample size calculation

Response. Thank you for suggesting this preprint. It is a very interesting paper. Specifically, the authors found that people generally have greater optimism that they will be less likely to infect other people with COVID-19, yielding large effect size $\eta_p^2 = .48$. However, this optimism yielded small correlation with how much they bias toward their own judgment about the necessity of social distancing more than what they thought others would judge the necessity of social distancing ($r = -.13$), with a sample of more than $n = 800$ (approximately our Time 1 sample). This suggests that people are not good judges of how their attitudes toward an issue are linked to their attitudes toward the behaviour necessary to solve that issue. In our study, we look at the link between people’s reasoning (i.e. preference and motivation) toward a behaviour and their reported psychological experience while carrying out that behaviour a week and two weeks later.

To summarise, we have very carefully considered each recommendation. We thank reviewers and editor for providing helpful feedback to inform a study that had just days in between waves

of ongoing data collection. We look forward to your feedback on our revised manuscript and research plan.

Sincerely,

Netta Weinstein and Thuy-vy Nguyen

Appendix C

Running Head: Preference and Motivation for Solitude 1

Motivation and Preference in Isolation: A test of their different influences on responses to self-isolation during the COVID-19 outbreak

Netta Weinstein
University of Reading
Cardiff University

Thuy-vy Nguyen
Durham University

Abstract

This multi-wave study examined the extent that both preference and motivation for time alone shapes ill-being during self-isolation. Those in the US and UK are ~~likely to self-isolate~~ ~~isolating fully or in part~~ in response to the COVID-19 outbreak. Different motivations may drive their self-isolation: some might see value in it (understood as the identified form of autonomous ~~form of~~ motivation), while others might feel forced into it ~~either by local~~ authorities or ~~by concerned~~ ~~close~~ others (family, friends, neighbourhoods, doctors; the external form of controlled motivation). ~~Many p~~ People who typically prefer ~~to be with others rather than alone~~ ~~company~~ will find themselves spending more time alone, and may experience differential ill-being ~~experienced either~~ uniformly, or as a function of their identified or external motivations for self-isolation. Self-isolation therefore offers a unique opportunity to distinguish two constructs coming from disparate literatures. This project ~~will examine~~ ~~examined~~ preference and motivation (identified and external) for solitude, and ~~tested~~ their independent and interacting contributions to ill-being (loneliness, depression, and anxiety during time spent alone) across two weeks. ~~Confirmatory~~ hypotheses regarding preference and motivation were not supported by the data. A statistically significant effect of controlled motivation on change in ill-being was observed one week later, and preference predicted ill-being across two weeks. However, effect sizes for both were below our minimum threshold of interest.

Keywords: *Motivation; Solitude; Self-isolation; COVID-19; Loneliness*

~~1) What is the main question being addressed in your study?~~

Formatted: Font: Bold

Motivation and Preference in Isolation: A test of their different influences on responses to self-isolation during the COVID-19 outbreak

Formatted: Centered, Indent: First line: 0 pi

~~Purpose.~~ The purpose of this study ~~is~~ was to understand how ~~both~~ motivation and preference for solitude - two conceptually distinct constructs from disparate literatures that might have unique or interacting effects on individuals' responses to being in solitude - influence ill-being during self-isolation. To achieve this purpose, we focused on solitude during self-isolation operationalised in terms of both physical and social isolation, arguably its purest form (Nguyen et al., 2018).

When self-isolating, individuals reduce the number of others with whom they have contact by staying at home, but whether it involves physical isolation (being physically separated from others) and social isolation (being physically separate from, and also not interacting with, others) can vary. Many, including those living with others as well as those living alone, are likely to resort to virtual interactions, which replace in-person interactions as the main source of communication (Chambers, 2013). For those who live with others, time without virtual interactions can still be rich with in-person interactions. For those living alone, time without virtual interactions means truly being in solitude. Because our interest is in the influence of both preference and motivation for time alone on the psychological consequences for adults, we focus on a context when psychological isolation becomes the most significant issue: when living-alone adults do not interact with anyone in person and virtually.

Significance. While self-isolating is an effective strategy to “flatten the curve” of coronavirus infections by preventing the risks of contracting the virus, it may have psychological consequences studied in past research. To the extent that self-isolation results in loneliness, it may increase cardiovascular activation and cortisol levels, and reduce sleep, interfering with health and recovery (Cacioppo et al., 2002). Isolation is furthermore a risk factor for depression and anxiety among other indicators of ill-being (Cuenya, Fosachecca, Mustaca, & Kamenetzky, 2012; Franck, Fosachecca, Mustaca, & Kamenetzky, 2016). Dispositional preference to be alone, and motivation for self-isolation, may both mitigate the potential psychological costs (Nguyen, Ryan, & Deci, 2018), though little is understood about their independent and interdependent contributions. This study will build our understanding of the roles these constructs play using transparent and robust research methods.

Preference and Motivation for Solitude

Background. Few conclusions can be drawn from the existing literature about how preference for solitude predicts ill-being associated with time spent in isolation. Research shows that people who prefer to be alone (Burger, 1995) are broadly vulnerable and report ~~poor~~ ill-being, although there are mixed findings concerning their ill-being when they are in solitude, specifically (Nguyen, Weinstein, & Ryan, Under Review). Although no research of which we are aware has investigated this, it is also plausible that preference for solitude would lead individuals to tolerate it better and therefore report *less* ill-being when self-isolating.

A separate literature, self-determination theory (SDT) highlights the importance of motivation driving behaviour (Ryan & Deci, 2017). Motivations are “hot”, energising reasons for behaviour in contrast to the “cold” cognition characterising preference; both meaningfully contribute to behaviour (Sorrentino & Higgins, 1986). Based in SDT, less autonomous or self-

driven motivation for solitude, and more controlled motivation reflecting pressure and choicelessness, relate to feeling lonely when alone (Nguyen et al.; Study 4). Further, endorsing autonomous reasons for spending time alone correlates negatively with ill-being outcomes like depression, loneliness, and anxiety (Thomas & Azmitia, 2019). However, these studies have conflated enjoying being alone with finding value in the activity, because both are theorised to ~~compromise~~ comprise autonomous motivation. One form of autonomous motivation is identified motivation, or selecting to spend time alone because it is seen as being beneficial and valuable (Nguyen et al., 2018). Identified motivation is particularly relevant in the case of self-isolation, since the reason for this behavior is its importance to the health of the self-isolating individual and/or to society rather than because it is intrinsically enjoyable or rewarding - another aspect of autonomous motivation. Therefore, the extent to which this motivation is endorsed needs to be examined in its own right rather than as part of an autonomy composite. Further, one form of controlled motivation, external motivation or feeling pressured by others and choiceless (Ryan & Deci, 2017), is relevant to instances of self-isolation which are felt to be driven by pressures from social-societal (government guidelines or mandates) or close relationship (family, friends, doctors). Further, external motivation is important to understanding the outcomes of solitude: forced solitude is thought to be the most detrimental form of solitude, and perhaps *only* when solitude is forced does it result in ill-being (Storr, 1988). The current study will therefore examine the main effects of both identified (autonomous) and external (controlled) motivations - “hot” energizing reasons to self-isolate, and their moderating effects on the relationship between preference, a “cold” reason, and ill-being.

Present Research

Though preference and motivation for solitude show different patterns of influence on ill-being, those who prefer solitude often also report more autonomous motivation for it (Nguyen et al., 2018; Nguyen et al., 2019; Thomas & Azmitia, 2019). However, self-isolation is an occasion in which one's longstanding preference to be alone versus in the company of others, and their current motivations to be alone, should be more strongly differentiated. Those who typically prefer social situations may now find additional value in time spent alone (e.g., for protecting their health), or alternatively they may feel pressured or choiceless around self-isolating. It may be either their preference, or their motivation, driving effects of ill-being during time spent alone. Self-isolation is therefore ideal for gaining a deeper theoretical understanding of these two predictors of ill-being in solitude.

~~We~~ ~~We~~ ~~have~~ pre-registered and collected Time 1 of the research

([OSF.IO/MN8KXhttps://osf.io/mn8kx/?view_only=bca3f190b56847d8994f2fe886def0ba](https://osf.io/mn8kx/?view_only=bca3f190b56847d8994f2fe886def0ba)) to minimise the amount of time respondents had spent in self-isolation before the start of the study. Following initial review of the Registered Report, we ~~further~~ collected Time 2 just over one week later (March 22) and ~~we plan to collect~~ Time 3 one week after Time 2 (March 29). The verbatim hypothesis, methodological and analytic sections of the Stage 1 manuscript were registered following Stage 1 in-principle acceptance and before Time 3 data collection at (https://mfr.de-1.osf.io/render?url=https://osf.io/g9dxb/?direct%26mode=render%26action=download%26mode=render). See also the full Stage 1 accepted manuscript at (https://mfr.de-1.osf.io/render?url=https://osf.io/y9xbt/?direct%26mode=render%26action=download%26mode=render), and the originally submitted research plan as a supplementary table.

~~We had initially planned to collect two, not three time-points, and that those would take place two weeks apart. The move towards to weekly data collection (and three time points) was made following reviewers' advice to recognise the quickly changing nature of communications related to COVID-19. For example, the UK government announced school and bar/pub closures to take place by the end of March 20th, 2020 (The Guardian), Furthermore, although the US government has had not yet made any decisions to take more stringent social distancing measures, we can also expected a shift in motivation and perceived urgency of self-isolation on March 23th, 2020, following earlier recent recommendation on March 15th, 2020, by CDC to prevent gatherings of more than 50 people, and by statewide school closures announced by more than 30 states on that date (CNN). It seems, then, that weekly, not bi-weekly, measurements of ill-being are were needed to understand ill-being before and after these significant changes to people's understanding and expectations for self-isolation.~~

We will set out to test four sets of hypotheses guided by current theory and research on preference and motivation for solitude. Simply, these concerned the potential effects of preference on ill-being at Times 2 and 3 (Hypothesis 1:1, 1:2), of identified and external motivations on ill-being at Times 2 and 3, above and beyond preference (Hypothesis 2:1, 2:2), and the possible moderating effects of identified (Hypothesis 3:1, 3:2; 3:3) and external (Hypothesis 4:1, 4:2; 4:3) motivations on the main effect of preference and ill-being at Times 2 and 3.

Competing Hypothesis Set 1. For the link between preference for solitude and psychological consequences of isolation, we ~~have had~~ competing hypotheses as follows:

Hypothesis 1:1 Preference for solitude would yield a negative correlation with ill-being 1 and 2 weeks later when participants report about time spent alone in the past week

Formatted: Font: Italic

(operationalised here and below as a composite formed of loneliness; depletion; loneliness; isolation; depression; anxiety). This ~~is~~ was based on the idea that for individuals with a greater preference for being alone, isolation will be a more pleasant experience.

Hypothesis 1:2 Alternatively, based on previous positive correlation between preference for solitude and general loneliness (Burger, 1995), preference for solitude ~~might~~ would be positively correlated with ill-being ~~in follow-ups~~. This ~~is~~ was based on the idea that preference for solitude is a symptom of psychological vulnerability (i.e., chronic loneliness), ~~this will manifest~~ manifesting as negative ~~experiences~~ emotions during time in isolation.

Formatted: Font: Italic

Hypothesis Set 2. Research ~~suggested~~ suggests that autonomous, and less controlled, motivations for solitude shapes ill-being during solitude experiences (e.g., Thomas & Azmitia, 2019). We ~~focused~~ ed on one form of autonomous motivation (identified motivation, or acting because of the value and importance of the activity) and one form of controlled motivation (external motivation, or acting because one feels externally compelled and choiceless). We ~~predicted~~ ed that identified and external motivation for solitude will predict ill-being one and two weeks later. Importantly, we ~~anticipated~~ ed these relations will be in evidence above and beyond Time 1 preference for solitude, as well as Time 1 ill-being, and controlling for covariates listed above.

Hypothesis 2:1 Identified motivation for solitude ~~will~~ would negatively correlate with residual change in ill-being 1 and 2 weeks later, above and beyond Time 1 preference for solitude.

Formatted: Font: Italic

Hypothesis 2:2 External motivation for solitude ~~will~~ would positively correlate with residual change in ill-being 1 and 2 weeks later, above and beyond Time 1 preference for solitude.

Formatted: Font: Italic

Competing Hypothesis Set 3. It has been suggested, but not tested, that “healthy” motivation for solitude (~~in this study, expected to be~~ identified motivation and less external motivation) reflects adaptive self-regulatory capacity that might moderate the effects of preference for solitude (Nguyen et al., under review). In other words, healthy motivation for solitude might explain when preference for solitude will or will not relate to ill-being. We ~~will~~ tested three competing hypotheses involving moderation effects of preference X identified motivation.

Hypothesis 3:1 Following Hypothesis 1:1, if preference for solitude yields a negative correlation with ill-being, this ~~will~~ would only be the case when identified motivation for solitude is high (estimated at 1 standard deviation (*SD*) above mean) relative to when identified motivation for solitude is low (-1 *SD*). In other words, we ~~expected~~ expected to see a *stronger* negative correlation between preference for solitude and residual change in ill-being from Time 1 to 2 and Time 1 to 3 at $+1$ *SD* level of identified motivation, relative to the negative correlation between preference for solitude and residual change in ill-being across one and two weeks at -1 *SD* level of identified motivation.

Hypothesis 3:2 Following Hypothesis 1:2, if preference for solitude yields a positive correlation with ill-being, this ~~will~~ would only be the case when identified motivation for solitude is low (-1 *SD*) relative to when identified motivation for solitude is high ($+1$ *SD*). In other words, we ~~expected~~ expected to see a *stronger* positive correlation between preference for solitude and residual change in ill-being from Time 1 to 2 and Time 1 to 3 at -1 *SD* level of identified motivation, relative to the positive correlation between preference for solitude and residual change in ill-being across one and two weeks at $+1$ *SD* level of identified motivation.

Formatted: Font: Italic

Formatted: Font: Italic

Hypothesis 3:3 It may also be the case that motivation distinguishes healthy versus unhealthy preference for solitude. Burger (1995) suggested that some people prefer alone time because of shyness and anxiety, while others might prefer being alone because of their enjoyment with solitude. ~~To test this, we~~ we therefore predicted that preference for solitude will yield a positive correlation with ill-being for those at -1 *SD* level of identified motivation for solitude. On the other hand, preference for solitude ~~will~~ would yield negative correlation with ill-being for those at +1 *SD* level of identified motivation for solitude. For this hypothesis, we ~~are~~ were less interested in the magnitude of difference between the two correlation coefficients, but expected to see *opposite* directions of relationship between preference for solitude with residual change in ill-being across one and two weeks at different levels (+1 *SD* vs. -1 *SD*) of identified motivation.

Formatted: Font: Italic

Competing Hypothesis Set 4. Finally, Storr (1988) proposed that only when solitude is forced (external motivation) does it become a detriment to ~~our~~ psychological health. Therefore, it might be the unhealthy motivation for solitude that explains when the preference for it will relate to ill-being. ~~To test this, we~~ We outlined three competing hypotheses involving moderation effects of preference X external motivation:

Hypothesis 4:1 Following Hypothesis 1:1, if preference for solitude yields a negative correlation with ill-being, this ~~will~~ would only be the case when external motivation for solitude is low (-1 *SD*) relative to when external motivation for solitude is high (+1 *SD*). In other words, we expected to see a *stronger* negative correlation between preference for solitude and residual change in ill-being across one ~~week and two weeks~~ week and two weeks at -1 *SD* level of external motivation, relative to the negative correlation between preference for solitude and residual change in ill-being across one ~~week and two weeks~~ week and two weeks at +1 *SD* level of external motivation.

Formatted: Font: Italic

Hypothesis 4:2 Following Hypothesis 1:2, if preference for solitude yields a positive correlation with ill-being, this ~~will~~would only be the case when external motivation for solitude is high (+1 *SD*) relative to when external motivation for solitude is low (-1 *SD*). In other words, we expected to see a *stronger* positive correlation between preference for solitude and residual change in ill-being across one and two weeks at +1 *SD* level of external motivation, relative to the positive correlation between preference for solitude and residual change in ill-being across one and two weeks at -1 *SD* level of external motivation.

Formatted: Font: Italic

Hypothesis 4:3 ~~Again, s~~Similar to Hypothesis 3:3, It may be the case that motivation distinguishes healthy ~~versus~~- unhealthy preference for solitude. ~~To test this, w~~We therefore predicted that preference for solitude will yield a positive correlation with ill-being for those at +1 *SD* level of external motivation for solitude. On the other hand, preference for solitude will yield negative correlation with ill-being for those at -1 *SD* level of external motivation for solitude. For this hypothesis, we ~~are~~were less interested in the magnitude of difference between the two correlation coefficients, but expected to see *opposite* directions of the relationship between preference for solitude with residual change in ill-being across one and two weeks at different levels (+1 *SD* vs. -1 *SD*) of external motivation.

Formatted: Font: Italic

Method2) Describe the key independent and dependent variable(s), specifying how they will be measured.

Formatted: Centered

Participants and Procedure

Formatted: Indent: First line: 0 pi

Recruiting strategy. ~~Time 1 p~~Participants were recruited via Prolific.co. Participants living in the UK or US aged 35+ years who had reported living alone could take part were recruited via Prolific.co. To ensure a good spread across ages, 200 slots were specifically set

aside for those aged 65+ years or older. The geographic restriction was included so that the study captured a sample at the start of the national push for self-isolation (which was set by the U.K. government two days later, and the U.S. days after that; this is as compared to countries such as China and Italy, which had already enforced strict restrictions on movement). We furthermore selected for individuals living alone so that their self-isolation would reflect time spent in solitude (physically alone and not interacting with others). We anticipated that those who were self-isolating *with others* might have less, if not similar, amounts of solitude as they typically would, since they were confined to a shared space together with their cohabitants. Finally, the inclusion criterion of 35+ years of age was set because this age represents a transition to adulthood, rather than young adulthood (Coleman, 1972). The focus on adults was important for two reasons: First, since COVID-19 targets-is harmful to adults and older adults at higher rates, we anticipated that self-isolation is more relevant to, and more likely in, these individuals as compared to young or emerging adults, typically considered under 35 years (Coleman, 1972). Second, since risky behaviour is higher in adolescents to young adults under age 35 years (Bingham & Shope, 2004; Holliday, 1988; Mata et al., 2011), we sought a sample that would make more responsible and consistent health decisions to self-isolate regardless of their personal preferences.

Data termination rule. Our aim was to recruit 800 people in the initial, Time 1, sample recruited on Prolific (~~though we received 29 additional responses before submitting the Stage 1 report~~). Participants were invited to take part only if they: 1) report living alone in a Prolific prescreen, are 35+ years or older, and live in the UK or US. There were also additional participants recruited through social media platforms like Facebook and Twitter; with this group, we stopped data collection at the time of this submission (March 19, 2020).

We aimed for power (95%, alpha level .05) to detect an effect size as small as $f^2 = .016$ for additional variance explained by the main effects of preference and motivation and interaction of these two variables, above and beyond covariates. Following Stage 1 review, covariates were: gender, age, follow-up in-person social contact, follow-up virtual contact (average of chat, text, and phone frequency), subjective health at Time 1, virus diagnosis or suspicion at follow-up (Y/N), health anxiety at follow-up, and the stressor checklist at follow-up). In G*power, we performed sensitivity test to estimate increase in variance explained by each of our predictors of interest (main effects of preference and identified or external motivation, and their interaction) out of a total of 11 variables entered into the regression model (2 main effects, 1 interaction, 8 covariates). This conservative effect size $f^2 = .016$ represents approximately 1.6% of variance explained by each of the main effects and interaction, after removing the variance explained by covariates. However, if we assume up to 50% data loss due to attrition across the two time-points, with $n = 400$, we can expected to have 95% power to detect an effect size as small as $f^2 = .033$.

Participants characteristics and data collection. We received responses from 823 adults living in the UK and US through the online platform Prolific.co and these comprise the final Time 1 sample¹. Of these, 457 (55.46%) were women, 363 (44.05%) were male, and 2 (0.20%) reported another gender (1 did not respond). Their ages ranged from 19 to 87 years ($M = 52.93$, $SD = 12.13$). Although we selected only for individuals stating they are living alone, some

¹ We had two kinds of additional responses. First, we received duplicate and empty responses that our online software, Qualtrics, recorded but we excluded from all analyses. Second, we also shared the survey on social media platforms (i.e., Twitter, Facebook) who met one prescreen criterion: (1) Residing in a single person household (living alone). At the time of submission of this proposal, 18 people have filled out the survey. Because the number achieved through social media was too small for independent analyses and the criteria and recruiting method were different, we will exclude these participants from analyses.

Formatted: Font: Italic

participants reported marital status that implied they are living together: married $n = 29$ (3.5%) and living together as married $n = 23$ (2.8%), though the majority were single or never married 434 (52.67%), divorced 173 (21%); separated 28 (3.40%); or widowed 61 (7.40%). Although the latter group reflected participants who should not have been eligible to take part in the study, they comprised a small part of the sample and we retained them because we had not set this as an exclusion criterion. Their employment status also varied, with 323 (39.20%) reporting full-time work; 99 (12.01%) part-time; 120 (14.56%) self-employment, and 178 (21.60%) as-reporting they are retired. Finally, they varied by the location of their residence in a large city, 212 (25.73%), small city 257 (31.19%), rural area 105 (12.74%) and suburban area 249 (30.22%).

Guided by feedback received during Stage 1 review, the research team contacted participants for Time 2 of the project on March 22, 2020, just over one week after Time 1 survey was launched (March 14, 2020), and for Time 3 on March 29, 2020, one week after Time 2. All participants who took part in Time 1 were contacted and given a chance to take part in Time 2 and 3². We received ethical approval from Durham University's Ethics Committee (PSYCH-2020-03-11T23_41_08). Participants provided informed consent before taking part, received an in-part debrief following Wave 1 of the research, and were invited to skip items or withdraw from the study at any time.

Background Measures

~~Measures-~~The full set of measures collected for all time-points are available at [OSF.IO/MN8KXhttps://osf.io/mn8kx/?view_only=bea3f190b56847d8994f2fe886def0ba](https://osf.io/mn8kx/?view_only=bea3f190b56847d8994f2fe886def0ba).
~~Participants reported their gender, age, country of residence, marital status (e.g., married;~~

² It should be noted that although the study was launched on March 14, not all participants completed the Time 1 survey on the day, so some of them might not have the full 1-week gap between Time 1 and Time 2 surveys.

~~divorced; separated; widowed), employment status (e.g., full time; part time; self-employed; retired), and location of residence (large/small city, suburban area). Of these, we plan for gender, age, and subjective health to be entered as covariates, and for all other characteristics of participants will be reported for descriptive purposes.~~

Formatted: Font color: Auto

Subjective health (Time 1; covariate). We assessed subjective health with one item from the World Value Survey (<http://www.worldvaluessurvey.org/wvs.jsp>). Specifically, participants were asked “All in all, how would you describe your state of health these days? Would you say it is...”, and received the options: *Very poor; Poor; Fair; Good; Very good; I’m not sure*.

Frequency of time spent alone (Times 1-3; quality check). At Time 1, participants were asked: “Considering the waking time you have, how much time *per day* do you typically spend *by yourself at home*?”, and reported with options between 0-20+ waking hours. Additionally, participants were asked: “How frequently in the past 2 weeks have you performed your daily activities at home by yourself?”, using a scale with the anchors: “*Most of the day*”, “*A few times a day*”, “*Once a day*”, “*Several times a week*”, “*Once a week*”, and “*Almost never*”. At Times 2, and 3, participants reported on these items, reflecting on the past week instead. See Table 1 for means and correlations of these items and other measurements of the quantity of solitude.

Frequency of time spent with others (Times 1-3; quality check & covariate). Participants also reported on the frequency of social interactions using instructions from Rook (1984). They were given the prompt: “Think about the people with whom you socialize or enjoy conversations on a regular basis. How often do you interact with these people in general?”, which was followed by four items specifying: In-person (face-to-face), Online (social media),

Phone, or Text (written messages). These items were paired with a Likert-type scale including (1 = hourly or several times a day, 2 = once a day, 3 = every couple of days, 4 = once a week, 5 = less than once per week, 6 = not at all). Responses to all items were reverse-coded so that higher scores reflected higher levels of interactions. Then, items for online, phone, and text messages were averaged to create a single “virtual interactions” score ($\alpha_{\text{time 1}} = .56$; $\alpha_{\text{time 2}} = .55$; $\alpha_{\text{time 3}} = .55$).

Self-Isolation (Times 1-3; quality check). Participants responded to items: “In the past week, did you self-isolate in response to the coronavirus outbreak (COVID-19)?” with options including “no”, “somewhat / in part”, and “yes”. At time 1, the percentage of participants who endorsed the “somewhat / in part” or “yes” responses was: 48%, at Time 2: 85%, and at Time 3, 88%. If responding “somewhat/in part” or “yes”, they were asked: “In the past week, what percentage of your time spent at home alone was deliberate self-isolation (keeping away from other people and places) in response to the coronavirus outbreak (COVID-19)?”; see Table 1 for means, standard deviations, and correlations with other measurements of solitude.

Ill-Being Measures Time 1 (baseline Times 1-3)

Ill-being composite. An ill-being composite will be was constructed by averaging the scales of ill-being described below (composite $\alpha_{\text{time 1}} = .94$; $\alpha_{\text{time 2}} = .93$; $\alpha_{\text{time 3}} = .93$). All measured at Time 1 were paired with the prompt: “In the past 6 months, I have felt...”, and at Times 2 and 3 with the prompt: “In the past week, when I was home alone and not interacting with anyone in-person or virtually, I have felt...”. This composite will serve as a covariate in our main analyses.

Loneliness. Participants reported on the Depletion and Isolation subscales of the Loneliness Rating Scale (Scalise, Ginter, & Gerstein, 1984), and specifically on its frequency,

since frequency of loneliness is more influential than its intensity (Diener, Sandvik, & Pavot, 2009). They were first provided the prompt “In the past 6 months, I have felt...” They, and responded on a scale of 0 (*never*) to 6 (*always*). Depletion items included “empty”, “secluded”, “alienated”, “withdrawn”, and “numb” ($\alpha_{\text{time 1}} = .94$; $\alpha_{\text{time 2}} = .95$; $\alpha_{\text{time 3}} = .95$). Isolation items included “unloved”, “worthless”, “hopeless”, “abandoned”, and “deserted” ($\alpha_{\text{time 1}} = .96$; $\alpha_{\text{time 2}} = .96$; $\alpha_{\text{time 3}} = .96$).

Depression. The shortened 10-item version of the CES-D (Andersen, Byers, Friary, Kosloski, & Montgomery, 2013) was used to measure depression. Participants reported how often: *in the past 6 months* “I felt depressed”, “I felt everything I did was an effort”, and “I felt fearful”, with a scale ranging from 1 (*not at all*) to 6 (*most of the time*) ($\alpha_{\text{time 1}} = .91$; $\alpha_{\text{time 2}} = .92$; $\alpha_{\text{time 3}} = .92$).

Anxiety. Anxiety was measured with a validated 10-item version of the State Trait Anxiety Inventory (STAI; Tluczek, Henriques, & Brown, 2009). Items testing anxiety *over the past 6 months* included: “I felt calm”, “I felt tense”, and “I was worried”, paired with a scale ranging from 1 (*not at all*) to 7 (*very much so*) ($\alpha_{\text{time 1}} = .93$; $\alpha_{\text{time 2}} = .92$; $\alpha_{\text{time 3}} = .93$).

Predictor Variables Time 1 (baseline)

Preference for solitude. We took six items from the 12-item measure Preference for Solitude scale (Burger, 1995). We chose items that were generally worded so that they apply to most people, whereas items that refer to specific circumstances were not included (e.g., “One feature I look for in a job is the opportunity to spend time by myself”). We drew two items from each of the three factors identified by Cramer and Lake (1998) to cover all dimensions of this construct. Instructions were as follows: “For each of the following pairs of statements, select the one that best describes you. In some cases, neither statements may describe you well or both may

describe you somewhat. In those cases, please select the statement that best describes you or that describes you more often.” Respondents selected one of two options per item, e.g., “I enjoy being around people”, or “I enjoy being by myself” ($\alpha = .72$).

Motivation for solitude. Identified (autonomous) and external (controlled) motivations for solitude were measured with a 10-item scale adapted from Nguyen and colleagues (2018). Participants were asked: “Think of times when you will be by yourself, at home, in the next 2 weeks. Those are times when you do not interact with anyone in person or virtually”, followed by items measuring autonomous motivation through its form most applicable to the context of self-isolation, identified motivation (finding value and importance in the activity; Ryan & Deci, 2017). Five items for this subscale included “Having time to myself is important and beneficial to me” (original item), and “Spending time alone is important for protecting my health” (new item). Overall internal reliability was acceptable ($\alpha = .73$). Participants also responded to five items measuring controlled motivation through its form most applicable to the context of self-isolation, external motivation (feeling external pressures and choicelessness), with items such as “I am forced into it” (original item), “I would get in trouble with others if I didn’t” (original item), and “I feel I have no choice” (new item). Items were paired with a 7-point scale ranging from 1 (*this does not apply to me at all*) to 7 (*this applies to me very much*). Overall internal reliability was high ($\alpha = .84$).

Descriptive and Covariate Measures Time 2 and 3 Following Stage-1 Review

New Measures for Time 2

COVID-19 concerns. Two items ~~will~~assessed COVID-19 health concerns, first participants ~~will respond~~responded to the items: “I have been diagnosed with COVID-19” (Yes / No). “I strongly suspect that I currently have a COVID-19 infection” (Yes/No). Participants who

reported “Yes” to either of these items ~~will be~~ considered as having serious concerns about COVID-19.

Health anxiety. Typically a disposition-level measure, participants ~~will~~ also responded to five items of the Health Anxiety Inventory (Salkovskis et al., 2002) that measure health anxiety at a state level³. Participants selected from 4 ~~four~~ options reflecting increasing concerns. For example one item is: I have not worried about my health” “–I have occasionally worried about my health” “–I have spent much of my time worrying about my health” and “I have spent most of my time worrying about my health”- ($\alpha_{\text{time 2}} = .87$; $\alpha_{\text{time 3}} = .87$).

Stressors checklist. Participants ~~are were~~ asked: “In the past week, did you experience any of the following? (tick all that apply)”, and selected the options: “Lost your job”, “Lost a substantial amount of earnings from your job”, “Were worried that COVID-19 would impact your job security”, “Did not have enough food”, “Could not access food”, “Could not access important medical supplies”, “Knew someone who was diagnosed with COVID-19”, or “Were diagnosed with any new health condition”. See Table 2 for frequencies of each stressor experienced at Time 1 and 2.

Overall depression. We used the same measure described above for testing depression in solitude to also test depression overall, with the instructions to participants to reflect on how they felt generally (that is presumably, both within and outside of solitude) in the past week, at Times 2 (\$\alpha = .91\$ ) and 3 (\$\alpha = .92\$ ). Following Stage 1 review, this measure was intended to correlate against our ill-being in solitude measures and provide an exploratory test of the correlations between ill-being in solitude, and ill-being in general.

³ Originally, we picked out six items from the Short Health Anxiety Inventory (SHAI). However, after we launched the Time 2 survey, we caught one erroneous item that said “I noticed aches and pains less than/more than usual”. We excluded this item from the composite, so the final SHAI measure has 5 instead of 6 items.

Results

Effect Size Estimates and Interpretation

Informed by related solitude work described below, we set criteria for smallest effect sizes of interest in our hypotheses to predetermine standards for what we will decide as meaningful when evaluating the results instead of relying on p values. We ~~will use~~ $.15$ *partial r 's* as the smallest effect size of interest (equivalent to .023% of variance explained) to determine whether main effect hypotheses (H1:1, H1:2, H2:1, H2:2) are supported. For moderation effects hypothesised in H3:1, H3:2, H4:1, H4:2, when calculating the relative difference between two correlation coefficients, we ~~will use~~ $.10$ *difference between the two standardized coefficients* (for example, between $-.40$ and $-.30$, or $-.20$ and $-.10$) as the smallest effect size of interest to determine whether this hypothesis is supported.

Rationale for choosing partial r around $.15$. A previous paper by Nguyen, Werner, and Soenens (2019); looked at the link between autonomous motivation (conceptually similar to high identified motivation and low external motivation) and well-being and ill-being variables. In Study 1, sampling first-year university students ($n = 147$), the authors measured motivation for spending time alone and loneliness experienced in the past 2 weeks (not specifically related to time spent alone) and assessed these variables three times. On average, the correlation between autonomous motivation and loneliness was approximately $r = -.30$. Extraversion - a concept similar to preference for solitude - also correlated with loneliness at also around $r = -.30$. However, because these correlations were obtained from a rather small sample, effect sizes were likely overestimated (Yartoki, 2009). As such, we ~~decide~~ ~~selected~~ ~~to go with~~ *place our benchmark at $\frac{1}{2}$ a partial r half the size of those correlations obtained by past research.*

Rationale for choosing .10 difference between regression coefficients. In Study 2 of Nguyen et al. (2019), the researchers also examined the interaction between participants' perceived belonging with autonomous motivation predicting levels of loneliness experienced in the past one month; this moderation test is the closest available data to our hypothesized hypothesised moderation effects (Hypothesis sets 3 and 4) and yielded a coefficient of, $\beta = -.32$, $CI_{95\%} = [-0.44, -0.20]$, when predicting loneliness. At higher levels of autonomous motivation, perceived belonging related to lower loneliness, $\beta = -0.04$, $CI_{95\%} = [-0.21, 0.13]$. Again, findings relied on a sample of $n = 223$, so we expect the effect sizes were likely overestimated. So if we go with Going with Selecting regression coefficients half the size of those coefficients obtained by past research, we ~~might anticipated~~, $r = -0.16$ at low levels of autonomous motivation, and, $r = -0.02$ at high levels. The difference between these two coefficients was about, $\Delta r = .14$, but a conservative expectation is, $\Delta r = .10$.

Exclusion Criteria

~~We have set our most critical inclusion criterion for taking part in the study, namely living alone. We also~~ We have set three exclusion criteria to guide which participants would be excluded from analyses: First, ~~We we will also test~~ tested for inattention with two items "Choose "somewhat agree"" and "Choose "Very true"" embedded in two different scales of Times 2 and 3. We excluded participants who ~~either~~ 1) failed to select the corresponding answers as instructed, ~~and or~~ 2) ~~fail~~ failed at least one attention check item (if the participant misses the item, that is considered failing the attention check). Second, we ~~anticipated we would~~ will exclude participants who responded after we submitted the stage 1 Registered Report, ~~but~~ when we downloaded the data, we saw those were duplicates or missing responses. Procedure for removing duplicates was included in our analysis script (OSF.IO/MN8KXlink). Finally, we will

~~exclude~~ participants if we ~~cannot~~ could not link their data across at least two time-points from the three that will ultimately be collected.

Confirmatory Quality Checks

This study does not involve an experimental manipulation, and for this reason we did not include a manipulation check. However, the assumption underlying our hypotheses is that our sample, on the whole, will find themselves spending more time alone than usual as a function of the COVID-19 outbreak. As such, we conducted three texts-analyses (based on measures described as quality indicators in the Materials description above) to test our belief that any changes in ill-being from Time 1 to Time 2 are due to the effects of social isolation:

Time spent alone. We expected significant (using the $p < .05$ cutoff) increases in frequency (i.e., “How frequently have you performed your daily activities at home by yourself?”) and length (i.e., “Considering the waking time you have had in the past week, how much time per day did you typically spend at home alone?”) of time spent alone from Time 1 to Time 2 and Time 1 to Time 3. Results of two paired samples t -tests showed that on average participants performed daily activities by themselves at home significantly less frequently at Time 2 ($M = 5.23$, $SD = 1.29$) as compared to Time 1 ($M = 5.35$, $SD = 1.13$; $t(703) = 2.37$, $p = 0.018$). The difference between Time 1 and Time 3 ($M = 5.28$, $SD = 1.32$) did not reach statistical significance ($t(701) = 1.39$, $p = .164$). This pattern was not as we expected.

On the other hand, participants reported spending increased numbers of waking hours alone from Time 1 ($M = 10.26$, $SD = 5.34$) to Time 2 ($M = 11.52$, $SD = 5.54$; $t(748) = -6.18$, $p < .001$), and from Time 1 to Time 3 ($M = 12.14$, $SD = 5.25$; $t(751) = -8.71$, $p < .001$). As we expected, on average people reported spending more hours at home alone in the second and third assessment.

In-person interactions. We expected a significant (at $p < .05$) drop in *in-person* interactions from Time 1 to Time 2 and Time 1 to Time 3. Results showed that, when compared to Time 1 ($M = 4.04, SD = 1.43$), people reported having in-person interactions less frequently at Time 2 ($M = 2.78, SD = 1.73; t(749) = 19.43, p < .001$) and Time 3 ($M = 2.16, SD = 1.60; t(751) = 27.36, p < .001$).

Self-isolation questions. We expected a statistically significant (at $p < .05$) increase in the percentage of people endorsing “yes” and “in part” to isolating because of COVID-19, and an increase in the percentage of time spent alone from Time 1 to Time 2 and Time 1 to Time 3. Results showed that significantly higher percentages of participants endorsing “yes” and “in part” to isolating because of COVID-19 at Time 2 (85%; $t(735) = -19.52, p < .001$) and Time 3 (88%; $t(735) = -21.54, p < .001$) as compared to Time 1 (48%). Further, among those who answer “yes” and “in part”, these people also reported spending larger proportion of time alone at home deliberately in response to COVID-19 at Time 2 (71%; $t(333) = -15.37, p < .001$) and Time 3 (80%; $t(328) = -17.20, p < .001$) as compared to Time 1.

Overall, out of five quality checks, four showed expected patterns. Participants indeed engaged in less in-person interactions and spent more waking hours alone between Time 1 and the two follow-up time points. In addition, more participants reported self-isolating in response to COVID-19 at Time 2 and Time 3 than at Time 1, and those who self-isolated also spent higher percentages of time alone self-isolating in response to COVID-19.

Confirmatory Analyses

Correlations. Table 3 presents the means, standard deviations, and planned Pearson correlations of our three main predictors and specific ill-being variables at all time-points. We tested unadjusted links between preference, identified motivation, and external motivation to ill-

being at Time 2 and Time 3. Those who preferred to be alone reported more identified motivation ($r = .39$), and less external motivation ($r = -.13$) for solitude, suggesting that there is some overlapping between these variables but only to a small degree. With 15% shared variance between preference and identified motivation and less than 2% shared variance between preference and external motivation. This indicated that preference and motivation are distinct reasons for seeking solitude. Further, no relations were evident of preference with ill-being constructs at any of the time-points, and weak and mixed relations were evident of identified motivation with ill-being at Times 1-3. The most consistent correlations were observed between external motivation and ill-being, which averaged $r = .26$ across ill-being indicators and time-points (see Table 3). For readers' interest, Table 4 also presents correlations between our three main predictors and ill-being, our outcome of interest, along with study covariates.

Primary models. As planned, primary analyses regressed the composite of our outcomes (loneliness: depletion; loneliness: isolation; depression; anxiety) at Time 2 and Time 3, controlling for the Time 1 ill-being composite. Main models testing our hypotheses accounted for eight potentially confounding effects: gender, age, follow-up in-person social contact, follow-up virtual contact (average of chat, text, and phone frequency), virus diagnosis or suspicion (Y/N), health anxiety, subjective health at Time 1, and the stressor checklist at follow-up.

For each model, we first plotted the distribution of the residual errors to check for normality. All plots indicated that residual errors were normally distributed, and therefore no transformation was needed. We tested our hypotheses using linear regression models as assumption of normality was satisfied.

Hypothesis 1:1, 1:2, 2:1, and 2:2 (concerning main effects of preference and motivations on ill-being). In two hierarchical linear regression analysis, we will

regress/regressed Time 2 and Time 3 ill-being on covariates (listed above) at Step 1, Time 1 preference for solitude at Step 2, and Time 1 identified and external motivation at Step 3.

Findings are summarised in Table 5 and showed that preference and both types of motivation for solitude explained little variance in residual change in ill-being from Time 1 to Time 2, as well as from Time 1 to Time 3. External motivation for time alone predicted an increase in Time 2 ill-being ($\beta = .06$, CI 95% [.01, .10], $p = .010$), yielding a *partial r* of .10, which is smaller than the effect size of interest we determined a priori. Likewise, preference for solitude predicted a decrease in Time 3 ill-being ($\beta = -.06$, CI 95% [-.10, -.1], $p = .015$), yielding a *partial r* of .10, which is also smaller than the effect size of interest.

Hypothesis 3:1, Hypothesis 3:2 & Hypothesis 3:3 (concerning interacting effects with identified motivation). In two hierarchical linear regression analyses, we will regress/regressed follow-up (Time 2 and 3) ill-being on covariates at Step 1, Time 1 preference for solitude and Time 1 identified motivation at Step 2, and the interaction term of preference for solitude and identified motivation for solitude at Step 3. Findings are summarised in Table 6 and did not show evidence of interaction between preference for solitude and identified motivation for time alone for both Time 2 and Time 3 analyses.

Hypothesis 4:1, Hypothesis 4:2 & Hypothesis 4:3 (concerning interacting effects with external motivation). In two hierarchical linear regression analyses, we will regress/regressed follow-up (Time 2 and 3) ill-being on covariates at Step 1, Time 1 preference for solitude and Time 1 external motivation at Step 2, and the interaction term of preference for solitude and external motivation for solitude at Step 3. As summarised in Table 7, we also did not find evidence supporting our hypothesis that there is an interaction between preference for solitude and identified motivation for time alone for both Time 2 and Time 3 analyses.

Summary of confirmatory analyses. Planned unadjusted correlations showed a consistent link between external motivation and higher ill-being as measured through loneliness-depletion, loneliness-isolation, depression, and anxiety at all time-points. However, after accounting for controls including Time 1 ill-being, we found little evidence suggesting that initial preference for solitude and motivation behind spending time alone were predictive of change in ill-being across a period of two weeks. Instead, we found that health anxiety and life stressors, measured simultaneously with the outcome variables, linked strongly with increases in ill-being at Time 2 and Time 3. To the extent that individuals felt anxiety surrounding their health, they reported increased ill-being in their solitude reported at Time 2 and Time 3. Further, with more stressors experienced in the past week (of which job and food insecurity were most predominant, see Table 2), participants reported more ill-being during their solitude across two weeks. Only health anxiety, however, yielded *partial r* = .47, which is larger than our smallest effect size of interest, whereas life stressors yielded an independent effect size below our predetermined threshold, *partial r* = .14. Other characteristics like age and gender, subjective health, frequency of in-person interactions or virtual interactions in the past week also did not independently predict Time 2 and Time 3 ill-being in our fully adjusted models.

Exploratory Analyses

We conducted additional exploratory – unplanned – tests to better understand how change in ill-being occurred across the two weeks of the study and what this means for our three main predictors.

Change in ill-being across early weeks of self-isolation. First, we were interested in the extent participants experienced change in ill-being over the course of two weeks, over and above their pre-existing levels of ill-being. We sought to test, in part, public concerns that self-isolation

in itself would yield loneliness and depression (e.g., CNBC.com, 2020; Guadiran.com, 2020; Health.com, 2020; NewYorker, 2020). Our sample on one hand may more easily access online resources since they participated in an online study, but on the other hand they represented older adults that have been the focus of primary concern (average age app. $M = 52$ years), who are living alone – presumably both of these were risk factors for more ill-being.

Paired sample t-tests comparing Time 1 ill-being with Time 2 and Time 3 ill-being suggested that levels of ill-being over two weeks did not differ from participants' prior ill-being levels. At the beginning of the study, the average level of ill-being experienced in the past 6 months was, $M = 2.75$, $SD = 1.19$. One week after, the average ill-being was no different, ($M = 2.74$, $SD = 1.24$ ($t(749) = .43$, $p = 0.733$), and two weeks after, the average ill-being was no different ($M = 2.75$, $SD = 1.28$ ($t(751) = 0.16$, $p = 0.728$). When we coded for participants who scored higher than mid-point (4 on 7-point scale), we found that 16% fell under this category at Time 1, 17% at Time 2, and 18% at Time 3. In the discussion section, we elaborated on these percentages in relation to recent data reported by a review by Brook et al. (2020) on percentages of people experiencing psychological distress while in quarantine in previous outbreaks (i.e., H1N1, SARS, etc). Convergent evidence suggested that we did not observe expected inclines in ill-being across these two weeks of self-isolation.

Note that in this sample, when we coded for those who reported spending 100% of their day self-isolating due to COVID-19, we found that 6% fell under this category at Time 1, 21% at Time 2, and 34% at Time 3. Further, when we compared those who reported spending 100% of their day at Time 2 and Time 3 self-isolating due to COVID-19, with others who spent a lower percentage of time, ill-being scores between those two groups did not meaningfully differ (see *SOM 1*).

Relations of ill-being in solitude and global depression. Second, following advice in Stage 1 review, we were interested in whether measures of ill-being in *solitude* were distinguished from depressive symptoms, *overall*. That is, we wanted to know whether ill-being in solitude can be distinguished from global ill-being, occurring in the same timeframe. Ill-being at Time 2 correlated, $r = .88$, with overall depressive symptoms at Time 2, and ill-being at Time 3 correlates $r = .90$ with overall depressive symptoms at Time 3. Most notable for us, *depression* in solitude correlated, $r = .93$, and $r = .94$, with depression in general at Times 2 and 3, respectively. These high correlations meant that we do not have evidence that, during this time, reports of ill-being during solitude were different from overall ill-being; the two were, for all intents and purposes, synonymous.

Links of Preference and Motivation to Ill-being. In Pearson correlation analyses between major study variables (reported in Table 4), we observed strong relations between Times 1, 2, and 3 ill-being ranging from $rs = .77 - .89$, and in exploratory analyses we found that not much change occurred in ill-being. We were therefore concerned that the vulnerability model that drove our views that preference and motivation would have predictive value for later ill-being, controlling for earlier ill-being, was unwarranted. In other words, the data did not seem to support a view that self-isolating living-alone adults would have the potential for increases in ill-being, that could be buffered or exacerbated by our predictors.

We therefore conducted two linear regression models similar to the models used to test Hypothesis 1:1, 1:2, 2:1, and 2:2, except that this time we did not control for Time 1 ill-being. By doing this, we simply tested whether preference and motivation measured at Time 1 would be linked with later assessments of ill-being. This analysis should be interpreted with caution, because it reflects, essentially, a series of partial correlations. While one measure precedes the

other, this should not be interpreted as a causal effect. For this reason, although vulnerability due to health concerns, demographics, and actual quantity of solitude was accounted for, many other possible confounds may play a role in inflating these relations. Findings are summarised in SOM 1.

A number of meaningfully significant links were in evidence in these models. First, in this model older age related to *lower* ill-being, $pr = -.17$, and the relation of stressors slightly increased and met our planned effect size of interest, $pr = .17$. As before, the relation of health anxiety with ill-being measured at the same time-point was by far the most robust one identified, $pr = .51$. Accounting for covariates, external motivation linked to more ill-being at Time 2, $pr = .22$, and at Time 3, $pr = .17$. In summary, these analyses show non-causal links over the short-term that may not extend across longer periods.

Discussion

Confirmatory results did not support our hypotheses regarding the role of preference and motivation for solitude on ill-being across time. The links between preference and motivation for solitude, and later ill-being at Time 2 and Time 3, which controlled for Time 1 ill-being along with a number of vulnerability factors, did not meet the threshold we had set for the minimum effect size of interest. Our minimum effect size of interest was set based on daily experiences of solitude in studies of undergraduate students (Nguyen, Werner, & Soenens, 2019). Although we had assumed a correlation half the size of the one identified in this literature ($r = .30$; Nguyen, Werner, & Soenens, 2019), the effect observed for external motivation was even lower, $r = .09$, than our minimum effect of interest, $r = .15$ set at the outset.

Similarly, the effect of preference was, $r = .04$ at Week 2, and $r = .06$ at Week 3. Thus, observed increases in ill-being for those who feel pressured to self-isolate, and those who prefer

to spend time alone more than with others, are very small. Further, these two variables did not interact to predict change in ill-being over the course of two weeks. The conclusion of this finding is that, even when individuals might prefer time alone or endorsed the importance of self-isolating, for example for their own or others' benefit, this study did not reveal evidence supporting that it ameliorates or exacerbates their loneliness, depression, or anxiety during time spent alone. Even with additional exploratory analyses to test relations of both preference and motivation to ill-being at each of the two follow-up time-points, we did not observe an effect above our minimum effect of interest, except in the case of external motivation predicting higher ill-being one week later.

We see three possibilities for why we observed lower effect size than those previously observed, which merit further study. First, it could be argued that any effects of identified and external motivation for solitude do not apply to *extreme* solitude. However, even in studies of forced solitude in the extreme, such as solitary confinement in prisons where choicelessness and external motivation are extremely high, motivation is linked to negative mental health consequences for prisoners (Arrigo & Bullock, 2008; Metzner & Fellner, 2013). It is unlikely that our study captured the extreme cases of isolating individuals, and that for this reason preference and motivation played a less important role.

It is worth noting that the sample obtained in this research was not comparable to other cases of isolation due to quarantine. Obligatory quarantine of those who have been exposed to a contagious illness causes great distress to quarantined individuals, but quarantined cases and those who are in voluntary self-isolation like the cases we are studying in our sample are in very different circumstances (Brooks et al., 2020). The levels of distress experienced among those in quarantine are likely to be higher than those in voluntary self-isolation. For example, in a study

looking at experiences of horse owners during quarantine due to an equine influenza outbreak in Australia, 34% of those in quarantine reported distress; this is notably higher than the percentage of those experiencing distress in the Australian general population (Taylor et al., 2008). In another sample of parents impacted by H1N1 and SARS outbreaks, 28% of parents in quarantine reported trauma-related disorder, compared with 6% of parents not in quarantine (Sprang & Silman, 2013). In our sample, as reported in the exploratory tables, 16% of the participants scored higher than mid-point on ill-being (higher than 4 on 7-point range) at Time 1, 16% at Time 2, and 18% at Time 3.

Second, it is possible that there is little change for preference and motivation for solitude to predict. While previous research suggested that prolonged self-isolation would yield psychological consequences, in the current study we did not see evidence of change in ill-being over the course of two weeks of self-isolation. In our sample, which varied in ages from 35 to 87 years, participants across the adulthood spectrum, who reported engaging in less in-person interaction and spending more waking hours alone, felt similarly during this time of self-isolation. These findings speak to concerns regarding older adults during self-isolation. Certainly, older adults, who are more vulnerable to COVID-19 exposure, more highly restricted to time at home, and generally less comfortable with meeting their own needs online, benefit from the help and support of community. However, at least within our sample, we did not find evidence that current circumstances yield depression, loneliness, or anxiety, despite media warnings about negative consequences of rapid move to self-isolating during the period of the study (CNBC.com, 2020; Guadiran.com, 2020; Health.com, 2020; NewYorker, 2020). Thus, concerns that self-isolation is harmful to mental health of adults were unsubstantiated in our

findings. In fact, few of our predictors could affect change in ill-being at all, which remained low through the duration of the study.

Third, this study was conducted amidst precipitous changes in governmental regulations and guidelines, as well as individuals' understanding of and plans for self-isolation, and although preference is seen as a dispositional quality that should not much change (Burger, 1995), motivations may have shifted within the first week of the study. Since we did not measure motivation past Time 1, we cannot answer that question. However, we attempted to acknowledge the quickly changing landscape of COVID-19 by, part-way through the study, moving to weekly data collections following Stage 1 reviewer advice. Two days before Time 2 data collection, the U.K. shut down non-essential public places and residents were asked to stay at home at nearly all times (The Guardian); one day after the U.S. saw state-wide school closures by more than half of states (CNN). Since Time 2 referred primarily to the week before these changes took place, it reflected initial motivations more closely. Time 3 had seen additional extended self-isolation and self-isolation driven by more pressures, such as highly limited access to groceries for those who were home (e.g., The Guardian). Certainly, it is reasonable to assume that the sense of urgency and pressure around self-isolation changed following these decisions, and similarly that perceived importance may have changed. If it is the case, it speaks to the malleability of motivations as a function of communications and circumstances surrounding COVID-19. Thus, a fruitful topic of future research would be to experimentally test motivations to examine the effects of these communications on ill-being.

Interestingly, the effect size for identified motivation changing ill-being was close to 0. Therefore, we did not find evidence that when people recognised the importance of self-isolating, this protected them from ill-being during time spent alone. Future research may

consider other outcomes of identified motivation, for example on behavioural intention (Deci & Ryan, 2008). Providing people recognise the importance of self-isolation, they may be more likely to cooperate and engage in this behaviour (Ryan & Deci, 2017). Yet, our findings suggest that when they do so, they may experience the same mental health costs or benefits within solitude – we see evidence of neither – of this cooperation.

Importantly, even with the little change, the one predictor that robustly predicted change in ill-being was a concern about health, which was measured simultaneously with ill-being dependent variables. Higher health concerns measured at the same time as the dependent variable related to more loneliness, depression, and anxiety. Thus, health anxiety was by far the most compelling vulnerability for individuals in self-isolation, a finding which suggests that practical and emotional support for health needs is particularly important during self-isolation.

Limitations

These findings should be viewed in light of two limitations of this study design. First, as described previously, the study took place during a time of fast-paced changes to circumstances, perceived health concerns, and expectations for the duration of lifestyle restrictions, all of which changed between each time-point collected. On one hand, these changes contribute to the strength and richness of the study, since longitudinal analyses stretched the capacity of our predictors to test their boundary conditions in these unusual circumstances. They furthermore spoke to the resilience of the individuals we sampled – despite all these changes we did not see increases in ill-being that we anticipated. On the other hand, our predictor measures were collected at the start of the study (Time 1) only and we cannot know how preference and motivation for solitude might have changed during this time.

A second major concern is that it is unclear whether we tested ill-being in solitude, or ill-being in general. In fact, depression in solitude correlated ($r = .93$ (time 2) and $.95$ (time 3)) with a general measure of depression we had collected following Stage 1 reviewer recommendations to conduct this test. Understandably, when individuals spend a majority of their waking hours in solitude these two should be nearly identical. However, we cannot be certain whether solitude that defines daily experiences, as it does now when individuals are self-isolation, behaves the same as solitude that is incidental in daily life under normal circumstances. In future research or in meta-analytic work later, it would be insightful to compare the two conditions to understand where they differ and where they share characteristics.

Conclusions

The experience of living-alone individuals self-isolating is not atypical. They represent anywhere from 23.9% to 50% of households in the U.K. and U.S. (Our World in Data, 2019; Office for National Statistics, 2019), the two populations from which we sampled. In this study, we did not find evidence that the mental health of these individuals had been negatively affected by self-isolation in its early weeks, a finding that clearly differentiates this group from those who find themselves in forced quarantine (Brooks et al., 2020). Further, we did not find evidence in support of our planned hypotheses above the minimum effect size of interest. Other studies have similarly failed to evidence change in wellness as a function of motivation in solitude, though they evidenced direct links with concurrent wellness (Nguyen, Werner, & Soenens, 2019). However, we had anticipated that protective or aggravating predictive factors would play a stronger role predicting change because of the unexpected and sudden increases in the time spent alone in response to the COVID-19 outbreak. We would caution researchers to be sensitive to

possible nuances in reactions to COVID-19, including in the context of self-isolation, and to avoid making assumptions and post-hoc interpretations based on studies of forced quarantine.

Author contributions. NW and TVN worked in a fully collaborative fashion on this project. Together, they conceptualised the study, developed methods, ethical documentation, and analytic plan, prepared registrations and study materials, and wrote Stage 1 of the registered report. At Stage 2, they once again collaborated to analyse data and write the full report.

Funding. This work was supported in-part by a grant from the European Research Council [ERC; SOAR; 851890].

References

- Andresen, E. M., Byers, K., Friary, J., Kosloski, K., & Montgomery, R. (2013). Performance of the 10-item Center for Epidemiologic Studies Depression scale for caregiving research. *SAGE open medicine, 1*, 2050312113514576.
- Arrigo, B. A., & Bullock, J. L. (2008). The psychological effects of solitary confinement on prisoners in supermax units: Reviewing what we know and recommending what should change. *International journal of offender therapy and comparative criminology, 52*(6), 622-640.
- Berinsky, A. J., Margolis, M. F., Sances, M. W., & Warshaw, C. (2019). Using screeners to measure respondent attention on self-administered surveys: Which items and how many?. *Political Science Research and Methods, 1*-8.
- Bingham, C. R., & Shope, J. T. (2004). Adolescent developmental antecedents of risky driving among young adults. *Journal of studies on alcohol, 65*(1), 84-94.
- Brooks, S. K., Webster, R. K., Smith, L. E., Woodland, L., Wessely, S., Greenberg, N., & Rubin, G. J. (2020). The psychological impact of quarantine and how to reduce it: rapid review of the evidence. *The Lancet*. [https://doi.org/10.1016/S0140-6736\(20\)30460-8](https://doi.org/10.1016/S0140-6736(20)30460-8).
- Burger, J. M. (1995). Individual differences in preference for solitude. *Journal of Research in Personality, 29*(1), 85-108.
- Cacioppo, J. T., Hawley, L. C., Crawford, L. E., Ernst, J. M., Burleson, M. H., Kowalewski, R. B., ... & Berntson, G. G. (2002). Loneliness and health: Potential mechanisms. *Psychosomatic medicine, 64*(3), 407-417.
- Chambers, D. (2013). *Social media and personal relationships: Online intimacies and networked friendship*. Springer.

- Coleman, J. S. (1972). How do the young become adults?. *Review of Educational Research*, 42(4), 431-439.
- Cramer, K. M., & Lake, R. P. (1998). The Preference for Solitude Scale: Psychometric properties and factor structure. *Personality and Individual Differences*, 24(2), 193-199.
- Cuenya, L., Fosacheca, S., Mustaca, A., & Kamenetzky, G. (2012). Effects of isolation in adulthood on frustration and anxiety. *Behavioural processes*, 90(2), 155-160.
- Franck, L., Deci, E. L., & Ryan, R. M. (2008). Self-determination theory: A macrotheory of human motivation, development, and health. *Canadian psychology/Psychologie canadienne*, 49(3), 182-185.
- Diener, E., Sandvik, E., & Pavot, W. (2009). Happiness is the frequency, not the intensity, of positive versus negative affect. In *Assessing well-being* (pp. 213-231). Springer, Dordrecht.
- Emmons, R. A., & McCullough, M. E. (2003). Counting blessings versus burdens: an experimental investigation of gratitude and subjective well-being in daily life. *Journal of personality and social psychology*, 84(2), 377-389.
- Holliday, S. G. (1988). Risky-choice behavior: A life-span analysis. *The International Journal of Aging and Human Development*, 27(1), 25-33.
- Mata, R., Josef, A. K., Samanez-Larkin, G. R., & Hertwig, R. (2011). Age differences in risky choice: a meta-analysis. *Annals of the New York Academy of Sciences*, 1235, 18-29.
- Metzner, J. L., & Fellner, J. (2013). Solitary confinement and mental illness in US prisons: A challenge for medical ethics. *Health and Human Rights in a Changing World*, 316-323.
- Molyneux, N., & Parkinson, L. (2016). Systematic review of interventions addressing social isolation and depression in aged care clients. *Quality of Life Research*, 25(6), 1395-1407.

- Nguyen, T. V. T., Ryan, R. M., & Deci, E. L. (2018). Solitude as an approach to affective self-regulation. *Personality and Social Psychology Bulletin*, *44*(1), 92-106.
- Nguyen, T. V. T., Weinstein, N., & Ryan, R. M. (Under Review). *Unpacking the “Why” of Time Spent Alone: Who Prefers and Who Chooses it Autonomously*.
- Nguyen, T. V. T., Werner, K. M., & Soenens, B. (2019). Embracing me-time: Motivation for solitude during transition to college. *Motivation and Emotion*, *43*(4), 571-591.
- Office for national statistics (2019, Nov 15). *Families and households in the UK: 2019*.
<https://www.ons.gov.uk/peoplepopulationandcommunity/birthsdeathsandmarriages/families/bulletins/familiesandhouseholds/2019>
- Our World in Data (2019). *Percentage of americans living alone*.
<https://ourworldindata.org/grapher/percentage-of-americans-living-alone-by-age?time=2016&country=Age%2018+Age%2021+Age%2030+Age%2045+Age%2060+Age%2075+Age%2089>
- Rook, K. S. (1990). Stressful aspects of older adults' social relationships: Current theory and research. *Stress and coping in later-life families*, 173-192.
- Ryan, R. M., & Deci, E. L. (2017). *Self-determination theory: Basic psychological needs in motivation, development, and wellness*. NY: Guilford Publications.
- Salkovskis, P. M., Rimes, K. A., Warwick, H. M. C., & Clark, D. M. (2002). The Health Anxiety Inventory: development and validation of scales for the measurement of health anxiety and hypochondriasis. *Psychological medicine*, *32*(5), 843-853.
- Sarason, I. G., Levine, H. M., Basham, R. B., et al. (1983). Assessing social support: The Social Support Questionnaire. *Journal of Personality and Social Psychology*, *44*, 127-139.

- Scalise, J. J., Ginter, E. J., & Gerstein, L. H. (1984). Multidimensional loneliness measure: the loneliness rating scale (LRS). *Journal of Personality Assessment*, 48(5), 525-530.
- Sorrentino, R. M., & Higgins, E. T. E. (1986). *Handbook of motivation and cognition: Foundations of social behavior*. Guilford Press.
- Sprang, G., & Silman, M. (2013). Posttraumatic stress disorder in parents and youth after health-related disasters. *Disaster medicine and public health preparedness*, 7(1), 105-110.
DOI: <https://doi.org/10.1017/dmp.2013.22>
- Storr, A. (1988). *Solitude: A return to the self*. Simon and Schuster.
- Taylor, M. R., Agho, K. E., Stevens, G. J., & Raphael, B. (2008). Factors influencing psychological distress during a disease epidemic: data from Australia's first outbreak of equine influenza. *BMC public health*, 8(1), 347. DOI: <https://doi.org/10.1186/1471-2458-8-347>
- Thomas, V., & Azmitia, M. (2019). Motivation matters: Development and validation of the Motivation for Solitude Scale—Short Form (MSS-SF). *Journal of adolescence*, 70, 33-42.
- Tluczek, A., Henriques, J. B., & Brown, R. L. (2009). Support for the reliability and validity of a six-item state anxiety scale derived from the State-Trait Anxiety Inventory. *Journal of nursing measurement*, 17(1), 19-28.
- Yarkoni, T. (2009). Big correlations in little studies: Inflated fMRI correlations reflect low statistical power—Commentary on Vul et al. (2009). *Perspectives on Psychological Science*, 4(3), 294-298.
- Zimet, G. D., Dahlem, N. W., Zimet, S. G., & Farley, G. K. (1988). The multidimensional scale of perceived social support. *Journal of personality assessment*, 52(1), 30-41.

Appendix D

Motivation and Preference in Isolation: A test of their different influences on responses to self-isolation during the COVID-19 outbreak

Netta Weinstein

University of Reading

Cardiff University

Thuy-vy Nguyen

Durham University

Abstract

This multi-wave study examined the extent that both preference and motivation for time alone shapes ill-being during self-isolation. Those-Individuals in the US and UK are self-isolating in response to the COVID-19 outbreak. Different motivations may drive their self-isolation: some might see value in it (understood as the identified form of autonomous motivation), while others might feel forced into it by authorities or close others (family, friends, neighbourhoods, doctors; the external form of controlled motivation). People who typically prefer company will find themselves spending more time alone, and may experience differential ill-being uniformly, or as a function of their identified or external motivations for self-isolation. Self-isolation therefore offers a unique opportunity to distinguish two constructs coming from disparate literatures. This project examined preference and motivation (identified and external) for solitude, and tested their independent and interacting contributions to ill-being (loneliness, depression, and anxiety during time spent alone) across two weeks. Confirmatory hypotheses regarding preference and motivation were not supported by the data. A statistically significant effect of controlled motivation on change in ill-being was observed one week later, and preference predicted ill-being across two weeks. However, effect sizes for both were below our minimum threshold of interest.

Keywords: *Motivation; Solitude; Self-isolation; COVID-19; Loneliness*

Motivation and Preference in Isolation: A test of their different influences on responses to self-isolation during the COVID-19 outbreak

The purpose of this study was to understand how motivation and preference for solitude - two conceptually distinct constructs from disparate literatures that might have unique or interacting effects on individuals' responses to being in solitude - influence ill-being during self-isolation. To achieve this purpose, we focused on solitude during self-isolation operationalised in terms of both physical and social isolation, arguably its purest form (Nguyen et al., 2018).

When self-isolating, individuals reduce the number of others with whom they have contact by staying at home, but ~~whether it involves~~ their physical isolation (being physically separated from others) and social isolation (being physically separate from, and also not interacting with, others) can vary. Many, including those living with others as well as those living alone, are likely to resort to virtual interactions, which replace in-person interactions as the main source of communication (Chambers, 2013). For those who live with others, time without virtual interactions can still be rich with in-person interactions. For those living alone, time without virtual interactions means truly being in solitude. Because our interest is in the influence of both preference and motivation for time alone on the psychological consequences for adults, we focus on a context when psychological isolation becomes the most significant issue: when living-alone adults do not interact with anyone in person and virtually.

While self-isolating is an effective strategy to “flatten the curve” of coronavirus infections by preventing the risks of contracting the virus, it may have psychological consequences studied in past research. To the extent that self-isolation results in loneliness, it may increase cardiovascular activation and cortisol levels, and reduce sleep, interfering with

health and recovery (Cacioppo et al., 2002). Isolation is furthermore a risk factor for depression and anxiety among other indicators of ill-being (Cuenya, Fosachea, Mustaca, & Kamenetzky, 2012; Franck, Fosachea, Mustaca, & Kamenetzky, 2016). Dispositional preference to be alone, and motivation for self-isolation, may both mitigate the potential psychological costs (Nguyen, Ryan, & Deci, 2018), though little is understood about their independent and interdependent contributions. This study will build our understanding of the roles these constructs play using transparent and robust research methods.

Preference and Motivation for Solitude

Few conclusions can be drawn from the existing literature about how preference for solitude predicts ill-being associated with time spent in isolation. Research shows that people who prefer to be alone (Burger, 1995) are broadly vulnerable and report ill-being, although there are mixed findings concerning their ill-being when they are in solitude, specifically (Nguyen, Weinstein, & Ryan, Under Review). Although no research of which we are aware has investigated this, it is also plausible that preference for solitude would lead individuals to tolerate it better and therefore report *less* ill-being when self-isolating.

A separate literature, self-determination theory (SDT) highlights the importance of motivation driving behaviour (Ryan & Deci, 2017). Motivations are “hot”, energising reasons for behaviour in contrast to the “cold” cognition characterising preference; both meaningfully contribute to behaviour (Sorrentino & Higgins, 1986). Based in SDT, less autonomous or self-driven motivation for solitude, and more controlled motivation reflecting pressure and choicelessness, relate to feeling lonely when alone (Nguyen et al.; Study 4). Further, endorsing autonomous reasons for spending time alone correlates negatively with ill-being outcomes like depression, loneliness, and anxiety (Thomas & Azmitia, 2019). However, these studies have

conflated enjoying being alone with finding value in the activity, because both are theorised to comprise autonomous motivation. One form of autonomous motivation is identified motivation, or selecting to spend time alone because it is seen as being beneficial and valuable (Nguyen et al., 2018). Identified motivation is particularly relevant in the case of self-isolation, since the reason for this behaviour is its importance to the health of the self-isolating individual and/or to society rather than because it is intrinsically enjoyable or rewarding - another aspect of autonomous motivation. Therefore, the extent to which this motivation is endorsed needs to be examined in its own right rather than as part of an autonomy composite. Further, one form of controlled motivation, external motivation or feeling pressured by others and choiceless (Ryan & Deci, 2017), is relevant to instances of self-isolation which are felt to be driven by pressures from societal (government guidelines or mandates) or closer relationship (family, friends, doctors). Further, external motivation is important to understanding the outcomes of solitude: forced solitude is thought to be the most detrimental form of solitude, and perhaps *only* when solitude is forced does it result in ill-being (Storr, 1988). The current study will therefore examine the main effects of both identified (autonomous) and external (controlled) motivations - “hot” energizing reasons to self-isolate, and their moderating effects on the relationship between preference, a “cold” reason, and ill-being.

Present Research

Though preference and motivation for solitude show different patterns of influence on ill-being, those who prefer solitude often also report more autonomous motivation for it (Nguyen et al., 2018; Nguyen et al., 2019; Thomas & Azmitia, 2019). However, self-isolation is an occasion in which one’s longstanding preference to be alone versus in the company of others, and their current motivations to be alone, should be more strongly differentiated. Those who typically

prefer social situations may now find additional value in time spent alone (e.g., for protecting their health), or alternatively they may feel pressured or choiceless around self-isolating. It may be either their preference, or their motivation, driving effects of ill-being during time spent alone. Self-isolation is therefore ideal for gaining a deeper theoretical understanding of these two predictors of ill-being in solitude.

We pre-registered and collected Time 1 of the research ([OSF.IO/MN8KX](https://osf.io/MN8KX)) to minimise the amount of time respondents had spent in self-isolation before the start of the study. Following initial review of the Registered Report, we collected Time 2 just over one week later (March 22) and Time 3 one week after Time 2 (March 29). The verbatim hypothesis, methodological and analytic sections of the Stage 1 manuscript were registered following Stage 1 in-principle acceptance and before Time 3 data collection at (OSF). See also the full Stage 1 accepted manuscript at (OSF), and the originally submitted research plan as a supplementary table.

We had initially planned to collect two, not three time-points, and that those would take place two weeks apart. The move to weekly data collection and three time points was made following reviewers' advice to recognise the quickly changing nature of communications related to COVID-19. For example, the UK government announced school and bar/pub closures to take place by the end of March 20th, 2020 (The Guardian). Furthermore, although the US government had not yet made any decisions to take more stringent social distancing measures, we also expected a shift in motivation and perceived urgency of self-isolation on March 23th, 2020, following earlier recommendation by the CDC to prevent gatherings of more than 50 people, and by-after statewide school closures announced by more than 30 states on that date (CNN). It seems, then, that weekly, not bi-weekly, measurements of ill-being were needed to understand

ill-being before and after these significant changes to people's understanding and expectations for self-isolation.

We set out to test four sets of hypotheses guided by current theory and research on preference and motivation for solitude. Simply, these concerned the potential effects of preference on ill-being at Times 2 and 3 (Hypothesis 1:1, 1:2), of identified and external motivations on ill-being at Times 2 and 3, above and beyond preference (Hypothesis 2:1, 2:2), and the possible moderating effects of identified (Hypothesis 3:1, 3:2; 3:3) and external (Hypothesis 4:1, 4:2; 4:3) motivations on the main effect of preference and ill-being at Times 2 and 3.

Competing Hypothesis Set 1. For the link between preference for solitude and psychological consequences of isolation, we had competing hypotheses as follows:

Hypothesis 1:1 Preference for solitude would yield a negative correlation with ill-being 1 and 2 weeks later when participants report about time spent alone in the past week. This was based on the idea that for individuals with a greater preference for being alone, isolation ~~will~~ would be a more pleasant experience.

Hypothesis 1:2 Alternatively, based on previous positive correlation between preference for solitude and general loneliness (Burger, 1995), preference for solitude would be positively correlated with ill-being. This was based on the idea that preference for solitude is a symptom of psychological vulnerability (i.e., chronic loneliness) manifesting as negative emotions during time in isolation.

Hypothesis Set 2. Research suggests that autonomous, and less controlled, motivations for solitude shapes ill-being during solitude experiences (e.g., Thomas & Azmitia, 2019). We focused on one form of autonomous motivation (identified motivation, or acting because of the

value and importance of the activity) and one form of controlled motivation (external motivation, or acting because one feels externally compelled and choiceless). We predicted that identified and external motivation for solitude ~~will~~would predict ill-being one and two weeks later.

Importantly, we anticipated these relations ~~will~~would be in evidence above and beyond Time 1 preference for solitude, as well as Time 1 ill-being, and controlling for covariates listed above.

Hypothesis 2:1 Identified motivation for solitude would negatively correlate with residual change in ill-being 1 and 2 weeks later, above and beyond Time 1 preference for solitude.

Hypothesis 2:2 External motivation for solitude would positively correlate with residual change in ill-being 1 and 2 weeks later, above and beyond Time 1 preference for solitude.

Competing Hypothesis Set 3. It has been suggested, but not tested, that “healthy” motivation for solitude (identified motivation and less external motivation) reflects adaptive self-regulatory capacity that might moderate the effects of preference for solitude (Nguyen et al., under review). In other words, healthy motivation for solitude might explain when preference for solitude ~~will~~or will~~would~~ not relate to ill-being. We tested three competing hypotheses involving moderation effects of preference X identified motivation.

Hypothesis 3:1 Following Hypothesis 1:1, if preference for solitude yields a negative correlation with ill-being, this would only be the case when identified motivation for solitude is high (estimated at 1 standard deviation (*SD*) above mean) relative to when identified motivation for solitude is low (-1 *SD*). In other words, we expected to see a *stronger* negative correlation between preference for solitude and residual change in ill-being from Time 1 to 2 and Time 1 to 3 at $+1$ *SD* level of identified motivation, relative to the negative correlation between preference

for solitude and residual change in ill-being across one and two weeks at -1 *SD* level of identified motivation.

Hypothesis 3:2 Following Hypothesis 1:2, if preference for solitude yields a positive correlation with ill-being, this would only be the case when identified motivation for solitude is low (-1 *SD*) relative to when identified motivation for solitude is high (+1 *SD*). In other words, we expected to see a *stronger* positive correlation between preference for solitude and residual change in ill-being from Time 1 to 2 and Time 1 to 3 at -1 *SD* level of identified motivation, relative to the positive correlation between preference for solitude and residual change in ill-being across one and two weeks at +1 *SD* level of identified motivation.

Hypothesis 3:3 It may also be the case that motivation distinguishes healthy versus unhealthy preference for solitude. Burger (1995) suggested that some people prefer alone time because of shyness and anxiety, while others might prefer being alone because of their enjoyment with solitude. We therefore predicted that preference for solitude ~~will~~would yield a positive correlation with ill-being for those at -1 *SD* level of identified motivation for solitude. On the other hand, preference for solitude would yield negative correlation with ill-being for those at +1 *SD* level of identified motivation for solitude. For this hypothesis, we were less interested in the magnitude of difference between the two correlation coefficients, but expected to see *opposite* directions of relationship between preference for solitude with residual change in ill-being across one and two weeks at different levels (+1*SD* vs. -1 *SD*) of identified motivation.

Competing Hypothesis Set 4. Finally, Storr (1988) proposed that only when solitude is forced (external motivation) does it become a detriment to psychological health. Therefore, it might be the unhealthy motivation for solitude that explains when the preference for solitude ~~it~~

~~will~~-relates to ill-being. We outlined three competing hypotheses involving moderation effects of preference X external motivation:

Hypothesis 4:1 Following Hypothesis 1:1, if preference for solitude yields a negative correlation with ill-being, this would only be the case when external motivation for solitude is low ($-1 SD$) relative to when external motivation for solitude is high ($+1 SD$). In other words, we expected to see a *stronger* negative correlation between preference for solitude and residual change in ill-being across one and two weeks at $-1 SD$ level of external motivation, relative to the negative correlation between preference for solitude and residual change in ill-being across one and two weeks at $+1 SD$ level of external motivation.

Hypothesis 4:2 Following Hypothesis 1:2, if preference for solitude yields a positive correlation with ill-being, this would only be the case when external motivation for solitude is high ($+1 SD$) relative to when external motivation for solitude is low ($-1 SD$). In other words, we expected to see a *stronger* positive correlation between preference for solitude and residual change in ill-being across one and two weeks at $+1 SD$ level of external motivation, relative to the positive correlation between preference for solitude and residual change in ill-being across one and two weeks at $-1 SD$ level of external motivation.

Hypothesis 4:3 Similar to Hypothesis 3:3, It may be the case that motivation distinguishes healthy versus unhealthy preference for solitude. We therefore predicted that preference for solitude ~~will~~-would yield a positive correlation with ill-being for those at $+1 SD$ level of external motivation for solitude. On the other hand, preference for solitude ~~will~~-would yield negative correlation with ill-being for those at $-1 SD$ level of external motivation for solitude. For this hypothesis, we were less interested in the magnitude of difference between the two correlation coefficients, but expected to see *opposite* directions of the relationship between

preference for solitude with residual change in ill-being across one and two weeks at different levels (+1 *SD* vs. -1 *SD*) of external motivation.

Method

Participants and Procedure

Recruiting strategy. Participants living in the UK or US aged 35+ years who had reported living alone could take part were recruited via Prolific.co. To ensure a good spread across ages, 200 slots were specifically set aside for those aged 65 years or older. The geographic restriction was included so that the study captured a sample at the start of the national push for self-isolation (which was set by the U.K. government two days later, and the U.S. days after that; this is as compared to countries such as China and Italy, which had already enforced strict restrictions on movement). We furthermore selected for individuals living alone so that their self-isolation would reflect time spent in solitude (physically alone and not interacting with others). We anticipated that those who were self-isolating *with others* might have less, if not similar, amounts of solitude as they typically would, since they were confined to a shared space together with their cohabitants. Finally, the inclusion criterion of 35+ years of age was set because this age represents a transition to adulthood, rather than young adulthood (Coleman, 1972). The focus on adults was important for two reasons: First, since COVID-19 is harmful to adults and older adults at higher rates, we anticipated that self-isolation is more relevant to, and more likely in, these individuals as compared to young or emerging adults, typically considered under 35 years (Coleman, 1972). Second, since risky behaviour is higher in adolescents to young adults under age 35 years (Bingham & Shope, 2004; Holliday, 1988; Mata et al., 2011), we sought a sample that would make more responsible and consistent health decisions to self-isolate regardless of their personal preferences.

Data termination rule. Our aim was to recruit 800 people in the initial, Time 1, sample recruited on Prolific. We aimed for power (95%, alpha level .05) to detect an effect size as small as $f^2 = .016$ for additional variance explained by the main effects of preference and motivation and interaction of these two variables, above and beyond covariates. Following Stage 1 review, covariates were: gender, age, follow-up in-person social contact, follow-up virtual contact (average of chat, text, and phone frequency), subjective health at Time 1, virus diagnosis or suspicion at follow-up (Y/N), health anxiety at follow-up, and the stressor checklist at follow-up). In G*power, we performed sensitivity test to estimate increase in variance explained by each of our predictors of interest (main effects of preference and identified or external motivation, and their interaction) out of a total of 11 variables entered into the regression model (2 main effects, 1 interaction, 8 covariates). This conservative effect size $f^2 = .016$ represents approximately 1.6% of variance explained by each of the main effects and interaction, after removing the variance explained by covariates. However, assuming up to 50% data loss due to attrition across the two time-points, with $n = 400$, we expected to have 95% power to detect an effect size as small as $f^2 = .033$.

Participants characteristics and data collection. We received responses from 823 adults living in the UK and US through the online platform Prolific.co and these comprise the final Time 1 sample¹. Of these, 457 (55.46%) were women, 363 (44.05%) were male, and 2 (0.20%) reported another gender (1 did not respond). Their ages ranged from ~~19~~24 to 87 years

¹ We had two kinds of additional responses. First, we received duplicate and empty responses that our online software, Qualtrics, recorded but we excluded from all analyses. Second, we also shared the survey on social media platforms (i.e., Twitter, Facebook) who met one prescreen criterion: (1) Residing in a single person household (living alone). At the time of submission of this proposal, 18 people have filled out the survey. Because the number achieved through social media was too small for independent analyses and the criteria and recruiting method were different, we will exclude these participants from analyses.

($M = 52.93$, $SD = 12.13$)². Further examining this discrepancy with our recruitment selection criterion of being 35+ years of age, we saw that five participants reported an age below this (of which four were 32 years or older). Because we did not plan to exclude them at the data analysis step, they were retained. Although we selected only for individuals stating they are living alone, some participants reported marital status that implied they are living together: married $n = 29$ (3.5%) and living together as married $n = 23$ (2.8%), though the majority were single or never married 434 (52.67%), divorced 173 (21.00%); separated 28 (3.40%); or widowed 61 (7.40%). Although the ~~latter~~-living together as married group reflected participants who should not have been eligible to take part in the study, they comprised a small part of the sample and we retained them because we had not set this as an exclusion criterion. Their employment status also varied, with 323 (39.20%) reporting full-time work; 99 (12.01%) part-time; 120 (14.56%) self-employment, and 178 (21.60%) reporting they are retired. Finally, they varied by the location of their residence in a large city, 212 (25.73%), small city 257 (31.19%), rural area 105 (12.74%) and suburban area 249 (30.22%).

Guided by feedback received during Stage 1 review, the research team contacted participants for Time 2 of the project on March 22, 2020, just over one week after Time 1 survey was launched (March 14, 2020), and for Time 3 on March 29, 2020, one week after Time 2. All participants who took part in Time 1 were contacted and given a chance to take part in Time 2 and 3³. We received ethical approval from Durham University's Ethics Committee (PSYCH-2020-03-11T23_41_08). Participants provided informed consent before taking part, received an

² Of the sample, 32.4% were below 45 years of age, 23.7% were aged 45-54 years, 20.5% were aged 55-64 years, 20.3% were 65-74 years, and 3.1% were aged 75 years or older.

³ It should be noted that although the study was launched on March 14, not all participants completed the Time 1 survey on the day, so some of them might not have the full 1-week gap between Time 1 and Time 2 surveys.

in-part debrief following Wave 1 of the research, and were invited to skip items or withdraw from the study at any time.

Background Measures

The full set of measures collected for all time-points are available at OSF.IO/MN8KX.

Subjective health (Time 1; covariate). We assessed subjective health with one item from the World Value Survey (<http://www.worldvaluessurvey.org/wvs.jsp>). Specifically, participants were asked “All in all, how would you describe your state of health these days? Would you say it is...”, and received the options: *Very poor; Poor; Fair; Good; Very good; I’m not sure*.

Frequency of time spent alone (Times 1-3; quality check). At Time 1, participants were asked: “Considering the waking time you have, how much time *per day* do you typically spend *by yourself at home?*”, and reported with options between 0-20+ waking hours. Additionally, participants were asked: “How frequently in the past 2 weeks have you performed your daily activities at home by yourself?”, using a scale with the anchors: “*Most of the day*”, “*A few times a day*”, “*Once a day*”, “*Several times a week*”, “*Once a week*”, and “*Almost never*”. At Times 2, and 3, participants reported on these items, reflecting on the past week instead and on how much time they spent at home alone, “...not interacting with anyone in person or virtually”. See Table 1 for means and correlations of these items and other measurements of the quantity of solitude.

Frequency of time spent with others (Times 1-3; quality check & covariate).

Participants also reported on the frequency of social interactions using instructions from Rook (1984). They were given the prompt: “Think about the people with whom you socialize or enjoy conversations on a regular basis. How often do you interact with these people in general?”,

which was followed by four items specifying: In-person (face-to-face), Online (social media), Phone, or Text (written messages). These items were paired with a Likert-type scale including (1 = *hourly or several times a day*, 2 = *once a day*, 3 = *every couple of days*, 4 = *once a week*, 5 = *less than once per week*, 6 = *not at all*). Responses to all items were reverse-coded so that higher scores reflected higher levels of interactions. Then, items for online, phone, and text messages were averaged to create a single “virtual interactions” score ($\alpha_{\text{time 1}} = .56$; $\alpha_{\text{time 2}} = .55$; $\alpha_{\text{time 3}} = .55$).

Self-Isolation (Times 1-3; quality check). Participants responded to items: “In the past week, did you self-isolate in response to the coronavirus outbreak (COVID-19)?” with options including “no”, “*somewhat / in part*”, and “yes”. At Time 1, the percentage of participants who endorsed the “somewhat / in part” or “yes” responses was: 48%, at Time 2: 85%, and at Time 3, 88%. If responding “somewhat/in part” or “yes”, they were asked: “In the past week, what percentage of your time spent at home alone was deliberate self-isolation (keeping away from other people and places) in response to the coronavirus outbreak (COVID-19)?”; see Table 1 for means, standard deviations, and correlations with other measurements of solitude.

Ill-Being Measures Time 1 (Times 1-3)

Ill-being composite. An ill-being composite was constructed by averaging the scales of ill-being described below (composite $\alpha_{\text{time 1}} = .94$; $\alpha_{\text{time 2}} = .93$; $\alpha_{\text{time 3}} = .9394$). All measured at Time 1 were paired with the prompt: “In the past 6 months, I have felt...”, and at Times 2 and 3 with the prompt: “In the past week, when I was home alone and not interacting with anyone in-person or virtually, I have felt...”.

Loneliness. Participants reported on the Depletion and Isolation subscales of the Loneliness Rating Scale (Scalise, Ginter, & Gerstein, 1984), and specifically on its frequency,

since frequency of loneliness is more influential than its intensity (Diener, Sandvik, & Pavot, 2009). They responded on a scale of 1 (*never*) to 7 (*always*). Depletion items included “empty”, “secluded”, “alienated”, “withdrawn”, and “numb” ($\alpha_{\text{time } 1} = .94$; $\alpha_{\text{time } 2} = .95$; $\alpha_{\text{time } 3} = .95$). Isolation items included “unloved”, “worthless”, “hopeless”, “abandoned”, and “deserted” ($\alpha_{\text{time } 1} = .96$; $\alpha_{\text{time } 2} = .96$; $\alpha_{\text{time } 3} = .96$).

Depression. The shortened 10-item version of the CES-D (Andersen, Byers, Friary, Kosloski, & Montgomery, 2013) was used to measure depression. Participants reported how often: “I felt depressed”, “I felt everything I did was an effort”, and “I felt fearful”, with a scale ranging from 1 (*not at all*) to 6 (*most of the time*) ($\alpha_{\text{time } 1} = .91$; $\alpha_{\text{time } 2} = .92$; $\alpha_{\text{time } 3} = .92$). This measure was transformed into a 7-point scale before it was combined with other measures to calculate the ill-being composite. To do this, we applied the linear stretch method (De Jonge, Veenhoven, & Arends, 2014) using the following formula: \$\text{new score} = (7 - 1) * (\text{old score} - 1) / (6 - 1) + 1\$.

Anxiety. Anxiety was measured with a validated 6-item measure of the State-Trait Anxiety Inventory (STAI) as recommended by (Tluczek, Henriques, & Brown, 2009)⁴. Items were paired with a scale ranging from 1 (*not at all*) to 7 (*very much so*) ($\alpha_{\text{time } 1} = .93$; $\alpha_{\text{time } 2} = .92$; $\alpha_{\text{time } 3} = .93$).

Predictor Variables Time 1 (baseline)

Preference for solitude. We took six items from the 12-item measure Preference for Solitude scale (Burger, 1995). We chose items that were generally worded so that they apply to most people, whereas items that refer to specific circumstances were not included (e.g., “One

⁴ This is a partial set of the State-Trait Anxiety Inventory for Adults items used with the permission of the publisher. The instructions and response scales have been modified for the present study. We included the authorized sample items of the copy-righted version in Appendix B. These sample items are not authorized for reuse or modification.

feature I look for in a job is the opportunity to spend time by myself”). We drew two items from each of the three factors identified by Cramer and Lake (1998) to cover all dimensions of this construct. Instructions were as follows: “For each of the following pairs of statements, select the one that best describes you. In some cases, neither statements may describe you well or both may describe you somewhat. In those cases, please select the statement that best describes you or that describes you more often.” Respondents selected one of two options per item, e.g., “I enjoy being around people”, or “I enjoy being by myself” ($\alpha = .72$).

Motivation for solitude. Identified (autonomous) and external (controlled) motivations for solitude were measured with a 10-item scale adapted from Nguyen and colleagues (2018). Participants were asked: “Think of times when you will be by yourself, at home, in the next 2 weeks. Those are times when you do not interact with anyone in person or virtually”, followed by items measuring autonomous motivation through its form most applicable to the context of self-isolation, identified motivation (finding value and importance in the activity; Ryan & Deci, 2017). Five items for this subscale included “Having time to myself is important and beneficial to me” (original item), and “Spending time alone is important for protecting my health” (new item). Overall internal reliability was acceptable ($\alpha = .73$). Participants also responded to five items measuring controlled motivation through its form most applicable to the context of self-isolation, external motivation (feeling external pressures and choicelessness), with items such as “I am forced into it” (original item), “I would get in trouble with others if I didn’t” (original item), and “I feel I have no choice” (new item). Items were paired with a 7-point scale ranging from 1 (*this does not apply to me at all*) to 7 (*this applies to me very much*). Overall internal reliability was high ($\alpha = .84$).

Descriptive and Covariate Measures Time 2 and 3 Following Stage-1 Review

COVID-19 concerns. Two items assessed COVID-19 health concerns. ~~first-First~~, participants responded to the items: “I have been diagnosed with COVID-19” (Yes / No). “I strongly suspect that I currently have a COVID-19 infection” (Yes/No). Participants who reported “Yes” to either of these items were considered as having serious concerns about COVID-19.

Health anxiety. Typically a disposition-level measure, participants also responded to five items of the Health Anxiety Inventory (Salkovskis et al., 2002) that measure health anxiety at a state level⁵. Participants selected from four options reflecting increasing concerns. For example one item is: I have not worried about my health” “I have occasionally worried about my health” “I have spent much of my time worrying about my health” and “I have spent most of my time worrying about my health” ($\alpha_{\text{time 2}} = .87$; $\alpha_{\text{time 3}} = .87$).

Stressors checklist. Participants were asked: “In the past week, did you experience any of the following? (tick all that apply)”, and selected the options: “Lost your job”, “Lost a substantial amount of earnings from your job”, “Were worried that COVID-19 would impact your job security”, “Did not have enough food”, “Could not access food”, “Could not access important medical supplies”, “Knew someone who was diagnosed with COVID-19”, or “Were diagnosed with any new health condition”. See Table 2 for frequencies of each stressor experienced at Time 1 and 2.

Overall depression. We used the same measure described above for testing depression in solitude to also test depression overall, with the instructions to participants to reflect on how they

⁵ Originally, we picked out six items from the Short Health Anxiety Inventory (SHAI). However, after we launched the Time 2 survey, we caught one erroneous item that said “I noticed aches and pains less than/more than usual”. We excluded this item from the composite, so the final SHAI measure ~~has had 5~~ instead of 6 items.

felt generally (that is presumably, both within and outside of solitude) in the past week, at Times 2 ($\alpha = .91$) and 3 ($\alpha = .92$). Following Stage 1 review, this measure was intended to correlate against our ill-being in solitude measures and provide an exploratory test of the correlations between ill-being in solitude, and ill-being in general.

Results

Effect Size Estimates and Interpretation

Informed by related solitude work described below, we set criteria for smallest effect sizes of interest in our hypotheses to predetermine standards for what we will decide as meaningful when evaluating the results instead of relying on p values. We used $-.15$ *partial rs* as the smallest effect size of interest (equivalent to .023% of variance explained) to determine whether main effect hypotheses (H1:1, H1:2, H2:1, H2:2) are supported. For moderation effects hypothesised in H3:1, H3:2, H4:1, H4:2, when calculating the relative difference between two correlation coefficients, we used .10 difference between the two standardized coefficients (for example, between $-.40$ and $-.30$, or $-.20$ and $-.10$) as the smallest effect size of interest to determine whether this hypothesis is supported.

Rationale for choosing partial r around .15. A previous paper by Nguyen, Werner, and Soenens (2019) looked at the link between autonomous motivation (conceptually similar to high identified motivation and low external motivation) and well-being and ill-being variables. In Study 1, sampling first-year university students ($n = 147$), the authors measured motivation for spending time alone and loneliness experienced in the past 2 weeks (not specifically related to time spent alone) and assessed these variables three times. On average, the correlation between autonomous motivation and loneliness was approximately $r = -.30$. Extraversion - a concept similar to preference for solitude - also correlated with loneliness at also around $r = -.30$.

However, because these correlations were obtained from a rather small sample, effect sizes were likely overestimated (Yartoki, 2009). As such, we selected to place our benchmark at a partial r half the size of those correlations obtained by past research.

Rationale for choosing .10 difference between regression coefficients. In Study 2 of Nguyen et al. (2019), the researchers also examined the interaction between participants' perceived belonging with autonomous motivation predicting levels of loneliness experienced in the past one month; this moderation test is the closest available data to our hypothesised moderation effects (Hypothesis sets 3 and 4) and yielded a coefficient of, $\beta = -.32$, CI 95% $[-0.44, -0.20]$, when predicting loneliness. At higher levels of autonomous motivation, perceived belonging related to lower loneliness, $\beta = -0.04$, CI 95% $[-0.21, 0.13]$. Again, findings relied on a sample of $n = 223$, so we expect the effect sizes were likely overestimated. Selecting regression coefficients half the size of those coefficients obtained by past research, we anticipated, $r = -0.16$ at low levels of autonomous motivation, and, $r = -0.02$ at high levels. The difference between these two coefficients was about, $\Delta r = .14$, but a conservative expectation is, $\Delta r = .10$.

Exclusion Criteria

We set three criteria to guide which participants would be excluded from analyses: First, we tested for inattention with two items “Choose “somewhat agree”” and “Choose “Very true”” embedded in two different scales of Times 2 and 3. We excluded participants who either 1) failed to select the corresponding answers as instructed, or 2) failed at least one attention check item (if the participant misses the item, that is considered failing the attention check). Second, we anticipated we would exclude participants who responded after we submitted the stage 1 Registered Report, but when we downloaded the data, we saw those were duplicates or missing

responses. Procedure for removing duplicates was included in our analysis script (OSF.IO/MN8KX). Finally, we excluded participants if we could not link their data across at least two time-points.

Confirmatory Quality Checks

This study does not involve an experimental manipulation, and for this reason we did not include a manipulation check. However, the assumption underlying our hypotheses is that our sample, on the whole, will find themselves spending more time alone than usual as a function of the COVID-19 outbreak. As such, we conducted three sets of analyses (based on measures described as quality indicators in the Materials description above) to test our belief that any changes in ill-being from Time 1 to Time 2 are due to the effects of social isolation:

Time spent alone. We expected significant (using the $p < .05$ cutoff) increases in frequency (i.e., “How frequently have you performed your daily activities at home by yourself?”) and length (i.e., “Considering the waking time you have had in the past week, how much time per day did you typically spend at home alone?”) of time spent alone from Time 1 to Time 2 and Time 1 to Time 3. Results of two paired samples t -tests showed that on average participants performed daily activities by themselves at home significantly less frequently at Time 2 ($M = 5.23$, $SD = 1.29$) as compared to Time 1 ($M = 5.35$, $SD = 1.13$; $t(703) = 2.37$, $p = 0.018$). The difference between Time 1 and Time 3 ($M = 5.28$, $SD = 1.32$) did not reach statistical significance ($t(701) = 1.39$, $p = .164$). This pattern was not as we expected.

On the other hand, participants reported spending increased numbers of waking hours alone from Time 1 ($M = 10.26$, $SD = 5.34$) to Time 2 ($M = 11.52$, $SD = 5.54$; $t(748) = -6.18$, $p < .001$), and from Time 1 to Time 3 ($M = 12.14$, $SD = 5.25$; $t(751) = -8.71$, $p < .001$). As we

expected, on average people reported spending more hours at home alone in the second and third assessment.

In-person interactions. We expected a significant (at $p < .05$) drop in *in-person* interactions from Time 1 to Time 2 and Time 1 to Time 3. Results showed that, when compared to Time 1 ($M = 4.04$, $SD = 1.43$), people reported having in-person interactions less frequently at Time 2 ($M = 2.78$, $SD = 1.73$; $t(749) = 19.43$, $p < .001$) and Time 3 ($M = 2.16$, $SD = 1.60$; $t(751) = 27.36$, $p < .001$).

Self-isolation questions. We expected a statistically significant (at $p < .05$) increase in the percentage of people endorsing “yes” and “in part” to isolating because of COVID-19, and an increase in the percentage of time spent alone from Time 1 to Time 2 and Time 1 to Time 3. Results showed that significantly higher percentages of participants endorsing “yes” and “in part” to isolating because of COVID-19 at Time 2 (85%; $t(735) = -19.52$, $p < .001$) and Time 3 (88%; $t(735) = -21.54$, $p < .001$) as compared to Time 1 (48%). Further, among those who answer “yes” and “in part”, these people also reported spending larger proportion of time alone at home deliberately in response to COVID-19 at Time 2 (71%; $t(333) = -15.37$, $p < .001$) and Time 3 (80%; $t(328) = -17.20$, $p < .001$) as compared to Time 1.

Overall, out of five quality checks, four showed expected patterns. Participants indeed engaged in less in-person interactions and spent more waking hours alone between Time 1 and the two follow-up time points. In addition, more participants reported self-isolating in response to COVID-19 at Time 2 and Time 3 than at Time 1, and those who self-isolated also spent higher percentages of time alone self-isolating in response to COVID-19.

Confirmatory Analyses

Correlations. Table 3 presents the means, standard deviations, and planned Pearson correlations of our three main predictors and specific ill-being variables at all time-points. We tested unadjusted links between preference, identified motivation, and external motivation to ill-being at Time 2 and Time 3. Those who preferred to be alone reported more identified motivation ($r = .39$), and less external motivation ($r = -.13$) for solitude, suggesting that there is some overlapping between these variables but only to a small degree. With 15% shared variance between preference and identified motivation and less than 2% shared variance between preference and external motivation, ~~This indicated that~~ preference and motivation are/were distinct reasons for seeking solitude. Further, no relations were evident of preference with ill-being constructs at any of the time-points, and weak and mixed relations were evident of identified motivation with ill-being at Times 1-3. The most consistent correlations were observed between external motivation and ill-being, which averaged $r = .26$ across ill-being indicators and time-points (see Table 3). For readers' interest, Table 4 also presents correlations between our three main predictors and ill-being, our outcome of interest, along with study covariates.

Primary models. As planned, primary analyses regressed the composite of our outcomes (loneliness: depletion; loneliness: isolation; depression; anxiety) at Time 2 and Time 3, controlling for the Time 1 ill-being composite. Main models testing our hypotheses accounted for eight potentially confounding effects: gender, age, follow-up in-person social contact, follow-up virtual contact (average of chat, text, and phone frequency), virus diagnosis or suspicion (Y/N), health anxiety, subjective health at Time 1, and the stressor checklist at follow-up.

For each model, we first plotted the distribution of the residual errors to check for normality. All plots indicated that residual errors were normally distributed, and therefore no

transformation was needed. We tested our hypotheses using linear regression models as assumption of normality was satisfied.

Hypothesis 1:1, 1:2, 2:1, and 2:2 (concerning main effects of preference and motivations on ill-being). In two hierarchical linear regression ~~analysis~~analyses, we regressed Time 2 and Time 3 ill-being on covariates (listed above) at Step 1, Time 1 preference for solitude at Step 2, and Time 1 identified and external motivation at Step 3. Findings are summarised in Table 5 and showed that preference and both types of motivation for solitude explained little variance in residual change in ill-being from Time 1 to Time 2, as well as from Time 1 to Time 3. External motivation for time alone predicted an increase in Time 2 ill-being ($\beta = .065$, CI 95% [.01, .1009], $p = .00920$), yielding a (*partial r*) *pr* of .10, which is smaller than the effect size of interest we determined a priori. Likewise, preference for solitude predicted a decrease in Time 3 ill-being ($\beta = -.065$, CI 95% [-.10, -.010], $p = .01330$), yielding a *pr* of .10, which is also smaller than the effect size of interest.

Hypothesis 3:1, Hypothesis 3:2 & Hypothesis 3:3 (concerning interacting effects with identified motivation). In two hierarchical linear regression analyses, we regressed follow-up (Time 2 and 3) ill-being on covariates at Step 1, Time 1 preference for solitude and Time 1 identified motivation at Step 2, and the interaction term of preference for solitude and identified motivation for solitude at Step 3. Findings are summarised in Table 6 and did not show evidence of interaction between preference for solitude and identified motivation for time alone ~~for both in~~ either Time 2 ~~and-or~~ Time 3 analyses.

Hypothesis 4:1, Hypothesis 4:2 & Hypothesis 4:3 (concerning interacting effects with external motivation). In two hierarchical linear regression analyses, we regressed follow-up (Time 2 and 3) ill-being on covariates at Step 1, Time 1 preference for solitude and Time 1

external motivation at Step 2, and the interaction term of preference for solitude and external motivation for solitude at Step 3. As summarised in Table 7, we also did not find evidence supporting our hypothesis that there is an interaction between preference for solitude and identified motivation for time alone ~~for in both either~~ Time 2 ~~and or~~ Time 3 analyses.

Summary of confirmatory analyses. Planned unadjusted correlations showed a consistent link between external motivation and higher ill-being as measured through loneliness-depletion, loneliness-isolation, depression, and anxiety at all time-points. However, after accounting for controls including Time 1 ill-being, we found little evidence suggesting that initial preference for solitude and motivation behind spending time alone were predictive of change in ill-being across a period of two weeks. Instead, we found that health anxiety and life stressors, measured simultaneously with the outcome variables, linked strongly with increases in ill-being at Time 2 and Time 3. To the extent that individuals felt anxiety surrounding their health, they reported increased ill-being in their solitude reported at Time 2 and Time 3. Further, with more stressors experienced in the past week (of which job and food insecurity were most predominant, see Table 2), participants reported more ill-being during their solitude across two weeks. Only health anxiety, however, yielded $prs \geq .4542$ at both Time 2 and ~~$pr = .41$ at~~ Time 3, which is larger than our smallest effect size of interest, whereas life stressors yielded an independent effect size below our predetermined threshold at both Time 2 and Time 3. Other characteristics like age and gender, subjective health, frequency of in-person interactions or virtual interactions in the past week also did not independently predict Time 2 and Time 3 ill-being in our fully adjusted models.

Exploratory Analyses

We conducted additional exploratory – unplanned – tests to better understand how change in ill-being occurred across the two weeks of the study and what this means for our three main predictors.

Change in ill-being across early weeks of self-isolation. First, we were interested in the extent participants experienced change in ill-being over the course of two weeks, over and above their pre-existing levels of ill-being. We sought to test, in part, public concerns that self-isolation in itself would yield loneliness and depression (e.g., CNBC.com, 2020; Guadiran.com, 2020; Health.com, 2020; NewYorker, 2020). On one hand, our sample may more easily access online resources since they participated in an online study, but on the other hand they represented older adults that have been the focus of primary concern (average age approximately: $M = 52$ years), who are living alone – presumably both of these were risk factors for more ill-being.

Paired sample t-tests comparing Time 1 ill-being with Time 2 and Time 3 ill-being suggested that levels of ill-being over two weeks did not differ from participants' prior ill-being levels. At the beginning of the study, the average level of ill-being experienced in the past 6 months was, $M = 2.8453$, $SD = 1.2416$. One week after, the average ill-being was no different, ($M = 2.8449$, $SD = 1.3023$ ($t(749) = .17485$, $p = 0.863065$), and two weeks after, the average ill-being was no different ($M = 2.8548$, $SD = 1.3426$ ($t(751) = -.49121$, $p = 0.626228$). When we coded for participants who scored higher than mid-point (4 on 7-point scale), we found that 1320% fell under this category at Time 1, 12% at Time 2, and 14% at Time 3 all three time points. In the discussion section, we elaborated on these percentages in relation to recent data reported by a review by Brook et al. (2020) on percentages of people experiencing psychological distress while in quarantine in previous outbreaks (i.e., H1N1, SARS, etc). Convergent evidence

suggested that we did not observe expected inclines in ill-being across these two weeks of self-isolation.

Note that in this sample, when we coded for those who reported spending 100% of their day self-isolating due to COVID-19, we found that 6% fell under this category at Time 1, 21% at Time 2, and 34% at Time 3. Further, when we compared those who reported spending 100% of their day at Time 2 and Time 3 self-isolating due to COVID-19, with others who spent a lower percentage of time, ill-being scores between those two groups did not meaningfully differ (see *SOM 1*).

Relations of ill-being in solitude and global depression. Second, following advice in Stage 1 review, we were interested in whether measures of ill-being in solitude were distinguished from depressive symptoms, overall. That is, we wanted to know whether ill-being in solitude can be distinguished from global ill-being, occurring in the same timeframe. Ill-being at Time 2 correlated, $r = .84$, with overall depressive symptoms at Time 2, and ill-being at Time 3 correlates $r = .86$ with overall depressive symptoms at Time 3. Most notable for us, *depression* in solitude correlated, $r = .93$, and $r = .94$, with depression in general at Times 2 and 3, respectively. These high correlations meant that we do not have evidence that, during this time, reports of ill-being during solitude were different from overall ill-being; the two were, for all intents and purposes, synonymous.

Links of Preference and Motivation to Ill-being. In Pearson correlation analyses between major study variables (reported in Table 4), we observed strong relations between Times 1, 2, and 3 ill-being ranging from $r_s = .78 - .89$, and in exploratory analyses we found that not much change occurred in ill-being. We were therefore concerned that the vulnerability model that drove our views that preference and motivation would have predictive value for later ill-

being, controlling for earlier ill-being, was unwarranted. In other words, the data did not seem to support a view that self-isolating living-alone adults would have the potential for increases in ill-being, that could be buffered or exacerbated by our predictors.

We therefore conducted two linear regression models similar to the models used to test Hypothesis 1:1, 1:2, 2:1, and 2:2, except that this time we did not control for Time 1 ill-being. By doing this, we simply tested whether preference and motivation measured at Time 1 would be linked with later assessments of ill-being. This analysis should be interpreted with caution, because it reflects, essentially, a series of partial correlations. While one measure precedes the other, this should not be interpreted as a causal effect. For this reason, although vulnerability due to health concerns, demographics, and actual quantity of solitude was accounted for, many other possible confounds may play a role in inflating these relations. Findings are summarised in SOM 1.

A number of meaningfully significant links were in evidence in these models. First, in this model older age related to *lower* ill-being, $pr = -.17$, and the relation of stressors slightly increased and met our planned effect size of interest, $pr = .20$ at both Time 2 and Time 3. As before, the relation of health anxiety with ill-being measured at the same time-point was by far the most robust one identified, $pr = .5145$ at Time 2 and $pr = .5247$ at Time 3. Accounting for covariates, external motivation linked to ~~more~~-higher ill-being at Time 2, $pr = .22$, and at Time 3, $pr = .17$. In summary, these analyses show non-causal links over the short-term that may not extend across longer periods.

Discussion

Confirmatory results did not support our hypotheses regarding the role of preference and motivation for solitude on ill-being across time. The links between preference and motivation for

solitude, and later ill-being at Time 2 and Time 3, which controlled for Time 1 ill-being along with a number of vulnerability factors, did not meet the threshold we had set for the minimum effect size of interest. Our minimum effect size of interest was set based on daily experiences of solitude in studies of undergraduate students (Nguyen, Werner, & Soenens, 2019). Although we had assumed a correlation half the size of the one identified in this literature ($r = .30$; Nguyen, Werner, & Soenens, 2019), the effect observed for external motivation was even lower, $pr = .09$ at Time 2 and $pr = .06$ at Time 3, than our minimum effect of interest, $pr = .15$ set at the outset.

Similarly, the effect of preference was, $pr = .04$ at TimeWeek 2, and $pr = .09$ at TimeWeek 3. Thus, observed increases in ill-being for those who feel pressured to self-isolate, and those who prefer to spend time alone more than with others, are very small. Further, these two variables did not interact to predict change in ill-being over the course of two weeks. The conclusion of this finding is that, even when individuals might prefer time alone or endorsed the importance of self-isolating, for example for their own or others' benefit, this study did not reveal evidence supporting that it ameliorates or exacerbates their loneliness, depression, or anxiety during time spent alone. Even with additional exploratory analyses to test relations of both preference and motivation to ill-being at each of the two follow-up time-points, we did not observe an effect above our minimum effect size of interest, except in the case of external motivation predicting higher ill-being one week later.

We see three possibilities for why we observed lower effect size than those previously observed, which merit further study. First, it could be argued that any effects of identified and external motivation for solitude do not apply to *extreme* solitude. However, even in studies of forced solitude in the extreme, such as solitary confinement in prisons where choicelessness and external motivation are extremely high, motivation is linked to negative mental health

consequences for prisoners (Arrigo & Bullock, 2008; Metzner & Fellner, 2013). It is unlikely that our study captured the extreme cases of isolating individuals, and that for this reason preference and motivation played a less important role.

~~It is worth noting that~~On a similar note, the sample obtained in this research was not comparable to other cases of isolation due to quarantine. Obligatory quarantine of those who have been exposed to a contagious illness causes great distress to quarantined individuals, but quarantined cases and those who are in voluntary self-isolation like the cases we are studying in our sample are in very different circumstances (Brooks et al., 2020). The levels of distress experienced among those in quarantine are likely to be higher than those in voluntary self-isolation. For example, in a study looking at experiences of horse owners during quarantine due to an equine influenza outbreak in Australia, 34% of those in quarantine reported distress; this is notably higher than the percentage of those experiencing distress in the Australian general population (Taylor et al., 2008). In another sample of parents impacted by H1N1 and SARS outbreaks, 28% of parents in quarantine reported trauma-related disorder, compared with 6% of parents not in quarantine (Sprang & Silman, 2013). In our sample, as reported in the exploratory tables, 13% of the participants scored higher than the mid-point on ill-being (higher than 4 on 7-point range) at Time 1, 12% at Time 2, and 14% at Time 3.

Further reflecting on the circumstances in which our participants found themselves, although our quality checks regarding the percentage of time spent alone and in self-isolation showed significant increases on these indicators, we also identified that the amount of time spent doing daily activities at home, alone, did not increase. While this finding was unexpected, it is likely this is due to the fact that our living alone adults already undertook a large number of tasks on their own, so that their daily at-home activities did not change much during the COVID-19

crisis. Thus, perhaps despite more time spent alone and in self-isolation, our participants may have experienced a gentler transition into self-isolation wherein some daily habits were uninterrupted, and therefore costs to wellness were lower.

Second, it is possible that there is-was little change for preference and motivation for solitude to predict. While previous research suggested that prolonged self-isolation would yield psychological consequences, in the current study we did not see evidence of change in ill-being over the course of two weeks of self-isolation. It is worth noting that the two-week span of the study represented a fairly short duration, and that ill-being changes may be observable as individuals spend additional weeks and months in self-isolation. Given this, it is worthwhile to engage ongoing research efforts tracking ill-being to identify a ‘breaking point’ in which increases in ill-being may be observed. Though we cannot speak to this possibility at present, it may be that non-linear trends of increasing ill-being will be evidenced across longer spans of time.

In our sample, which primarily varied in ages from 35 to 87 years, participants across the adulthood spectrum who reported engaging in less in-person interaction and spending more waking hours alone, felt similarly during this time of self-isolation. These findings speak to concerns regarding older adults during self-isolation. Certainly, older adults, who are more vulnerable to COVID-19 exposure, more highly restricted to time at home, and generally less comfortable with meeting their own needs online, benefit from the help and support of community. However, at least within our sample, we did not find evidence that current circumstances yield depression, loneliness, or anxiety, despite media warnings about negative consequences of rapid move to self-isolating during the period of the study (CNBC.com, 2020; Guadiran.com, 2020; Health.com, 2020; NewYorker, 2020). Thus, concerns that self-isolation is

harmful to mental health of adults were unsubstantiated in our findings. In fact, few of our predictors could affect change in ill-being at all, which remained low through the duration of the study.

Third, this study was conducted amidst precipitous changes in governmental regulations and guidelines, as well as individuals' understanding of and plans for self-isolation, and although preference is seen as a dispositional quality that should not much change (Burger, 1995), motivations may have shifted within the first week of the study. Since we did not measure motivation past Time 1, we cannot answer that question. However, we attempted to acknowledge the quickly changing landscape of COVID-19 by, part-way through the study, moving to weekly data collections following Stage 1 reviewer advice. Two days before Time 2 data collection, the U.K. shut down non-essential public places and residents were asked to stay at home ~~at~~ nearly all times-the time (The Guardian); one day after the U.S. saw state-wide school closures by more than half of states (CNN). Since Time 2 referred primarily to the week before these changes took place, it reflected initial motivations more closely. Time 3 had seen additional extended self-isolation and self-isolation driven by more pressures, such as highly limited access to groceries for those who were home (e.g., The Guardian). Certainly, it is reasonable to assume that the sense of urgency and pressure around self-isolation changed following these decisions, and similarly that perceived importance may have changed. If it is the case, it speaks to the malleability of motivations as a function of communications and circumstances surrounding COVID-19. Thus, a fruitful topic of future research would be to experimentally and longitudinally test motivations to examine the effects of these communications on ill-being.

Interestingly, the effect size for identified motivation changing ill-being was close to 0.

~~Therefore, w~~We did not find evidence that when people recognised the importance of self-

isolating, this protected them from ill-being during time spent alone. Future research may consider other outcomes of identified motivation, for example on behavioural intention (Deci & Ryan, 2008). Providing people recognise the importance of self-isolation, they may be more likely to cooperate and engage in this behaviour (Ryan & Deci, 2017). Yet, our findings suggest that when they do so, they may experience the same mental health costs or benefits within solitude – we see evidence of neither – of this cooperation.

Importantly, even with the little change increased ill-being, the one predictor that robustly predicted change in ill-being its change was a concern about health, which was measured simultaneously with ill-being dependent variables. Higher health concerns measured at the same time as the dependent variable related to more loneliness, depression, and anxiety. Thus, health anxiety was by far the most compelling vulnerability for individuals in self-isolation, a finding which suggests that practical and emotional support for health needs is particularly important during self-isolation.

Limitations

These findings should be viewed in light of two limitations of this study design. First, as described previously, the study took place during a time of fast-paced changes to circumstances, perceived health concerns, and expectations for the duration of lifestyle restrictions, all of which changed between each time-point collected. On one hand, these changes contribute to the strength and richness of the study, since longitudinal analyses stretched the capacity of our predictors to test their boundary conditions in these unusual circumstances. Further, because our predictor measures were collected at the start of the study (Time 1) only, we cannot know how preference and motivation for solitude might have changed during this time.

A second major concern is that it is unclear whether we tested ill-being in solitude, or ill-being in general. In fact, depression in solitude correlated ($r = .93$ (Time 2) and $.95$ (Time 3)) with a general measure of depression we had collected following Stage 1 reviewer recommendations to conduct this test. Understandably, when individuals spend a majority of their waking hours in solitude these two should be nearly identical. However, we cannot be certain whether solitude that defines daily experiences, as it does now when individuals are self-isolation, behaves the same as solitude that is incidental in daily life under normal circumstances. In future research or in meta-analytic work later, it would be insightful to compare the two conditions to understand where they differ and where they share characteristics.

Conclusions

The experience of living-alone individuals self-isolating is not atypical. They represent anywhere from 23.9% to 50% of households in the U.K. and U.S. (Our World in Data, 2019; Office for National Statistics, 2019), the two populations from which we sampled. In this study, we did not find evidence that the mental health of these individuals had been negatively affected by self-isolation in its early weeks, a finding that clearly differentiates this group from those who find themselves in forced quarantine (Brooks et al., 2020). Further, we did not find evidence in support of our planned hypotheses above the minimum effect size of interest. Other studies have similarly failed to evidence change in wellness as a function of motivation in solitude, though they evidenced direct links with concurrent wellness (Nguyen, Werner, & Soenens, 2019). However, we had anticipated that protective or aggravating predictive factors would play a stronger role predicting change because of the unexpected and sudden increases in the time spent alone in response to the COVID-19 outbreak.

One might conclude that the absence of evidence for increased ill-being reflects resilience in the face of self-isolation. However, such an interpretation is unwarranted without additional data that speak directly to the mechanisms underlying resilience. Specific groups might react differently to self-isolation with different personal or social resources, but testing this requires direct comparisons with a lower resilience group who report increased ill-being under similar conditions. We would caution researchers to be sensitive to possible nuances in reactions ~~to~~ COVID-19, including in the context of self-isolation, and to avoid making assumptions and post-hoc interpretations based on studies of forced quarantine or loosely applied psychological models. Doing so may raise false hopes and misdirect limited resources.

Ethics. This research was approved by Durham University's Ethics Committee (PSYCH-2020-03-11T23_41_08).

Data accessibility. Data are available on OSF (OSF.IO/MN8KX), where readers can find our pre-registrations, raw data and code. Analyses were primarily conducted in R (except for reliabilities), and we offer the R code, as well as SPSS files and syntax for composites and reliabilities.

Author contributions. NW and TVN worked in a fully collaborative fashion on this project. Together, they conceptualised the study, developed methods, ethical documentation, and analytic plan, prepared registrations and study materials, and wrote Stage 1 of the registered report. At Stage 2, they once again collaborated to analyse data and write the full report.

Competing interests. The authors have no competing interests to declare.

Funding. This work was supported in-part by a grant from the European Research Council [ERC; SOAR; 851890].

References

- Andresen, E. M., Byers, K., Friary, J., Kosloski, K., & Montgomery, R. (2013). Performance of the 10-item Center for Epidemiologic Studies Depression scale for caregiving research. *SAGE open medicine, 1*, 2050312113514576.
- Arrigo, B. A., & Bullock, J. L. (2008). The psychological effects of solitary confinement on prisoners in supermax units: Reviewing what we know and recommending what should change. *International journal of offender therapy and comparative criminology, 52*(6), 622-640.
- Berinsky, A. J., Margolis, M. F., Sances, M. W., & Warshaw, C. (2019). Using screeners to measure respondent attention on self-administered surveys: Which items and how many?. *Political Science Research and Methods, 1*-8.
- Bingham, C. R., & Shope, J. T. (2004). Adolescent developmental antecedents of risky driving among young adults. *Journal of studies on alcohol, 65*(1), 84-94.
- Brooks, S. K., Webster, R. K., Smith, L. E., Woodland, L., Wessely, S., Greenberg, N., & Rubin, G. J. (2020). The psychological impact of quarantine and how to reduce it: rapid review of the evidence. *The Lancet*. [https://doi.org/10.1016/S0140-6736\(20\)30460-8](https://doi.org/10.1016/S0140-6736(20)30460-8).
- Burger, J. M. (1995). Individual differences in preference for solitude. *Journal of Research in Personality, 29*(1), 85-108.
- Cacioppo, J. T., Hawkey, L. C., Crawford, L. E., Ernst, J. M., Burleson, M. H., Kowalewski, R. B., ... & Berntson, G. G. (2002). Loneliness and health: Potential mechanisms. *Psychosomatic medicine, 64*(3), 407-417.
- Chambers, D. (2013). *Social media and personal relationships: Online intimacies and networked friendship*. Springer.

- Coleman, J. S. (1972). How do the young become adults?. *Review of Educational Research*, 42(4), 431-439.
- Cramer, K. M., & Lake, R. P. (1998). The Preference for Solitude Scale: Psychometric properties and factor structure. *Personality and Individual Differences*, 24(2), 193-199.
- Cuenya, L., Fosachecca, S., Mustaca, A., & Kamenetzky, G. (2012). Effects of isolation in adulthood on frustration and anxiety. *Behavioural processes*, 90(2), 155-160.
- Deci, E. L., & Ryan, R. M. (2008). Self-determination theory: A macrotheory of human motivation, development, and health. *Canadian psychology/Psychologie canadienne*, 49(3), 182-185.
- De Jonge, T., Veenhoven, R., & Arends, L. (2014). Homogenizing responses to different survey questions on the same topic: Proposal of a scale homogenization method using a reference distribution. *Social Indicators Research*, 117(1), 275-300.
<https://doi.org/10.1007/s11205-013-0335-6>
- Diener, E., Sandvik, E., & Pavot, W. (2009). Happiness is the frequency, not the intensity, of positive versus negative affect. In *Assessing well-being*(pp. 213-231). Springer, Dordrecht.
- Emmons, R. A., & McCullough, M. E. (2003). Counting blessings versus burdens: an experimental investigation of gratitude and subjective well-being in daily life. *Journal of personality and social psychology*, 84(2), 377-389.
- Holliday, S. G. (1988). Risky-choice behavior: A life-span analysis. *The International Journal of Aging and Human Development*, 27(1), 25-33.
- Mata, R., Josef, A. K., Samanez-Larkin, G. R., & Hertwig, R. (2011). Age differences in risky choice: a meta-analysis. *Annals of the New York Academy of Sciences*, 1235, 18-29.

- Metzner, J. L., & Fellner, J. (2013). Solitary confinement and mental illness in US prisons: A challenge for medical ethics. *Health and Human Rights in a Changing World*, 316-323.
- Molyneux, N., & Parkinson, L. (2016). Systematic review of interventions addressing social isolation and depression in aged care clients. *Quality of Life Research*, 25(6), 1395-1407.
- Nguyen, T. V. T., Ryan, R. M., & Deci, E. L. (2018). Solitude as an approach to affective self-regulation. *Personality and Social Psychology Bulletin*, 44(1), 92-106.
- Nguyen, T. V. T., Weinstein, N., & Ryan, R. M. (Under Review). *Unpacking the “Why” of Time Spent Alone: Who Prefers and Who Chooses it Autonomously*.
- Nguyen, T. V. T., Werner, K. M., & Soenens, B. (2019). Embracing me-time: Motivation for solitude during transition to college. *Motivation and Emotion*, 43(4), 571-591.
- Office for national statistics (2019, Nov 15). *Families and households in the UK: 2019*.
<https://www.ons.gov.uk/peoplepopulationandcommunity/birthsdeathsandmarriages/families/bulletins/familiesandhouseholds/2019>
- Our World in Data (2019). *Percentage of americans living alone*.
<https://ourworldindata.org/grapher/percentage-of-americans-living-alone-by-age?time=2016&country=Age%2018+Age%2021+Age%2030+Age%2045+Age%2060+Age%2075+Age%2089>
- Rook, K. S. (1990). Stressful aspects of older adults' social relationships: Current theory and research. *Stress and coping in later-life families*, 173-192.
- Ryan, R. M., & Deci, E. L. (2017). *Self-determination theory: Basic psychological needs in motivation, development, and wellness*. NY: Guilford Publications.

- Salkovskis, P. M., Rimes, K. A., Warwick, H. M. C., & Clark, D. M. (2002). The Health Anxiety Inventory: development and validation of scales for the measurement of health anxiety and hypochondriasis. *Psychological medicine*, 32(5), 843-853.
- Sarason, I. G., Levine, H. M., Basham, R. B., et al. (1983). Assessing social support: The Social Support Questionnaire. *Journal of Personality and Social Psychology*, 44, 127–139.
- Scalise, J. J., Ginter, E. J., & Gerstein, L. H. (1984). Multidimensional loneliness measure: the loneliness rating scale (LRS). *Journal of Personality Assessment*, 48(5), 525-530.
- Sorrentino, R. M., & Higgins, E. T. E. (1986). *Handbook of motivation and cognition: Foundations of social behavior*. Guilford Press.
- Sprang, G., & Silman, M. (2013). Posttraumatic stress disorder in parents and youth after health-related disasters. *Disaster medicine and public health preparedness*, 7(1), 105-110.
DOI: <https://doi.org/10.1017/dmp.2013.22>
- Storr, A. (1988). *Solitude: A return to the self*. Simon and Schuster.
- Taylor, M. R., Agho, K. E., Stevens, G. J., & Raphael, B. (2008). Factors influencing psychological distress during a disease epidemic: data from Australia's first outbreak of equine influenza. *BMC public health*, 8(1), 347. DOI: <https://doi.org/10.1186/1471-2458-8-347>
- Thomas, V., & Azmitia, M. (2019). Motivation matters: Development and validation of the Motivation for Solitude Scale–Short Form (MSS-SF). *Journal of adolescence*, 70, 33-42.
- Tluczek, A., Henriques, J. B., & Brown, R. L. (2009). Support for the reliability and validity of a six-item state anxiety scale derived from the State-Trait Anxiety Inventory. *Journal of nursing measurement*, 17(1), 19-28.

Yarkoni, T. (2009). Big correlations in little studies: Inflated fMRI correlations reflect low statistical power—Commentary on Vul et al. (2009). *Perspectives on Psychological Science*, 4(3), 294-298.

Zimet, G. D., Dahlem, N. W., Zimet, S. G., & Farley, G. K. (1988). The multidimensional scale of perceived social support. *Journal of personality assessment*, 52(1), 30-41.